# SOLVABLE GRADIENT FLOW THROUGH DIAGRAM EXPANSIONS

## ABSTRACT

We propose a general diagram-based approach to analyze scaling regimes and obtain explicit analytic solutions for gradient descent evolution in large learning problems. We focus on a class of problems in which an identity tensor is learned by gradient descent starting from a sum of rank-one tensors with random normal weights. A central element of our approach is to expand the loss evolution in a formal power series over time. The coefficients of this expansion can be described in terms of suitable diagrams akin to Feynman diagrams. Depending on the scaling of the initial weight magnitude and the number of parameters, we find several extreme learning regimes, such as NTK, mean-field, under-parameterized learning, and free evolution. These regimes include lazy training as well as strong feature learning. We identify these regimes with extreme points and sides of a hyperparameter polygon. We then show that in some of these regimes, the loss power series satisfies a formal partial differential equation. For certain scenarios, this equation is first order and can be solved by the method of characteristics, producing explicit loss evolution formulas that agree very well with experiment. We give a series of specific examples where this methodology is fully implemented.

## 1 INTRODUCTION

**Motivation and setting.** The purpose of this work is to propose a general framework for analytical studies of learning in large models with tractable scalable structure. The framework allows to identify and classify various learning regimes and, in a number of scenarios, obtain explicit analytical solutions for the evolution of the loss $L(t)$ under Gradient Flow (GF). More specifically, our motivation is as follows.

1. We are interested in large theoretical learning problems in which the size of the target to be learned is described by a large parameter $p$, while another large parameter $H$ controls the complexity of the model (e.g., the number of weights). We are interested in scenarios with an easily tractable pattern of target scalability with $p$. We assume that the weights $\mathbf{u} = \{u\}$ of the model are initialized with i.i.d. Gaussians with variance $\sigma^2$ and are learned by standard gradient flow with learning rate $1/T$:

$$\frac{du}{dt} = -\frac{1}{T}\partial_u L(\mathbf{u}). \tag{1}$$

2. We want to classify and characterize the different learning regimes according to the mutual scaling of $p, H$ and $\sigma$.

3. Ideally, we want to obtain *explicit* formulas of the loss $L(t)$ as a function of time under gradient flow. We expect such formulas to emerge in a large-$p$ limit. We are particularly interested in nonlinear (non-lazy-training) regimes that display feature learning.

In the present work we consider the following class of naturally scalable problems. We assume the target to be the diagonal identity tensor $F_{i_1,\dots,i_\nu} = \delta_{i_1=\dots=i_\nu}$, and the model to be the following, along with the standard quadratic loss:

$$f_{i_1,\dots,i_\nu} = \sum_{k=1}^{H}\prod_{m=1}^{\nu} u_{k,i_m}^{(m)}, \qquad L(\mathbf{u}) = \frac{1}{2}\sum_{i_1,\dots,i_\nu=1}^{p}(f_{i_1,\dots,i_\nu} - F_{i_1,\dots,i_\nu})^2. \tag{2}$$

That is, the model factorizes an order-$\nu$ tensor by contracting $\nu$ matrices (indexed with $(m)$) along their row-dimensions (indexed with $k$). Putting it differently, we are looking for a *canonical polyadic decomposition* (see A.5): the goal is to decompose the order-$\nu$ identity tensor into a sum of $H$ rank-1 tensors $f_{i_1,\ldots,i_\nu}^{(k)} = \prod_{m=1}^{\nu} u_{k,i_m}^{(m)}$ indexed with $k$.

We also consider a *symmetric* version of the model, for which

$$u_{k,i}^{(1)} = \ldots = u_{k,i}^{(\nu)} =: u_{k,i}. \tag{3}$$

If this symmetry condition is not imposed, we refer to this scenario as *asymmetric*.

The two especially notable cases are $\nu = 2, 3$. The $\nu = 2$ model describes learning to approximate an identity matrix with matrix products $UU^T$ (symmetric scenario) or $UV^T$ (asymmetric). The $\nu = 3$ model describes learning modular addition on the full data set in a Fourier mode representation (up to complex conjugates), see B. We review modular arithmetic in B.4.

Despite apparent simplicity, this class of problems is quite rich. It is connected to a number of known problems and models, such as (1) modular arithmetic as a supervised learning problem, (2) infinitely wide networks, (3) gradient descent dynamics on linear networks, (4) diagrammatic methods in Deep Learning, (5) parameter expansions, and (6) canonical polyadic tensor decomposition. However, our work is not based directly on any of the present works we are aware of, and we are not aware of any systematic studies of learning in this setting from the above mentioned perspective. We invite the reader to read our Appendix A, where we present an extensive survey of the above-mentioned topics.

**Our contribution.**

1. **[Diagram expansions, diagram calculus]** We develop a general method for analyzing the expected loss evolution $\mathbb{E}[L(t)]$ in such problems based on the formal expansion

$$\mathbb{E}[L(t)] \sim \sum_{s=0}^{\infty} \mathbb{E}\Big[\frac{d^s L}{dt^s}(0)\Big]\frac{t^s}{s!}, \tag{4}$$

   where $\mathbb{E}$ is with respect to Gaussian initialization of the weights. We show that the co-efficients in this expansion admit an interpretation and computation in terms of suitable *diagrams* akin to Feynman diagrams. We develop the combinatorics relevant for computing the expansion terms ("*diagram calculus*").

2. **[Pareto-optimal terms, hyperparameter polygon, extreme regimes]** As $p, H \to \infty$, the leading contribution to the expansion terms comes from *minimally contracted diagrams* and depends on the mutual scaling of $p, H$, and $\sigma$. We describe all scaling possibilities via a *hyperparameter polygon*. We identify the vertices and sides of this polygon with several extreme learning regimes, including NTK, mean-field and free evolutions.

3. **[Formal PDE and solutions by characteristics]** At least in some regimes, the power series Eq. (4) can be extended to a multivariate generating function admitting a formal partial differential equation. Moreover, in some regimes this equation is first-order and can be explicitly solved by the method of characteristics. We demonstrate such solutions for free evolutions (i.e. no-learning), as well as for several interacting regimes (i.e. with learning).

We emphasize that we are not aware of any other works where diagram expansions would be employed to usefully classify and analyze learning regimes and obtain explicit nonlinear solutions of the GF evolution.

## 2 LOSS EXPANSION AND DIAGRAM CALCULUS

**Diagrams.** Loss (2) is polynomial in the weights. We can write it as a sum of three terms:

$$L(\mathbf{u}) = \frac{1}{2} \sum_{i_1,\ldots,i_\nu=1}^{p} \sum_{k=1}^{H} \sum_{k'=1}^{H} \prod_{m=1}^{\nu} u_{k,i_m}^{(m)} u_{k',i_m}^{(m)} - \sum_{i=1}^{p} \sum_{k=1}^{H} \prod_{m=1}^{\nu} u_{k,i}^{(m)} + \frac{p}{2}. \tag{5}$$

Each of the three terms is a sum of monomials in indexed variables over their indices. Let us identify with each such expression (summed monomial) a *diagram*:

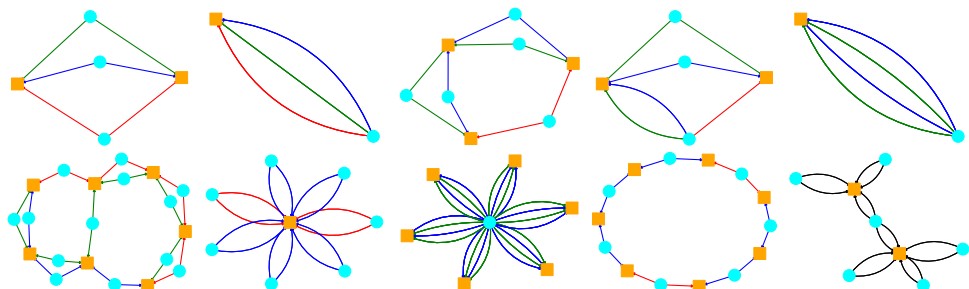

Figure 1: **Top row:** Diagrams in the asymmetric scenario with $\nu = 3$ (three colors). Yellow squares correspond to $H$-nodes and cyan circles to $p$-nodes. *Left to right:* $D_6$; $R_3$; diagrams appearing in $D_6 \star D_6$ (up to recoloring); in $D_6 \star R_3$; in $R_3 \star R_3$. **Bottom row:** generic diagrams in different regimes. *Left to right:* a diagram from $D_6^{\star s}$ (free evolution); a "flower" with one $H$-node (contracted, underparameterized); a "flower" with one $p$-node (contracted, overparameterized); circular ($\nu = 2$); circular contracted to a tree (symmetric case, $\nu = 2$).

1. The diagram is a multi-graph with nodes corresponding to summation indices (e.g., $i_m, k$) and edges corresponding to the respective weights (e.g., $u_{k,i_m}^{(m)}$).

2. There are two kinds of nodes: some ($i_1, i_2$, etc.) correspond to summation up to the target size $p$ – we call them $p$-*nodes*. Others ($k, k'$) correspond to summation up to the model size parameter $H$ – we call them $H$-*nodes*. An edge always connects a $p$-node with an $H$-node.

3. Edges can have $\nu$ different *colors* corresponding to the index $m$ of the weight. In symmetric models (3) all edges have the same color.

We review diagrammatic approaches in DL in Appendix A.3. A diagram implies summation of monomials over the respective node indices. The first term in the loss expansion (5) corresponds to a diagram with $\nu$ $p$-nodes, two $H$-nodes and $2\nu$ edges – let us call this diagram $D_{2\nu}$. The second term corresponds to a diagram with one $p$-node, one $H$-node and $\nu$ edges – let us call this diagram $R_\nu$. See illustration for $\nu = 3$ in top row of Fig. 1. The third term corresponds to the empty diagram. We will occasionally slightly abuse notation and identify diagrams with respective summed monomials. We can then schematically represent the loss (5) as

$$L = \frac{1}{2}D_{2\nu} - R_\nu + \frac{p}{2}. \tag{6}$$

**Loss evolution and diagram merging.** Representation (6) is written in terms of current weights. We, however, would like to write the loss in terms of initial, non-evolved, i.i.d. normal weights. To this end, recall the GF evolution law (1) showing that the time derivatives of the weight under GF is given by a suitable polynomial determined by $L$. It is convenient to formulate a respective general rule describing the time derivative $dG/dt$ of an arbitrary summed monomial associated with a diagram $G$. Specifically, we write this rule in terms of the *diagram merging operation* $\star$:

$$\frac{dG}{dt} = -\frac{1}{T}\sum_u \frac{\partial G}{\partial u}\frac{\partial L}{\partial u} = -\frac{1}{T}G \star (\tfrac{1}{2}D_{2\nu} - R_\nu + \tfrac{p}{2}) = -\frac{1}{T}G \star (\tfrac{1}{2}D_{2\nu} - R_\nu). \tag{7}$$

Here, given two diagrams $G_1$ and $G_2$, we define $G_1 \star G_2$ as the sum of diagrams resulting by choosing two edges in $G_1, G_2$ of matching colors, removing these edges, and identifying the respective $p$- and $H$-nodes. The operation $\star$ is bilinearly extended to linear combinations of diagrams, i.e. $(\sum_k c_k G_k) \star (\sum_{k'} c'_{k'} G'_{k'}) = \sum_{k,k'} c_k c'_{k'} G_k \star G_{k'}$.

For example, in the asymmetric scenario $D_6 \star R_3$ consists of $3 \cdot 2 = 6$ diagrams with 5 nodes and 7 edges, while in the symmetric scenario $D_6 \star R_3$ consists of $6 \cdot 3 = 18$ diagrams. See Fig. 1 for the examples of diagrams occuring in $D_6 \star D_6, D_6 \star R_3, R_3 \star R_3$.

The formalism of merging allows us to write the loss at any time $t$ as the formal expansion

$$L(t) \sim \sum_{s=0}^{\infty} \frac{d^s L}{dt^s}(0)\frac{t^s}{s!} = \frac{p}{2} + \sum_{s=0}^{\infty}(\tfrac{1}{2}D_{2\nu} - R_\nu)^{\star(s+1)}\frac{(-t)^s}{T^s s!}. \tag{8}$$

Here the diagrams $D_{2\nu}, R_\nu$ involve initial rather than current weights. We review closely related expansions in Appendix A.4.

**Expectations, edge pairing, and diagram contraction.** We are interested in the expected loss

$$\mathbb{E}[L(t)] \sim \frac{p}{2} + \sum_{s=0}^{\infty} \mathbb{E}[(\tfrac{1}{2}D_{2\nu} - R_\nu)^{\star(s+1)}]\frac{(-t)^s}{T^s s!}. \tag{9}$$

Each coefficient $\mathbb{E}[(\tfrac{1}{2}D_{2\nu} - R_\nu)^{\star(s+1)}]$ in this power series is an expectation of a Gaussian polynomial in initial weights represented by a linear combination of diagrams. Let us analyze how to compute the expectation $\mathbb{E}[G]$ for one such diagram $G$. By Wick theorem, the expectation $\mathbb{E}[X_1 \cdots X_{2l}]$ of a product of jointly Gaussian variables equals the sum of products $\prod_{r=1}^{l} \mathbb{E}[X_r X_{\phi(r)}]$ over all partitions $\{1, \ldots, 2l\} = \sqcup_{r=1}^{l}\{r, \phi(r)\}$ of the variables into disjoint pairs. In our context, since the weights are represented by edges, this means that we need to consider all *edge pairings* in the diagram $G$. Since the initial weights are independent, a paring has a non-vanishing contribution only if the paired edges have a matching color ($m$). Moreover, the expectation $\mathbb{E}[u_{k,i_r}^{(m)} u_{k',i_{r'}}^{(m)}]$ vanishes unless $k = k'$ and $i_r = i_{r'}$, meaning that we need to impose these additional constraints when summing over the indices. This shows that $\mathbb{E}[G]$ can be computed as follows.

1. Consider all possible pairings of edges with matching colors.

2. For each such pairing, identify ("*contract*") the diagram nodes corresponding to paired edges. (Note that an identification can only involve same-type nodes, i.e. $p$-nodes are never identified with $H$-nodes).

3. If the resulting diagram is left with $n$ $H$-nodes and $q$ $p$-nodes, and the number of edges is $2l$, then this pairing contributes the term $p^q H^n \sigma^{2l}$ to $\mathbb{E}[G]$.

We will henceforth refer to the identification of some of the same-type nodes as *diagram contraction*.

**Leading terms and Pareto-minimal contractions.** The above procedure computes the coefficients in expansion (9) as linear combinations of terms $p^q H^n \sigma^{2l}$. Here, $2l$ is the number of edges in a diagram and does not depend on the contraction. On the other hand, the powers $q$ and $n$ in $p^q H^n$ do depend on the contraction. To simplify the combinatorics of contractions, it is natural to only consider *Pareto-optimal pairings* and *Pareto-minimal contractions* that corespond to leading terms under our assumption $p, H \to \infty$.

Specifically, we say that an edge pairing in $G$ is *Pareto-optimal* if the factor $p^q H^n$ induced by the respective contraction is not dominated by the factor $p^{q'} H^{n'}$ of any other pairing, that is, if there are no pairings such that $q' \geq q, n' \geq n$, and at least one of these inequalities is strict. This means that if $p, H \to \infty$, the leading terms in $\mathbb{E}[G]$ will be given by some of the Pareto-optimal pairings. Given a specific scaling relation between $p$ and $H$ (a "learning regime"), some particular Pareto-optimal pairings will provide leading contributions to $\mathbb{E}[G]$. For example, if one pairing has factor $p^2 H^3$ and another $p^3 H^2$, then the first pairing will provide the leading contribution if $H \gg p$. At the same time, a pairing with factor $p^2 H^2$ is Pareto-suboptimal and will never provide a leading contribution.

We call, accordingly, a contraction *Pareto-minimal* if it corresponds to one or more Pareto-optimal pairings. We also call the respective contributions $p^q H^n$ (or $p^q H^n \sigma^{2l}$) Pareto-optimal. In the sequel, we will only consider Pareto-optimal components in the coefficients $\mathbb{E}[(\tfrac{1}{2}D_{2\nu} - R_\nu)^{\star(s+1)}]$.

## 3 CLASSIFICATION OF LEARNING REGIMES

**Pareto frontier of leading terms.** We will now describe the Pareto-optimal terms $p^q H^n \sigma^{2l}$ appearing in the coefficient $\mathbb{E}[(\tfrac{1}{2}D_{2\nu} - R_\nu)^{\star(s+1)}]$. The description depends on the presence of symmetry in the model, and for symmetric models also on the parity of the model order $\nu$. The case of symmetric models with odd $\nu$ is more complicated, and we skip it in the present paper.

Observe that the power $2l$ is simply the number of edges in a diagram and can be computed by binomially expanding the expression $(\tfrac{1}{2}D_{2\nu} - R_\nu)^{\star(s+1)}$. If a diagram $G$ results from merging $s_D$ diagrams $D_{2\nu}$ and $s_R$ diagrams $R_\nu$ in some order, then $G$ has $2l = 2\nu s_D + \nu s_R - 2s$ edges.

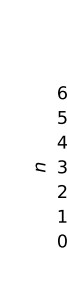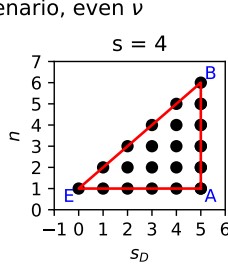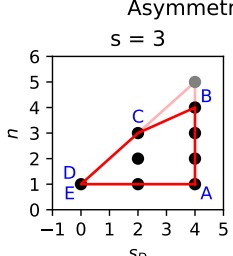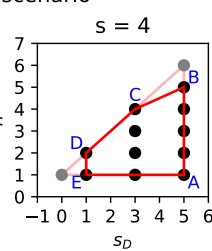

Figure 2: Pareto optimal points (black) and the corresponding hyperparameter polygon (red) for the symmetric scenario (i.e. when the matrices are shared), left, and for the asymmetric one, right; for a few small values of $s$. The extremal points we had to remove from the triangle when switching to the asymmetric scenario are colored gray.

| Simplex | Scaling condition | Natural $T$ | Parame­terization | Learning | Interpretation |
|---------|-------------------|-------------|-------------------|----------|----------------|
| A – B | $H \asymp p^{\nu-1}, p^{\nu-1}\sigma^\nu \to \infty$ | $H\sigma^{2\nu-2}$ | Balanced | No | Free evolution |
| B – C | $p^{\nu-1}H\sigma^{2\nu} \asymp 1, H\sigma^\nu \to \infty$ | $H\sigma^{2\nu-2}$ | Over- | Lazy | NTK |
| C | $p^{\nu-1}H\sigma^{2\nu} \to 0, H\sigma^\nu \to \infty$ | $H\sigma^{2\nu-2}$ | Over- | Lazy | NTK, $f(0) \equiv 0$ |
| C – D | $H\sigma^\nu \asymp 1, H/p^{\nu-1} \to \infty$ | $\sigma^{\nu-2}$ | Over- | Rich | Mean-field |
| D – E | $H \asymp p^{\nu-1}, H\sigma^\nu \to 0$ | $\sigma^{\nu-2}$ | Balanced | Rich | — |
| E – A | $p^{\nu-1}\sigma^\nu \asymp 1, H/p^{\nu-1} \to 0$ | $\sigma^{\nu-2}$ | Under- | Rich | — |

Table 1: Extremal simplices along with their interpretations; asymmetric scenario. See Table 2 for a similar table covering both scenarios.

Since $s_D + s_R = s + 1$, we can equivalently write this number as $2l = \nu(s_D + 1) + (\nu - 2)s$. If both $\nu$ and $s_R$ are odd, then the number of edges is odd so that such diagrams do not contribute to $\mathbb{E}[(\frac{1}{2}D_{2\nu} - R_\nu)^{\star(s+1)}]$. Also note that the power $n$ of $H$ can only take values between 1 and $s_D + 1$, since merging a diagram with $D_{2\nu}$ adds one $H$-node while merging with $R_\nu$ adds no $H$ nodes.

A complete description of the Pareto optimal terms is given by the following theorem.

**Theorem 1** (E). *Up to nonzero numerical coefficients, Pareto-optimal terms in* $\mathbb{E}[(\frac{1}{2}D_{2\nu} - R_\nu)^{\star(s+1)}]$ *have the form*

$$p^{Q(n,s_D)}H^n\sigma^{\nu(s_D+1)+(\nu-2)s},\tag{10}$$

*where* $0 \leq s_D \leq s+1, 1 \leq n \leq s_D + 1$ *and*

$$Q(n,s_D) = \begin{cases} 1 + (\nu-1)s_D - \frac{\nu}{2}(n-1), & \text{symmetric models, even } \nu, \\ 1 + (\nu-1)(s_D + 1 - n), & \text{asymmetric models}. \end{cases}\tag{11}$$

*In the symmetric scenario with even* $\nu$ *all of these terms occur. The asymmetric scenario has excep­tional terms that do not occur: a) terms with odd* $s_R = s+1-s_D$*; b) term* $(n, s_D) = (s+2, s+1)$*.*

**Hyperparameter polygon.** For a fixed value of $s$, the Pareto-optimal terms found in the above theorem could be conveniently visualized as points on a two-dimensional grid with axes associated with $n$ and $s_D$. We refer to the convex hull of this set of points as *hyperparameter polygon*: Fig. 2.

Although Pareto-optimal terms dominate all others as $p, H \to \infty$, a subset of these terms may become dominant relative to others in this limit, depending on the relative rate of divergence of $p$, $H$, and $1/\sigma$. To be precise, only an extremal simplex of the hyperparameter polygon (i.e. a vertex or an edge), or the whole polygon could become dominant as $p$, $H$, and $1/\sigma$ diverge at the same time.

These extremal simplices allow for natural interpretations summarized in Table 1 for the asymmetric scenario, and in Table 2 for both scenarios. For example, some of the edges correspond to limit

regimes that previously appeared in the literature: namely, the constant NTK and the mean-field regimes; see Appendix C for a detailed description and derivations of the scaling conditions for each extremal simplex, and Appendix A.1 for a detailed literature overview on limit regimes.

For each of the identified limit regimes, there is a natural scaling for the inverse learning rate $T$ that leads to a nondegenerate evolution of the average loss as a function of $t$. We obtained it by balancing the base of power $s$ in the leading Pareto-term provided by Theorem 1 with $T$. See Appendix C for details.

Here we describe only the high-level properties of the hyperparameter polygon. As Theorem 1 claims, it is a triangle in the symmetric scenario, while in the asymmetric one, one or two summits have to be removed, which turns the polygon either into a quadrangle, or into a pentagon, see Fig. 2.

As one moves from left to right, $s_D$ increases, hence $s_R = s + 1 - s_D$ decreases. The rightmost edge, A-B, corresponds to $s_R = 0$, which means that the target is never used for computing loss derivatives, hence the target is never learned. This edge becomes dominant when initialization is large: in this case, the identity target is essentially zero compared to the initial model, and the model attempts to learn zero throughout most of its training process. We call this regime *free evolution*.

Moving left gives $s_R > 0$, and we expect learning to occur. When only points with $s_R$ bounded from above by a constant independent on $s$ become dominant, we say that *weak learning* occurs: loss derivatives do depend on the target but only weakly (polynomially with bounded degree). A notable qualitative difference between the symmetric and the asymmetric scenarios is that weak learning exists in the latter, but does not exist in the former. Indeed, Point C and Edge B-C are extremal simplices with $s_R = 2$ and $s_R \in \{0, 2\}$, respectively. On the other hand, the only extremal simplices that contain $s_R > 0$ in the symmetric scenario are Edges B-E and E-A, Point E, and the whole triangle A-B-E. All three contain Point E with $s_R = s + 1$.

A special case is $\nu = 2$ which corresponds to matrix factorization; in this case, the model is a conventional two-layer fully-connected network. For Edge B-C to become dominant, one needs $\sigma^2 \asymp 1/\sqrt{pH}$, which is a conventional NTK scaling that results in a linearized (or, *lazy*) training in the limit of inifinite $H$: see Appendix A.1. In App. D, we give an alternative explanation for why linearized training does not exist in the symmetric scenario, but does exist in the asymmetric one.

Following previous works (e.g. Chizat et al. (2019)), we call the training process *rich* if it is not lazy: i.e. when higher-order target correlations occur when computing loss derivatives. Important examples of rich regimes are Edges B-E (symmetric) and C-D (asymmetric), who require $\sigma^2 \asymp 1/H$ and $\sigma^2 \asymp 1/H^{2/\nu}$, respectively. For $\nu = 2$, these scalings coincide and yield a mean-field limit discovered independently in a number of works: see Appendix A.1.

Moving bottom-up on the hyperparameter polygon corresponds to increasing $n$. As Theorem 1 claims, the larger $n$, the more important is growing $H$ over growing $p$. For this reason, top edges, i.e. B-E (resp. B-C and C-D), correspond to overparameterized regimes, while the bottom one, E-A, is an underparameterized one. In turn, side edges, A-B and E (resp. D-E), give balanced regimes. We note that the balancedness condition depends on $\nu$: $H^2 \asymp p^\nu$ (resp. $H \asymp p^{\nu-1}$), while $H = p$ is always necessary and sufficient to fit an identity tensor for any $\nu$.

## 4 EXPLICIT SOLUTIONS

**General methodology.** We propose a general methodology that allows us to deduce an explicit loss evolution $L(t)$ from the diagram expansion in multiple scenarios and regimes.

1. We write a recurrence relation for the formal coefficients of the loss expansion. In these coefficients, we generally keep only the Pareto-optimal terms, and further drop terms that are subleading in the particular regime under considerations.

   Typically, such a recurrence relation cannot be written without considering additional auxiliary variables. Accordingly, this requires us to consider the diagram expansion of the loss as a special case of a more general generating function (g.f.) $f(\mathbf{x})$.

2. Using Theorem 2 below, we convert the recurrence relation into a (formal) partial differential equation (PDE) on $f$.

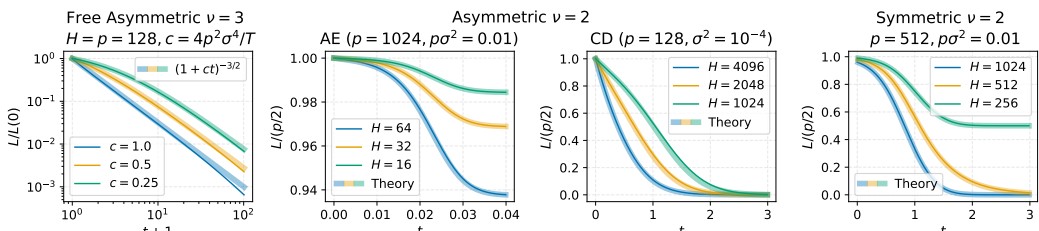

Figure 3: Experimental confirmation of theoretical loss evolution formulas from Section 4. Thin lines: experimental data; thick lines: theory. *Left to right:* (1) Eq. (17), (2) Eqs. (20) and (21), (3) Eq. (22), (4) Eqs. (23) and (28). See App. K for details about experimental part of our work.

3. In many cases, the resulting PDE is a first-order PDE of the form

$$\Phi(\mathbf{x})^T \nabla f(\mathbf{x}) = \phi(\mathbf{x}) \tag{12}$$

with some particular vector field $\Phi(\mathbf{x})$ and function $\phi(\mathbf{x})$. Such an equation can be solved by the method of characteristics:

$$f(\mathbf{x}(\tau_0)) = f(\mathbf{x}(\tau_1))e^{-\int_{\tau_0}^{\tau_1} \phi(\mathbf{x}(\tau))d\tau}, \tag{13}$$

where $\mathbf{x}(\tau)$ is any integral curve of the field $\Phi$, i.e. a solution of the ordinary differential equation (ODE) $\frac{d}{d\tau}\mathbf{x}(\tau) = \Phi(\mathbf{x}(\tau))$. We use Eq. (13) to transfer the values of $f$ from the points $\mathbf{x}(\tau_1)$ where we know $f$ to points $\mathbf{x}(\tau_0)$ of our interest.

While this method is not entirely rigorous, we show that it is applicable in many scenarios, and when applicable, the resulting solution agrees well with numerical gradient descent (Fig. 3 and App. K).

The PDE on g.f. $f$ is derived by the following general theorem (see F for proof and comments).

**Theorem 2.** *Given a formal multivariate power series $f(\mathbf{x}) \sim \sum_{\mathbf{n}\in\mathbb{N}_0^d} C_\mathbf{n}\mathbf{x}^\mathbf{n}$, suppose that its coefficients satisfy the conditions $\sum_{\mathbf{k}\in K} C_{\mathbf{n}+\mathbf{k}}P_\mathbf{k}(\mathbf{n}+\mathbf{k}) = 0, \quad \forall \mathbf{n} \in \mathbb{Z}^d$, where $K$ is a finite subset of $\mathbb{Z}^d$, $P_\mathbf{k}$ are some $d$-variate polynomials, and $C_\mathbf{n} = 0$ if $\mathbf{n} \in \mathbb{Z}^d \setminus \mathbb{N}_0^d$. Then $f$ formally satisfies the differential equation $\sum_{\mathbf{k}\in K} \mathbf{x}^{-\mathbf{k}}P_\mathbf{k}(\mathbf{x}\frac{\partial}{\partial\mathbf{x}})f = 0$.*

**Free evolutions. (G)** The simplest illustration of the above methodology is for free evolutions in the underparameterized setting. In this case, no auxiliary variables are needed and the resulting PDE is an easily solved first-order differential equation in one variable.

Indeed, free evolutions can be obtained by removing the target from loss (2) and accordingly removing the target interaction diagrams $R_\nu$ from expansion (9):

$$\mathbb{E}[L(t)] \sim \sum_{s=0}^{\infty} \mathbb{E}[(\tfrac{1}{2}D_{2\nu})^{\star(s+1)}]\frac{(-t)^s}{T^s s!} \sim \sum_{s=0}^{\infty} \mathbb{E}[D_{2\nu}^{\star(s+1)}]\frac{(-t)^s}{2^{s+1}T^s s!}. \tag{14}$$

In the underparameterized setting ($H \ll p^{\nu-1}$ in the asymmetric or $H \ll p^{\nu/2}$ in the symmetric even-$\nu$ scenario), by Theorem 1, the leading terms in the diagram expansions are given by contracted diagrams with $n = 1$ contracted $H$-node. Such contracted diagrams have a simple "flower-like" structure that allows to easily connect the coefficients $\mathbb{E}[D_{2\nu}^{\star(s+1)}]$ for neighboring values of $s$.

Consider, for example, the asymmetric underparameterized scenario. The respective leading contracted diagrams in $D_{2\nu}^{\star(s+1)}$ are "flowers" with a single $H$-node and $\nu + (\nu - 1)s$ "petals", each consisting of a $p$-node and a pair of same-color edges (see Fig. 1). These diagrams are obtained from the diagrams in $D_{2\nu}^{\star s}$ by merging with $D_{2\nu}$ over all edges in $D_{2\nu}^{\star s}$ and one of the two edges of the matching color in $D_{2\nu}$, giving the recurrence

$$\mathbb{E}[D_{2\nu}^{\star(s+1)}] \sim 4(1 + (\nu - 1)s)p^{\nu-1}\sigma^{2\nu-1}\mathbb{E}[D_{2\nu}^{\star s}]. \tag{15}$$

Theorem 2 and expansion (14) then give an easily solvable (formal) ODE

$$-T\frac{d}{dt}\mathbb{E}[L(t)] = 2p^{\nu-1}\sigma^{2\nu-1}(\nu + (\nu - 1)t\frac{d}{dt})\mathbb{E}[L(t)]. \tag{16}$$

The arguments in the symmetric even-$\nu$ scenario are similar. We thus obtain

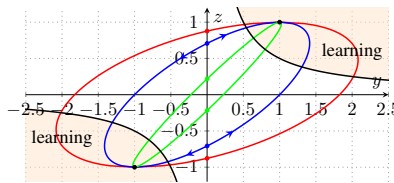

Figure 4: Elliptic characteristics of PDE (12) in the $yz$ plane for underparameterized asymmetric $\nu = 2$ scenario. "learning" indicate the zones where the loss drops below the loss $\frac{p}{2}$ of the trivial zero model. See H for details.

**Theorem 3.** *For free evolution in the underparameterized setting (i.e., keeping only contracted diagrams with $n = 1$ H-node in the loss expansion), the expected loss is given by*

$$\mathbb{E}[L(t)] \sim \frac{p^\nu H \sigma^{2\nu}}{2} \begin{cases} (1 + \frac{2(\nu-1)p^{\nu-1}\sigma^{2(\nu-1)}t}{T})^{-\frac{\nu}{\nu-1}}, & asymmetric, \\ (1 + \frac{2\nu(\nu-1)p^{\nu-1}\sigma^{2(\nu-1)}t}{T})^{-\frac{\nu}{\nu-1}}, & symmetric. \end{cases} \tag{17}$$

Here we observe a natural scaling for $T$ for free evolutions, summarized in Tables 1 and 2 (note that $p^{\nu-1} \asymp H$ for asymmetric free evolutions).

**Underparameterized asymmetric model with $\nu = 2$. (H)** This model involves interactions with the target through diagrams $R_2$. As an underparameterized model, it only involves contracted diagrams having the form of "flowers" with "petals" consisting of a $p$-node and two edges of two possible colors. However, combinatorics now is more complicated than in the free case. Merging with $R_2$ has the effect of recoloring one of the edges. For the diagram to admit a pairing without additional contractions, all petals must have same-color edges. Let $q$ denote the number of differently-colored petals, and consider a generalized generating function

$$f(x, y, z) = \sum_{s_R=0}^\infty \sum_{q=0}^{s_R} \sum_{s=\max(0,s_R-1)}^\infty C_{s_R,q,s} x^s y^{s_R} z^q, \tag{18}$$

where $C_{s_R,q,s}s!$ is the combinatorial factor corresponding to diagrams with $q$ differently-colored petals in expansion terms of $(\frac{1}{2}D_4 + R_2)^{\star(s+1)}$ with exactly $s_R$ factors $R_2$. It is easy to derive a recurrence for the coefficients $C_{s_R,q,s}$. Theorem 2 then gives first-order PDE (12) with

$$\Phi(x, y, z) = \begin{pmatrix} 1+2xyz-2x \\ 2(y-y^2z) \\ 2(y-yz^2) \end{pmatrix}, \quad \phi(x, y, z) = 4(1 - yz). \tag{19}$$

The corresponding characteristic ODE is explicitly solvable; the characteristics are elliptic in the $yz$ plane (see Figure 4). The model loss is expressed in terms of the generating function $f$ by

$$\mathbb{E}[L(t)] \sim \frac{p}{2} + p^2 H \sigma^4 f(-p\sigma^2 t/T, 1/(p\sigma^2), 0), \tag{20}$$

so we need to find $f$ on the plane $z = 0$. On the other hand, we know $f$ on the plane $x = 0$, since Eq. (18) implies $f(0, y, z) = \frac{1}{2} - yz$. We transfer the values of $f$ from the plane $x = 0$ to the plane $z = 0$ along characteristics using Eq. (13) and obtain explicit solution

$$f(x, y, 0) = y^2 \frac{1 + 2yr - r^2}{2(y - r)^2}, \quad r = \sinh(2xy). \tag{21}$$

**Overparameterized asymmetric model with $\nu = 2$. (I)** We solve this model in a similar way as the underparameterized one above. It involves contracted diagrams which are again "flowers", but the petals consist of $H$ nodes instead of $p$ ones, while there is a $p$ node at the flower center. By performing similar steps as above, we arrive at the following asymptotic for the average loss:

$$\mathbb{E}[L(t)] \sim \frac{p}{2} \left( \frac{1 + \rho^2}{1 + \rho^2 \cosh\psi + \rho\sqrt{1 + \rho^2} \sinh\psi} \right)^2, \quad \psi = \frac{2t}{T}\sqrt{1 + \rho^2}, \quad \rho = H\sigma^2. \tag{22}$$

**General symmetric model with $\nu = 2$. (J)** The symmetric $\nu = 2$ model can be solved by our method for general $p, H \to \infty$ without assuming extreme under/over-parameterization or strong/weak target interaction regimes (i.e. without discarding any Pareto-optimal diagrams). The uncontracted diagrams resulting in $(\frac{1}{2}D_4 - R_2)^{\star(s+1)}$ in this case are single-color circular graphs of length $2s_D + 2$ with alternating $s_D + 1$ $p$-nodes and $s_d + 1$ $H$-nodes, where $s_D$ is the number of factors $D_4$. Minimal contractions contract these loops to trees. The loss can be represented as

$$\mathbb{E}[L(t)] \sim \frac{p}{2} + p^2\sigma^2\Psi(-t/T, H/p, p\sigma^2) \tag{23}$$

with the generating function

$$\Psi(x, y, z) = \sum_{s_D=0}^{\infty} z^{s_D} \sum_{s=\max(0,s_D-1)}^{\infty} \frac{M_{s,s_D}}{s!}x^s \sum_{n=1}^{s_D+1} N_{s_D+1,n}y^n, \tag{24}$$

where $M_{s,s_D}$ is the combinatorial uncontracted diagram-count coefficient resulting from $\mathbb{E}[(\frac{1}{2}D_4 - R_2)^{\star(s+1)}]$ with exactly $s_D$ factors $\frac{1}{2}D_4$, while $N_{m,n}$ is the number of minimal contractions of circular graphs of length $2m$ with alternating $p$- and $H$-nodes to trees with $n$ $H$-nodes.

A notable property of this 3-variable g.f. $\Psi$ is that its coefficients $\frac{M_{s,s_D}N_{s_D+1,n}}{s!}$ are partially factorized, which allows to express $\Psi$ in terms of two 2-variable g.f.'s $f$ and $h$:

$$\Psi(x, y, z) = \frac{1}{2\pi i}\oint_\gamma h(\zeta, y)f\left(x, \frac{z}{\zeta}\right)\frac{d\zeta}{\zeta}, \tag{25}$$

$$f(x, z) \sim \sum_{s=0}^{\infty}\sum_{s_D=0}^{s+1}\frac{M_{s,s_D}}{s!}x^s z^{s_D}, \quad h(z, y) \sim \sum_{s_D=0}^{\infty}\sum_{n=1}^{s_D+1}N_{s_D+1,n}z^{s_D}y^n. \tag{26}$$

The combinatorial numbers $N_{m,n}$ are known as *Narayana numbers*, and their g.f. $h$ is known:

$$h(z, y) = \frac{1 - z(y+1) - \sqrt{1 - 2z(y+1) + z^2(y-1)^2}}{2z^2}. \tag{27}$$

On the other hand, the g.f. $f$ satisfies PDE (12) with explicit $\Phi, \phi$ and can again be found by transfering along characteristics by (13). Performing integration in (25) by a residue computation, we find an explicit form of $\Psi$:

$$\boxed{\Psi(x, y, z) = \left(\frac{ze^{-4x}}{2}\partial_z h(z(1 - e^{-4x}), y) - h(z(1 - e^{-4x}), y)\right)e^{-4x}.} \tag{28}$$

## 5 SUMMARY, FUTURE WORK, AND LIMITATIONS

We presented a general method for analyzing the average loss $\mathbb{E}[L(t)]$ evolution by combining a formal power series expansion and a diagram calculus similar to Feynman diagrams for computing asymptotics of each term. We applied our method to the problem of factorizing the identity tensor. Despite its specificity, this problem exhibits a rich family of limit regimes. We derived a *complete* classification of these regimes. Some of them (e.g. NTK, mean-field) were previously studied for conventional neural architectures, while others did not receive much attention. Our diagrammatic approach allowed us to derive the average loss *explicitly* as a function of training time in the specific case of matrix factorization for most of the identified limit regimes.

Due to a lack of space, we have not included several other results (e.g., solutions for higher-order tensors and overparameterized free models, a detailed diagrammatic picture of NTK). Considering future work, our diagrammatic method is directly applicable to more general scenarios, including (1) GF with momentum and weight decay, (2) unbalanced initialization (as in, e.g., Kunin et al. (2024)), (3) deep linear nets (see A.2), (4) incomplete training dataset. The last option immediately pushes us forward to the question of *generalization*: arriving at an explicit evolution of the average *distribution* loss will make our model one of the few (if not the only) *nonlinear* models exhibiting *feature learning* whose generalization ability is precisely quantified. We discuss some limitations of our approach in L.

## REPRODUCIBILITY STATEMENT

We have taken several steps to ensure the reproducibility of our work:

1. All necessary experimental details are provided in Appendix K.
2. We include example Jupyter notebooks in the supplementary materials, which illustrate the experimental pipeline and enable verification of our results.

We believe these resources will facilitate successful reproduction and validation of the findings presented in this paper.

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

CONTENTS

## A  LITERATURE SURVEY

Our work is not based directly on any of the present works we are aware of. Nevertheless, it draws connections with numerous topics, namely: (1) modular arithmetic as a supervised learning problem, (2) infinitely wide networks, (3) gradient descent dynamics on linear networks, (4) diagrammatic methods in Deep Learning, (5) parameter expansions, and (6) canonical polyadic decomposition. We overview the first topic in Appendix B and all the rest below.

### A.1  INFINITELY WIDE NETWORKS

**Neural Tangent Kernel.**  A pioneering work of Jacot et al. (2018) demonstrated that under a specific parameterization and in the limit of infinite width, the process of training a fully-connected neural network is equivalent to a kernel gradient descent for a specific kernel coined *Neural Tangent Kernel, or NTK*. Notably, the required parameterization is non-standard; because of this, the equivalence does not hold for standartly parameterized neural nets. Since a kernel method could be seen as a linear model in a large (but fixed) feature space, the absence of a kernel method equivalence means that neural networks parameterized in a standard way, in fact, do learn features during training. This fact is believed to explain why neural nets perform better in some tasks compared to conventional kernel methods.

Two of the limit regimes we identify in our setting, namely, that corresponding to Point C and Edge B-C in the asymmetric scenario (see Fig. 2, right), correspond to the constant NTK limit of Jacot et al. (2018). Indeed, Point C corresponds only to diagrams in the loss expansion resulted from merging exactly two sub-diagrams $R_\nu$ together with any number of sub-diagrams $D_{2\nu}$. This implies that the loss at any time step $t$ depends only quadratically on the target. This is possible only when the learned weights depend linearly on the target. See Appendix C for further details.

**Mean-field limit.**  A number of works demonstrate that the gradient flow dynamics on a two-layer MLP is equivalent to an evolution of measure: Chizat & Bach (2018); Rotskoff & Vanden-Eijnden (2018); Sirignano & Spiliopoulos (2020). Under a specific parameter scaling, the moments of this measure stay finite as the number of neurons in the hidden layer grows. Thanks to this fact, a limit of infinite width comes naturally. Note that the required parameterization is different from that required for the NTK equivalence of Jacot et al. (2018). While the scaling leading to the NTK regime corresponds to Edge B-C on the hyperparameter polygon, that leading to the mean-field regime corresponds to Edge C-D in the asymmetric case, of Edge B-E in the symmetric one. See Appendix C for further details.

The name "mean-field" comes from the fact that the measure interacts with itself not directly, but through interacting with a scalar variable (namely, the loss) which it forms itself. This is in line with a mean-field approximation in particle physics, where particles are assumed not to interact with each other directly, but to interact with a field formed by these particles collectively (e.g. charged particles collectively form a magnetic field and interact with it).

The main advantage of the mean-field limit is that it allows for feature learning (that is, the associated NTK changes throughout training). However, the constructions of the above-mentioned works do not allow for a straightforward generalization to MLPs beyond two layers. Nevertheless, several works propose various formulations of the gradient flow in terms of a measure evolution for deep nets: see Araújo et al. (2019); Nguyen (2019); Sirignano & Spiliopoulos (2022); Nguyen & Pham (2023). We note also a $\mu P$ limit (Yang & Hu, 2021) which is well-defined for deep nets and coincides with the mean-field one for two-layered nets, but does not allow for a physical interpretation in terms of measures as above.

**Classification of infinite-width regimes.** Note that the network parameterizations required for the NTK limit and the $\mu P$ limit are different and lead to qualitatively different behavior: in the former case, the model becomes equivalent to a kernel method, hence does not learn features, while in the latter case it provably does (Yang & Hu, 2021). Are there are any other meaningful infinite-width regimes? One of the contribution of the present work is a classification of infinite $p$ and $H$ regimes for identity tensor decomposition: see Theorem 1 visualised with a hyperparameter polygon of Fig. 2. A few earlier works (Golikov, 2020a;b) classified all possible infinite-width limits (in our terms, only $H$ is infinite) for a two-layer net. It follows from their classification that all other limits are in some sense degenerate. Later, Yang & Hu (2021) provided an alternative classification for multi-layer nets.

## A.2 GRADIENT DESCENT DYNAMICS ON LINEAR NETWORKS

In the matrix case, one could think of the problem we study as training a linear network $f_{U,V}(x) = V^\top U x$ to fit an identity map. Linear networks, i.e. MLPs with identity activation functions, attracted a lot of attention during recent years. The reason is that even though they cannot express nonlinear maps, their training dynamics is nonlinear and might exhibit features which are typical for nonlinear nets, but could not be observed for a one-layer linear model.

In contrast to generic non-linear nets, the training dynamics of linear ones could be solved in several important cases which we review below. One of the main contributions of the present work is explicit solutions for our tensor decomposition model in various limit regimes, see Section 4. We underline that our model is a linear net with one hidden layer for $\nu = 2$, but not a linear net for $\nu \geq 3$.

The study of gradient descent dynamics for multi-layer linear nets dates back to a seminal work of Fukumizu (1998) who presented an exact analytic solution for gradient flow under a crucial assumption of *balanced initialization*. This assumption is quite specific and not satisfied by a random Gaussian initialization. Their solution was later revised by Braun et al. (2022) and generalized to $\lambda$-balanced initializations by Dominé et al. (2024).

Whereas the balancedness assumption of Fukumizu (1998) does not depend on data, an alternative solution proposed by Saxe et al. (2013) assumed the initial linear map to be aligned with the eigenvectors of the feature correlation matrix of the training dataset. If this is the case, the above alignment property holds throughout the whole training process, and the whole parameter evolution separates into independent scalar evolutions.

While the above assumption is neither satisfied automatically for standard initialization strategies, it is trivially satisfied for zero initialization. Though zero is a saddle point from which the training process cannot start, one can hope that initializing near the origin also approximately keeps the alignment property. Saxe et al. (2013) empirically observed this claim, while Braun et al. (2022) proved it for balanced initializations. We underline that the explicit solutions we present in Section 4 hold for a generic Gaussian initialization without any additional constraints.

**Saddle-to-saddle regime.** An interesting feature of aligned near-zero initialization is that in this case, the gradient flow empirically exhibits a peculiar dynamics. It starts near a saddle point (the origin), stays there for a while, then escapes it until it gets stuck near another saddle point, where the process repeats. The training process ends when the gradient flow arrives at a minimum instead of a saddle. Each saddle corresponds to learning a finite number of strongest principal components of the data. Hence, the data are learned sequentially. This regime is coined *saddle-to-saddle* and is studied in a number of works, including Li et al. (2020); Jacot et al. (2021).

Since we consider only identity targets, all our principal components have equal strength, and are learned the same time. Therefore the only saddle we encounter during the optimization process is that in the origin. However, we claim that our analysis generalizes easily to a number of other targets, including $F^0_{i_1,\ldots,i_\nu} = \delta_{i_1=\ldots=i_\nu=1}$, which corresponds to a tensor with a single "one" on a diagonal with all the rest entries being zeros, and a mix of the latter and the identity target: $F = F^I + \alpha F^0$ for some $\alpha > 0$ possibly depending on $p$ and $H$. In the latter case, the target has two distinct eigenvalues: 1 and $1 + \alpha$. Here, we expect to observe a non-trivial saddle-to-saddle dynamics, for which the strong component is learned first (it corresponds to the first non-trivial saddle in the weight evolution), while the rest are learned afterwards.

### A.3 Applications of diagrams to Deep Learning theory

Diagrams turn out to be a handy tool we use in our analysis to get leading terms of loss time derivatives for large $H$ and $p$; see examples in Fig. 1 and the discussion of our approach in Section 2. Diagrammatic approaches akin to Feynman diagrams in particle physics turned out to be a natural tool for studying small deviation corrections for limits of certain quantities. In Deep Learning, the most well-studied limit is the infinite width limit in the NTK regime (see the discussion above).

Before proceeding with a discussion of works that compute finite-width corrections for limit NTKs using diagrammatic approaches, we underline that in contrast to expanding quantities around infinite width, we expand them around zero training time. Surprisingly, our expansions match very well with numerical experiments even for very large training time: see Fig. 3.

**Finite-width corrections to NTK.** The work of Dyer & Gur-Ari (2019) considers so-called *correlation functions*, which are expectations of products of derivative tensors of a given scalar function $f$. NNGP, NTK, as well as higher-order kernels and kernel time-derivatives could be expressed as correlation functions. The main result is a simple upper-bound for an exponent of a correlation function as a function of width $n$.

This bound is computed by counting the numbers of even-sized and odd-sized components in the corresponding *cluster graph*. Vertices of this graph correspond to derivative tensors, while edges mark that the two tensors are tied with derivatives. While the definition of correlation functions is agnostic to the structure of $f$, the exponent upper-bound crucially realies on the structure of $f$ being an NTK-parameterized fully-connected network.

The bound is proven in some scenarios (e.g. linear nets), but empirically holds for all scenarios considered (including Tanh and ReLU nets). Feynman diagrams *are not* cluster graphs; they are used merely as a tool for proving the bound for linear nets. That is, for correlation functions not involving any derivatives (that is, $\mathbb{E}[f(x_1), \ldots, f(x_m)]$), the corresponding Feynman diagrams are graphs with vertices corresponding to inputs $x_1, \ldots, x_m$, and edges corresponding to paired weights of the same layer. The pairing is given by Wick's formula. That is, Feynman diagrams encode pairings involved in the Wick's formula. Each vertex has $L$ incident edges, where $L$ is the number of layers. When the correlation function involves derivatives, the two $f$'s contracted by the derivative act as a forced edge in the same type of diagrams.

In Dyer & Gur-Ari (2019), Feynman diagrams encode pairings in Wick's formula, while vertices encode different model inputs. In our work, these pairings are encoded by diagram contractions, while vertices encode common summation indices for adjacent weight matrices.

Higher-order derivatives we use in the loss expansion could be directly mapped to cluster graphs. However,

1. Initial weight variance $\sigma^2$ is a free parameter, dependence on which we study.

2. There are two non-equivalent notions of "width", which are $p$ and $H$.

3. We are interested not only in the leading term exponents, but also in exact constants in the leading terms, in their dependence on $s$ specifically. This is because we are looking for exact loss behavior for large $t$.

This is why the results of Dyer & Gur-Ari (2019) do not suffice in our case.

A follow up work of Aitken & Gur-Ari (2020) aims to prove the main exponent bound of Dyer & Gur-Ari (2019) for nets with polynomial activation functions. As before, the main theorem is formulated in terms of cluster graphs. However, Feynman diagrams are no longer used as a proof technique. Instead, they use "tree-like structures" (actually, forests), where vertices correspond to indices of weight matrices (that is, input and output weight matrices are mapped to a single vertex, while intermediary weight matrices are mapped to a pair of vertices), while edges mark the fact that adjacent weight matrices are tied with the corresponding summation index. Pairings defined by the Wick's formula are represented also as edges (as a pairing essentially ties the indices). These tree-like structure are more similar to diagrams in our work, but are still not the same thing.

Andreassen & Dyer (2020) is another follow up work. It aims to generalize the results of Dyer & Gur-Ari (2019) to networks with convolutions, global average poolings, and skip connections. Their main conjecture is very similar to that of Dyer & Gur-Ari (2019) but slightly more general: they consider *mixed correlation functions* for which derivative tensors are taken for different functions sharing weights (this is needed to model convolutions). The proof technique alsor relies on Feynman diagrams as in Dyer & Gur-Ari (2019).

### A.4 PARAMETER EXPANSIONS IN DEEP LEARNING

Recall that we expand the loss function as a function of time $t$: $L(t) = \sum_{k=0}^{\infty} \frac{d^k L(0)}{dt^k} \frac{t^k}{k!}$, see Section 2. A similar expansion appeared earlier in Dyer & Gur-Ari (2019) for the NTK: $\Theta(t) = \sum_{k=0}^{\infty} \frac{d^k \Theta(0)}{dt^k} \frac{t^k}{k!}$. The subsequent derivatives in that work were computed with a recursive formula. A parallel work of Huang & Yau (2020) expresses this recurrence as an infinite system of ODEs, where the time derivative of $\Theta$ is expressed using a higher-order kernel, whose derivative in turn is expressed with an even higher-order one.

### A.5 CANONICAL POLYADIC DECOMPOSITION

The loss function we consider in the present work is nothing more than a deviation of the identity tensor from its rank-$H$ approximation. Whenever the best approximation exists (i.e., when there exist weights that deliver a minimum to the loss), it is called *a canonical polyadic (CP) decomposition*, or, *PRAFAC*.

For a generic target tensor of order $\nu \geq 3$, computing its CP-decomposition with $H \geq 2$ is known to be NP-hard (Lim & Hillar, 2009), contrary to matrices ($\nu = 2$), where it could be derived from their SVD. Moreover, in some non-degenerate cases, the loss function does not even admit a minimum; recall a famous example of De Silva & Lim (2008):

$$A = u \otimes u \otimes v + u \otimes v \otimes u + v \otimes u \otimes u, \tag{29}$$

for unit non-identical vectors $u$ and $v$. Here, $A$ is a tensor of order $\nu = 3$ and rank 3; then $\lim_{n \to \infty} \|A_n - A\|_F = 0$ for

$$A_n = n \left( u + \frac{1}{n} v \right) \otimes \left( u + \frac{1}{n} v \right) \otimes \left( u + \frac{1}{n} v \right) - n u \otimes u \otimes u. \tag{30}$$

Here, all $A_n$ are of rank 2. In this case, looking for a rank-2 CP-decomposition of $A$ leads to a diverging sequence of approximations. De Silva & Lim (2008) proved that the set of order-3 tensors not admitting a best rank-2 approximation is strictly positive.

Nevertheless, in our study, we do not care about finding a best low-rank approximation of the target tensor at the first place, but rather describe the evolution of the loss function under the gradient flow dynamics. As we can see, the loss does not exhibit any divergences, even though the parameters might (as in the example of the above paragraph).

## B CONNECTION BETWEEN MODULAR ADDITION AND $\nu = 3$ PROBLEM

In this section we discuss a direct connection between our general framework and the modular addition task. This connection can be established in the case $\nu = 3$, where the tensor factorization problem can be interpreted as learning modular addition with a simple neural architecture.

### B.1 MODULAR ADDITION SETUP

Consider one-hot inputs $\mathbf{x}, \mathbf{y} \in \mathbb{R}^p$, and define a one-hidden-layer bilinear network, where the activation is given by multiplication:

$$f_l(\mathbf{x}, \mathbf{y}) = \sum_{k=1}^{H} w_{kl} \left( \sum_{i=1}^{p} u_{ki} x_i \right) \left( \sum_{j=1}^{p} v_{kj} y_j \right), \qquad l = 1, \ldots, p, \tag{31}$$

where $\{u_{ki}\}, \{v_{kj}\}$ are the input weights and $\{w_{kl}\}$ are the output weights. The target function is the modular addition map

$$f_{*,l}(\mathbf{x}, \mathbf{y}) = (\mathbf{x} * \mathbf{y})_l = \sum_{s=1}^{p} x_s y_{l-s}, \qquad l \in \mathbb{Z}_p, \tag{32}$$

and we train with the quadratic loss

$$L = \frac{1}{2} \mathbb{E}_{\mathbf{x}, \mathbf{y}} \sum_{l=1}^{p} \left( f_l(\mathbf{x}, \mathbf{y}) - f_{*,l}(\mathbf{x}, \mathbf{y}) \right)^2, \tag{33}$$

where the expectation is taken over the training dataset.

### B.2 FOURIER ANALYSIS

The map equation 32 is a cyclic convolution in $l$ and therefore diagonalizes under the unitary discrete Fourier transform (DFT) over $\mathbb{Z}_p$. We use the unitary DFT matrix $F \in \mathbb{C}^{p \times p}$ with entries

$$F_{rs} = p^{-1/2} e^{-2\pi i rs/p}, \qquad r, s \in \{1, \ldots, p\}, \tag{34}$$

so that $F^* F = F F^* = I$ (with $F^*$ the Hermitian conjugate). For any vector $\mathbf{a} = (a_1, \ldots, a_p)^\top \in \mathbb{C}^p$ we write

$$\hat{\mathbf{a}} = F \mathbf{a}, \qquad \check{\mathbf{a}} = F^* \mathbf{a}, \tag{35}$$

i.e., componentwise,

$$\hat{a}_r = p^{-1/2} \sum_{s=1}^{p} e^{-2\pi i rs/p} a_s, \qquad \check{a}_s = p^{-1/2} \sum_{r=1}^{p} e^{2\pi i rs/p} a_r. \tag{36}$$

We use $\hat{\mathbf{x}} = F\mathbf{x}, \hat{\mathbf{y}} = F\mathbf{y}, \hat{\mathbf{f}} = F\mathbf{f}$ for the Fourier transforms of inputs/outputs.

When taking the DFT along the *output* index $l$ (with hidden index $k$ fixed), we apply equation 36 to the vector $(w_{k1}, \ldots, w_{kp})^\top$:

$$\hat{w}_{kl} = p^{-1/2} \sum_{t=1}^{p} e^{-2\pi i lt/p} w_{kt}, \qquad l = 1, \ldots, p, \tag{37}$$

with inverse

$$w_{kt} = p^{-1/2} \sum_{l=1}^{p} e^{2\pi i lt/p} \hat{w}_{kl}, \qquad t = 1, \ldots, p. \tag{38}$$

Similarly, to rewrite the linear forms $\sum_{i=1}^{p} u_{ki} x_i$ and $\sum_{j=1}^{p} v_{kj} y_j$ in the Fourier basis, we apply the *inverse* DFT $F^*$ along the corresponding input indices: for each $k$, define

$$\check{u}_{kr} := p^{-1/2} \sum_{i=1}^{p} e^{2\pi i ri/p} u_{ki}, \qquad \check{v}_{kq} := p^{-1/2} \sum_{j=1}^{p} e^{2\pi i qj/p} v_{kj}, \tag{39}$$

so that, using $F^* F = I$,

$$\sum_{i=1}^{p} u_{ki} x_i = \sum_{r=1}^{p} \check{u}_{kr} \hat{x}_r, \qquad \sum_{j=1}^{p} v_{kj} y_j = \sum_{q=1}^{p} \check{v}_{kq} \hat{y}_q. \tag{40}$$

**Diagonalization of the target.** Applying the forward DFT (in $l$) to equation 32 and using changes of variables and sum reordering, we get

$$\hat{f}_{*,l}(\hat{\mathbf{x}}, \hat{\mathbf{y}}) = p^{-1/2} \sum_{t=1}^{p} e^{-2\pi i l t/p} f_{*,t}(\mathbf{x}, \mathbf{y}) = p^{-1/2} \sum_{t=1}^{p} e^{-2\pi i l t/p} \sum_{s=1}^{p} x_s \, y_{t-s} \tag{41}$$

$$= p^{-1/2} \sum_{s=1}^{p} x_s \sum_{t=1}^{p} e^{-2\pi i l t/p} \, y_{t-s} \qquad \text{(swap the order of summation)} \tag{42}$$

$$= p^{-1/2} \sum_{s=1}^{p} x_s \sum_{u=1}^{p} e^{-2\pi i l (u+s)/p} \, y_u \qquad \text{(let } u = t - s; \text{ indices are modulo } p) \tag{43}$$

$$= p^{-1/2} \left( \sum_{s=1}^{p} x_s \, e^{-2\pi i l s/p} \right) \left( \sum_{u=1}^{p} e^{-2\pi i l u/p} \, y_u \right) \qquad \text{(factor } e^{-2\pi i l s/p}) \tag{44}$$

$$= p^{-1/2} \left( \sqrt{p} \, \hat{x}_l \right) \left( \sqrt{p} \, \hat{y}_l \right) = p^{1/2} \, \hat{x}_l \, \hat{y}_l. \tag{45}$$

Thus the modular addition target is diagonal in the Fourier basis:

$$\hat{f}_{*,l}(\hat{\mathbf{x}}, \hat{\mathbf{y}}) = \sqrt{p} \, \hat{x}_l \, \hat{y}_l.$$

**Network in the Fourier basis.** By definition of the DFT of the output,

$$\hat{f}_l(\hat{\mathbf{x}}, \hat{\mathbf{y}}) = p^{-1/2} \sum_{t=1}^{p} e^{-2\pi i l t/p} f_t(\mathbf{x}, \mathbf{y}).$$

Substituting equation 31 and interchanging sums yields

$$\hat{f}_l(\hat{\mathbf{x}}, \hat{\mathbf{y}}) = \sum_{k=1}^{H} \left( p^{-1/2} \sum_{t=1}^{p} e^{-2\pi i l t/p} \, w_{kt} \right) \left( \sum_{i=1}^{p} u_{ki} x_i \right) \left( \sum_{j=1}^{p} v_{kj} y_j \right)$$

$$= \sum_{k=1}^{H} \hat{w}_{kl} \underbrace{\left( \sum_{i=1}^{p} u_{ki} x_i \right)}_{=\sum_{r=1}^{p} \check{u}_{kr} \hat{x}_r} \underbrace{\left( \sum_{j=1}^{p} v_{kj} y_j \right)}_{=\sum_{q=1}^{p} \check{v}_{kq} \hat{y}_q} \tag{46}$$

$$= \sum_{k=1}^{H} \hat{w}_{kl} \left( \sum_{r=1}^{p} \check{u}_{kr} \, \hat{x}_r \right) \left( \sum_{q=1}^{p} \check{v}_{kq} \, \hat{y}_q \right), \tag{47}$$

where $\sum_i u_{ki} x_i = \sum_r \check{u}_{kr} \hat{x}_r$ follows from $u_k^\top x = (F^* u_k)^\top (Fx)$ (and similarly for $v_k$).

**Averaging over the full dataset.** Because $F$ is unitary, Parseval's identity lets us evaluate the loss in the Fourier domain:

$$L = \frac{1}{2} \, \mathbb{E}_{\mathbf{x}, \mathbf{y}} \sum_{l=1}^{p} \left| \hat{f}_l(\hat{\mathbf{x}}, \hat{\mathbf{y}}) - \hat{f}_{*,l}(\hat{\mathbf{x}}, \hat{\mathbf{y}}) \right|^2 \tag{48}$$

$$= \frac{1}{2} \, \mathbb{E}_{\mathbf{x}, \mathbf{y}} \sum_{l=1}^{p} \left| \sum_{k=1}^{H} \sum_{r=1}^{p} \sum_{q=1}^{p} \hat{w}_{kl} \, \check{u}_{kr} \, \check{v}_{kq} \, \hat{x}_r \, \hat{y}_q \; - \; \sqrt{p} \, \hat{x}_l \, \hat{y}_l \right|^2, \tag{49}$$

where we used equation 47 and equation 45.

**Computing the expectations inside the sum.** Fix $l \in \{1, \ldots, p\}$ and set

$$A_{rq}^{(l)} := \sum_{k=1}^{H} \hat{w}_{kl} \, \check{u}_{kr} \, \check{v}_{kq} \qquad (r, q = 1, \ldots, p).$$

Then equation 49 reads

$$\hat{f}_l - \hat{f}_{*,l} = \sum_{r,q} A_{rq}^{(l)} \, \hat{x}_r \, \hat{y}_q \; - \; \sqrt{p} \, \hat{x}_l \, \hat{y}_l.$$

Expand the squared modulus and take the dataset expectation:

$$\mathbb{E}_{\mathbf{x},\mathbf{y}}\left|\hat{f}_l - \hat{f}_{*,l}\right|^2 = \mathbb{E}_{\mathbf{x},\mathbf{y}}\left[\left(\sum_{r,q} A_{rq}^{(l)}\hat{x}_r\hat{y}_q - \sqrt{p}\,\hat{x}_l\hat{y}_l\right)\left(\sum_{r',q'}\overline{A_{r'q'}^{(l)}}\check{x}_{r'}\check{y}_{q'} - \sqrt{p}\,\check{x}_l\check{y}_l\right)\right]$$

$$= \underbrace{\sum_{r,q}\sum_{r',q'} A_{rq}^{(l)}\overline{A_{r'q'}^{(l)}}\,\mathbb{E}[\hat{x}_r\check{x}_{r'}]\,\mathbb{E}[\hat{y}_q\check{y}_{q'}]}_{\text{(I)}}$$

$$- \underbrace{\sqrt{p}\sum_{r,q} A_{rq}^{(l)}\,\mathbb{E}[\hat{x}_r\check{x}_l]\,\mathbb{E}[\hat{y}_q\check{y}_l]}_{\text{(II)}}$$

$$- \underbrace{\sqrt{p}\sum_{r',q'}\overline{A_{r'q'}^{(l)}}\,\mathbb{E}[\hat{x}_l\check{x}_{r'}]\,\mathbb{E}[\hat{y}_l\check{y}_{q'}]}_{\text{(III)}}$$

$$+ \underbrace{p\,\mathbb{E}[\hat{x}_l\check{x}_l]\,\mathbb{E}[\hat{y}_l\check{y}_l]}_{\text{(IV)}}. \tag{50}$$

Here we used independence of $\mathbf{x}$ and $\mathbf{y}$ to factor the expectations. Since the dataset is the full Cartesian product with a uniform measure and the Fourier modes are orthogonal:

$$\mathbb{E}_{\mathbf{x}}[\hat{x}_r\check{x}_{r'}] = \delta_{rr'}, \qquad \mathbb{E}_{\mathbf{y}}[\hat{y}_q\check{y}_{q'}] = \delta_{qq'}.$$

Therefore each line in equation 50 simplifies as follows:

$$\text{(I)} = \sum_{r,q}\sum_{r',q'} A_{rq}^{(l)}\overline{A_{r'q'}^{(l)}}\,\delta_{rr'}\delta_{qq'} = \sum_{r,q}\left|A_{rq}^{(l)}\right|^2,$$

$$\text{(II)} = \sqrt{p}\sum_{r,q} A_{rq}^{(l)}\,\delta_{rl}\,\delta_{ql} = \sqrt{p}\,A_{ll}^{(l)},$$

$$\text{(III)} = \sqrt{p}\sum_{r',q'}\overline{A_{r'q'}^{(l)}}\,\delta_{lr'}\,\delta_{lq'} = \sqrt{p}\,\overline{A_{ll}^{(l)}},$$

$$\text{(IV)} = p\cdot 1\cdot 1 = p.$$

Putting the pieces together gives, for each fixed $l$,

$$\mathbb{E}_{\mathbf{x},\mathbf{y}}\left|\hat{f}_l - \hat{f}_{*,l}\right|^2 = \sum_{r,q}\left|A_{rq}^{(l)}\right|^2 - 2\sqrt{p}\,\Re(A_{ll}^{(l)}) + p.$$

Finally, use the elementary identity (valid for any $A_{rq}$)

$$\sum_{r,q}\left|A_{rq} - \sqrt{p}\,\delta_{r=q=l}\right|^2 = \sum_{r,q}|A_{rq}|^2 - 2\sqrt{p}\,\Re(A_{ll}) + p,$$

to rewrite the previous line as

$$\mathbb{E}_{\mathbf{x},\mathbf{y}}\left|\hat{f}_l - \hat{f}_{*,l}\right|^2 = \sum_{r,q}\left|A_{rq}^{(l)} - \sqrt{p}\,\delta_{r=q=l}\right|^2.$$

Summing over $l$ and restoring $A_{rq}^{(l)} = \sum_k \hat{w}_{kl}\check{u}_{kr}\check{v}_{kq}$ yields

$$L = \frac{1}{2}\sum_{l=1}^{p}\sum_{r=1}^{p}\sum_{q=1}^{p}\left|\sum_{k=1}^{H}\hat{w}_{kl}\check{u}_{kr}\check{v}_{kq} - \sqrt{p}\,\delta_{r=q=l}\right|^2 = \frac{1}{2}\sum_{i,j,l=1}^{p}\left|\sum_{k=1}^{H}\hat{w}_{kl}\check{u}_{ki}\check{v}_{kj} - \sqrt{p}\,\delta_{i=j=l}\right|^2. \tag{51}$$

**Problem simplification.** The following two simplifications let us connect the modular-addition objective to our general CP setup and keep the notation lightweight.

1. **Absorbing the $\sqrt{p}$ factor (w.l.o.g.).** In equation 51 the target carries a factor $\sqrt{p}$. We remove this by the readout reparameterization

$$\widetilde{w}_{kl} \;:=\; p^{-1/2}\,\hat{w}_{kl},$$

which turns $\sqrt{p}\,\delta_{i=j=l}$ into $\delta_{i=j=l}$. This is a one-to-one change of variables on the readout that leaves the optimization landscape unchanged. For brevity we drop the tilde and continue to write $w$.

2. **Passing to real-valued weights (surrogate).** The diagonal expression in equation 51 is written in complex Fourier coordinates (with conjugate symmetry when spatial-domain weights are real). One could keep the complex formulation or split into real and imaginary parts, but this introduces additional couplings and heavier notation. To streamline the analysis and align with our general framework, we adopt the following *real*, diagonal surrogate. This surrogate is not mathematically identical to the complex formulation; we proceed under the working assumption that it preserves the key features of the corresponding gradient-flow dynamics.

With these conventions, we get the real-valued tensor factorization objective

$$L_{\text{real}} = \frac{1}{2} \sum_{i,j,l=1}^{p} \left| \sum_{k=1}^{H} u_{ki} v_{kj} w_{kl} - \delta_{i=j=l} \right|^2. \tag{52}$$

This is exactly the instance of our general setup equation 2 with $\nu = 3$ (asymmetric case); imposing the constraint equation 3 gives the symmetric version.

### B.3 CONNECTION TO ANALYTIC SOLUTIONS

Gromov (2023) exhibit an exact solution to modular addition using a one-hidden-layer network with Fourier-aligned weights. Their construction identifies a family of global minimizers whose features are periodic and whose frequencies match the group structure of $\mathbb{Z}_p$. Importantly, that work does *not* analyze training dynamics (gradient descent or gradient flow); convergence to the constructed minimizers is supported empirically rather than proved.

Our viewpoint is complementary. In the complex Fourier basis, modular addition is diagonal, and the same structure is captured in our framework by the diagonal identity tensor $\delta_{i=j=l}$ for $\nu = 3$. Thus, Gromov's analytic Fourier solution and our CP formulation describe the same underlying modular structure in different coordinates. Beyond existence, however, our goal in the present work is dynamical: we aim to derive explicit, time-dependent formulas for the loss $L(t)$ under the gradient flow.

### B.4 RELATED WORKS

The primary motivation for the problem of identity tensor decomposition we study in the present paper is the problem of training a two-layer fully-connected network with a quadratic activation function on modular addition. We review various instances of modular arithmetic problems in Deep Learning.

**Modular arithmetic as a case study for grokking.** To the best of our knowledge, the problem of modular arithmetic first appeared in Power et al. (2022). In this work, the authors noticed that a shallow Transformer (Vaswani et al., 2017) trained to solve various modular arithmetic problems exhibits a remarkable phenomenon dubbed *grokking*: the model achieves perfect generalization much after reaching the perfect train accuracy. By the time the model achieves a perfect train accuracy, the test one usually stays at the random guess level.

Attempts to provide a theoretical explanation for this phenomenon in the original setting (i.e., Transformers trained on modular arithmetic) include *slingshot mechanism* (Thilak et al., 2022) and *circuit formation* (Nanda et al., 2023). This phenomenon has later been observed also for MLPs (Morwani

et al., 2023; Mohamadi et al., 2024) and for non-neural models trained not with gradient methods (Mallinar et al., 2024). Grokking has also been observed for various problems that differ from modular arithmetic, including (1) group operations (Chughtai et al., 2023), (2) sparse parity (Barak et al., 2022; Bhattamishra et al., 2022), (3) greatest common divisor (Charton, 2023), (4) image classification (Liu et al., 2022; Radhakrishnan et al., 2022), as well as generic supervised learning problems under specific conditions (Lyu et al., 2023).

**Mechanistic interpretation of learned algorithms for models trained on modular arithmetic.** A problem of modular addition could be naturally solved by first, Fourier-transforming one-hot representations of the summands, convolving them next, and finally, transforming them back to the original space. Applying methods of mechanistic interpretability, Nanda et al. (2023) demonstrated that a trained Transformer indeed converges to this kind of algorithm. Their conclusion was later refined by Zhong et al. (2023) who demonstrated that the claimed algorithm (dubbed *Clock*) is not unique a trained Transformer could converge to, but there is another typical one, dubbed *Pizza*, as well as a class of algorithms that mixes between these two. As a sequel, Furuta et al. (2024) extended their analysis to other modular arithmetic problems.

In the meantime, Gromov (2023) demonstrated that a two-layer MLP with a quadratic activation function could implement the above Fourier-feature algorithm in the limit of infinite width by explicitly providing the corresponding weights. This architecture is closely related to the model we study in the present work, as we explain Appendix B. A subsequent work of Doshi et al. (2024) extends their explicit construction to modular multiplication and modular addition with multiple terms. Finally, a recent work of McCracken et al. (2025) building on Nanda et al. (2023) and Zhong et al. (2023) claims that both MLPs and Transformers trained on modular addition implement an abstract algorithm they dub *approximate Chinese Remainder Problem* implicitly providing the corresponding weights.

## C  HYPERPARAMETER POLYGON

We interpret some of the extremal simplices of the hyperparameter polygon, Fig. 2, below. Our conclusions are summarized in Table 2. We compute the natural scaling for the inverse learning rate $T$ by balancing the base of power $s$ in the leading Pareto-term provided by Theorem 1.

### C.1  SYMMETRIC SCENARIO

In this scenario, as Theorem 1 states, the hyperparameter polygon is a triangle: see Fig. 2, left. Let us denote its summits as A, B, and E, as depicted on the figure.

**Edge A-B.** Here we have $s_D = s + 1$, hence $s_R = 0$. This means that the target is never used for computing the loss derivatives. We call this regime *free evolution*. It becomes dominant when $p^\nu \asymp H^2$, while $p^{\nu-1}\sigma^\nu \to \infty$. That is, when the initialization is huge, the identity target is essentially zero compared to the initial model, and the model attempts to learn zero throughout most of its training time. The natural scaling for $T$ is $T \asymp p^{\nu-1}\sigma^{2\nu-2}$.

The whole edge stays dominant when the model is balanced: $H \asymp p^{\nu/2}$. When this condition gets violated, either Point A (underparameterization), or Point B (overparameterization) becomes solely dominant.

**Edge B-E.** Here we have $n = s_D + 1$, hence $Q(n, s_D) = 1 + \left(\frac{\nu}{2} - 1\right) s_D$. It becomes dominant when $p^{\nu-2}H^2\sigma^{2\nu} \asymp 1$, while $H^2/p^\nu \to \infty$. That is, $H$ has to be large compared to some power of $p$, hence this regime is overparameterized. The natural scaling for $T$ is $T \asymp \sigma^{\nu-2}$.

Recall that $\nu = 2$ corresponds to matrix factorization, and the model is a conventional two-layer fully-connected network. In this case, the edge B-E becomes dominant when $\sigma^2 \asymp 1/H$, while $H/p \to \infty$. This is nothing but a hyperparameter scaling that yields a mean-field limit discovered independently in a number of works (see Appendix A.1). For two-layer nets, the training dynamics could be expressed as an evolution of measure. Under the above scaling, the moments of this measure stay finite in the limit.

| Simplex | Scaling condition | Natural $T$ | Parame-terization | Learning | Interpretation |
|---------|-------------------|-------------|-------------------|----------|----------------|
| Symmetric scenario, even $\nu$ | | | | | |
| A – B | $H^2 \asymp p^\nu$, $p^{\nu-1}\sigma^\nu \to \infty$ | $p^{\nu-1}\sigma^{2\nu-2}$ | Balanced | No | Free evolution |
| B – E | $p^{\nu-2}H^2\sigma^{2\nu} \asymp 1$, $H^2/p^\nu \to \infty$ | $\sigma^{\nu-2}$ | Over- | Rich | Mean-field |
| E | $p^{\nu-2}H^2\sigma^{2\nu} \to 0$, $p^{\nu-1}\sigma^\nu \to 0$ | $\sigma^{\nu-2}$ | Agnostic | Rich | — |
| E – A | $p^{\nu-1}\sigma^\nu \asymp 1$, $H^2/p^\nu \to 0$ | $\sigma^{\nu-2}$ | Under- | Rich | — |
| Asymmetric scenario | | | | | |
| A – B | $H \asymp p^{\nu-1}$, $p^{\nu-1}\sigma^\nu \to \infty$ | $H\sigma^{2\nu-2}$ | Balanced | No | Free evolution |
| B – C | $p^{\nu-1}H\sigma^{2\nu} \asymp 1$, $H\sigma^\nu \to \infty$ | $H\sigma^{2\nu-2}$ | Over- | Lazy | NTK |
| C | $p^{\nu-1}H\sigma^{2\nu} \to 0$, $H\sigma^\nu \to \infty$ | $H\sigma^{2\nu-2}$ | Over- | Lazy | NTK, $f(0) \equiv 0$ |
| C – D | $H\sigma^\nu \asymp 1$, $H/p^{\nu-1} \to \infty$ | $\sigma^{\nu-2}$ | Over- | Rich | Mean-field |
| D – E | $H \asymp p^{\nu-1}$, $H\sigma^\nu \to 0$ | $\sigma^{\nu-2}$ | Balanced | Rich | — |
| E – A | $p^{\nu-1}\sigma^\nu \asymp 1$, $H/p^{\nu-1} \to 0$ | $\sigma^{\nu-2}$ | Under- | Rich | — |

Table 2: Some extremal simplices along with their interpretations.

**Edge E-A.** Here we have $n = 1$, hence $Q(n, s_D) = 1 + (\nu - 1)s_D$. It becomes dominant when $p^{\nu-1}\sigma^\nu \asymp 1$, while $H^2/p^\nu \to 0$. The natural scaling for $T$ is $T \asymp \sigma^{\nu-2}$.

For $\nu = 2$, the edge E-A becomes dominant when $\sigma^2 \asymp 1/p$, while $p/H \to \infty$. That is, the required scaling coincides with the mean-field one with $p$ and $H$ swapped.

**Point E.** This summit deserves a special consideration. Recall that for Edge A-B the initialization variance $\sigma^2$ has to be large, while for the other two edges, it has to balance the growth of $p$ and $H$ in a specific way. For Point E to become dominant, this variance has to be small: $p^{\nu-2}H^2\sigma^{2\nu} \to 0$ along with $p^{\nu-1}\sigma^\nu \to 0$. For $\nu = 2$, this yields a saddle-to-saddle regime extensively studied for linear networks: see Appendix A.2. However, in our case, when the target is identity, the only saddle the training process encounters is the one at the origin. This makes the saddle-to-saddle regime trivial. Same as for Edges B-E and E-A, the natural scaling for $T$ is $T \asymp \sigma^{\nu-2}$.

## C.2 ASYMMETRIC SCENARIO

As Theorem 1 states, one or two extremal points are missing compared to the triangle of the symmetric scenario. This gives a quadrangle for odd $s$, or a pentagon for even $s$, hence more edge-cases: see Fig. 2, right. In particular, the point C which appears as an extremal point due to exclusion of Point B from the triangle, yields a linearized training regime which was absent in the symmetric scenario; see below.

**Edge A-B.** This is the same free evolution as before, but the dominance condition changes: $p^{\nu-1} \asymp H$, while $p^{\nu-1}\sigma^\nu \to \infty$ and the natural scaling for $T$ is $T \asymp p^{\nu-1}\sigma^{2\nu-2}$ as before, or, equivalently, $T \asymp H\sigma^{2\nu-2}$. Note that this condition coincides with that of the symmetic case when $\nu = 2$ (for matrix factorization).

**Point C.** This point corresponds to $(n, s_D) = (s, s-1)$. In this case, $s_R = 2$, hence in contrast to free evolution, the target does appear in loss derivatives but only weakly. This corresponds to a linearized learning regime akin to NTK, see Appendix A.1, but starting from a zero model.

The natural scaling for $T$ is $T \asymp H\sigma^{2\nu-2}$. Point C becomes dominant when $p^{\nu-1}H\sigma^{2\nu} \to 0$, while $H\sigma^\nu \to \infty$. In particular, this implies that $H/p^{\nu-1} \to \infty$, hence this regime is overparameterized.

**Edge B-C.** This edge always contains only two Pareto-optimal points: B and C. Point B adds terms with $s_R = 0$; because of this, the corresponding regime is still linearized but the initial model is no longer zero.

Edge B-C becomes dominant when $p^{\nu-1}H\sigma^{2\nu} \asymp 1$, while $H\sigma^\nu \to \infty$. We again need $H/p^{\nu-1} \to \infty$, hence this regime is still overparameterized. The natural scaling for $T$ is again $T \asymp H\sigma^{2\nu-2}$. When $\nu = 2$, for this edge to become dominant one needs $\sigma^2 \asymp 1/\sqrt{pH}$. This is nothing but a conventional NTK initialization scaling: see Appendix A.1.

We note that the original NTK paper of Jacot et al. (2018) applies a constant learning rate, while we claim $T \asymp H\sigma^2$ to be natural for $\nu = 2$. There is no contradiction. The reason of this discrepancy lies in the fact that Jacot et al. (2018) initialize neural network's weights from $\mathcal{N}(0, 1)$, while putting $\sigma$ as a pre-factor in the form $\phi(\sigma \times W)$ for $\phi$ being an activation function and $W$ being a weight matrix. In contrast, we initialize our weights from $\mathcal{N}(0, \sigma^2)$ with no pre-factors. The GF dynamics in these two cases become equivalent if we rescale the learning rate appropriately; this gives us the claimed scaling for $T$.

**Edge C-D.** Here we have $n = s_D + 1$, hence $Q(n, s_D) = 1$. It becomes dominant when $H\sigma^\nu \asymp 1$, while $p^{\nu-1}\sigma^\nu \to 0$. In particular, this implies that $H/p^{\nu-1} \to \infty$, hence this regime is overparameterized. Akin to the symmetric scenario, we recover a conventional mean-field scaling when $\nu = 2$. The natural scaling for $T$ is $T \asymp \sigma^{\nu-2}$.

**Edge D-E.** Here we have $s_D = 1$ for even $s$ and $s_D = 0$ for odd $s$. Therefore $s_R$ is maximal, and only the highest-order target correlations survive in loss derivatives.

The natural scaling for $T$ is again $T \asymp \sigma^{\nu-2}$. For this edge to become dominant, one needs $H\sigma^\nu \to 0$, while $H \asymp p^{\nu-1}$. In words, it is a critically overparameterized regime with small initialization. Same as for Point E under the symmetric scenario, this edge yields a trivial (single saddle) case of a well-studied saddle-to-saddle regime for linear nets: see Appendix A.2.

**Edge E-A.** Similar to the symmetic scenario, we have $n = 1$, hence $Q(n, s_D) = 1 + (\nu - 1)s_D$. This edge becomes dominant when $p^{\nu-1}\sigma^\nu \asymp 1$, while $H/p^{\nu-1} \to 0$. That is, here we have a critically initialized insufficiently overparameterized regime. Akin to the symmetric scenario, the dominance condition gives that of the mean-field regime (Edge C-D) after swapping $H$ and $p$ when $\nu = 2$. The natural scaling for $T$ is again $T \asymp \sigma^{\nu-2}$.

## D  NON-EXISTENCE OF LINEARIZED TRAINING REGIMES FOR THE SYMMETRIC SCENARIO

From the limit polygons we derived, we see that there is a linearized training regime (akin to NTK) for the identity matrix factorization in the asymmetric scenario, while there are no such regime in the symmetric one. We are going to demonstrate that in the former case, the NTK is not constant unless the model does not learn, while it is constant in the latter case.

Consider first the symmetric case. We could pose our loss minimization problem as a supervised learning problem with inputs $(x, y)$, targets $x^\top y$, and a model to learn $f(x, y) = x^\top U^\top U y$, where $x, y \in \mathbb{R}^d$. The training dataset consists of all (and only) one-hot inputs.

A model weight derivative is

$$\partial_U f(x, y) = Uxy^\top + Uyx^\top, \tag{53}$$

hence the NTK is given by

$$\begin{aligned}
\Theta(x, y, x', y') &= \mathrm{Tr}\left[\partial_U^\top f(x, y)\partial_U f(x', y')\right] \\
&= \langle y, y'\rangle f(x, x') + \langle x, x'\rangle f(y, y') + \langle x', y\rangle f(x, y') + \langle x, y'\rangle f(x', y).
\end{aligned} \tag{54}$$

As we see, there is a direct connection between the NTK and the model itself. Because of this, the NTK evolution is directly related to the model evolution. As we shall see shortly below, there is no such connection when the weights are not shared.

We are interested only in one-hot inputs. Let $x$ be a one-hot vector for $n$, while $y$ is a one-hot vector for $m$; similarly for $x'$ and $y'$. Then by slightly abusing notation,

$$\Theta(n, m, n', m') = \mathbf{1}_{m=m'} f(n, n') + \mathbf{1}_{n=n'} f(m, m') + \mathbf{1}_{m=n'} f(n, m') + \mathbf{1}_{n=m'} f(n', m). \tag{55}$$

**Proposition 1.** *For the above scenario, $d_t\Theta(n, m, n', m') = 0 \,\forall n, m, n', m' \in [0 : p - 1]$ if and only if $d_t f(n, m) = 0 \,\forall n, m \in [0 : p - 1]$.*

*Proof.* The "if" part is trivial. Let us prove the "only if" part. Suppose the kernel does not vary: $d_t\Theta(n, m, n', m') = 0 \,\forall n, m, n', m'$. Consider any $n, m, m'$ with no two of them being equal. Then $\Theta(n, m, m, m') = f(n, m')$. Therefore $d_t f(n, m') = 0 \,\forall n \neq m'$. Consider then any $n$. In this case, $\Theta(n, n, n, n) = 4f(n, n)$. Therefore $d_t f(n, n) = 0 \,\forall n$. $\square$

**Why there is a linearized regime in the asymmetric scenario.** On the other hand, when weight matrices are different, we get the following. The associated model becomes $f(x, y) = x^\top U^\top V y$. Its weight derivatives are given by

$$\partial_U f(x, y) = Vyx^\top, \qquad \partial_V f(x, y) = Uxy^\top, \tag{56}$$

while the NTK is given by

$$\begin{aligned}
\Theta(x, y, x', y') &= \mathrm{Tr}\left[\partial_U^\top f(x, y)\partial_U f(x', y') + \partial_V^\top f(x, y)\partial_V f(x', y')\right] \\
&= \langle x, x'\rangle y^\top V^\top V y' + \langle y, y'\rangle x^\top U^\top U x'.
\end{aligned} \tag{57}$$

As we see from here, in the symmetric case, the NTK is not directly connected to the model $f$. In order to study its evolution, we need the evolution of weights:

$$Td_t U = V \left(I - V^\top U\right), \qquad Td_t V = U \left(I - U^\top V\right).$$ (58)

Therefore

$$\begin{aligned}
Td_t \Theta(x, y, x', y') = \langle x, x'\rangle \, y^\top \left[V^\top U \left(I - U^\top V\right) + \left(I - V^\top U\right) U^\top V\right] y' \\
+ \langle y, y'\rangle \, x^\top \left[\left(I - U^\top V\right) V^\top U + U^\top V \left(I - V^\top U\right)\right] x'.
\end{aligned}$$ (59)

Turning to one-hot vectors as before, we get

$$\begin{aligned}
Td_t \Theta(n, m, n', m') = \mathbf{1}_{n=n'} \left[f(m', m) + f(m, m') - 2\sum_{k=0}^{p-1} f(k, m) f(k, m')\right] \\
+ \mathbf{1}_{m=m'} \left[f(n', n) + f(n, n') - 2\sum_{k=0}^{p-1} f(n, k) f(n', k)\right].
\end{aligned}$$ (60)

As we demonstrated in Appendix C, in order to have a nondegenerate loss evolution with training time $t$ at Point C or Edge B-C, we need $T \asymp H\sigma^2$, while we need $pH\sigma^4 = \Omega(1)$ and $H/p \to \infty$ as $H, p, 1/\sigma \to \infty$ for these simplices to become dominant. Because of this,

$$T \asymp H\sigma^2 = \Omega\left(\sqrt{\frac{H}{p}}\right) \to \infty.$$ (61)

Therefore as long as the model stays finite, $d_t \Theta(n, m, n', m')$ vanishes in this limit.

# E  PROOF OF THEOREM 1

## E.1  SYMMETRIC MODELS, EVEN $\nu$

### E.1.1  REALIZABILITY

We need to prove that for each term

$$p^{1+(\nu-1)s_D - \frac{\nu}{2}(n-1)} H^n \sigma^{\nu(s_D+1)+(\nu-2)s}$$ (62)

with some $0 \le s_D \le s+1$ and $1 \le n \le s_D + 1$, there exists a diagram $G$ obtained by merging, in some order, $s_D$ diagrams $D_{2\nu}$ and $s_R = s + 1 - s_D$ diagrams $R_\nu$, such that some contraction $\widehat{G}$ of $G$ has $n$ $H$-nodes, $1 + (\nu-1)s_D - \frac{\nu}{2}(n-1)$ $p$-nodes, and admits an edge pairing. If true, such a diagram contributes the respective term (62) to $\mathbb{E}[(\frac{1}{2}D_{2\nu} - R_\nu)^{\star(s+1)}]$ with the coefficient $\frac{(-1)^{s_R}}{2^{s_D}}$ multiplied by the number of admitted pairings.

Note that there can be no cancellations of contributions of different diagrams: the triplet of powers

$$\begin{pmatrix} q \\ n \\ 2l \end{pmatrix} = \begin{pmatrix} 1 + (\nu-1)s_D - \frac{\nu}{2}(n-1) \\ n \\ \nu(s_D+1) + (\nu-2)s \end{pmatrix}$$ (63)

in (62) is uniquely determined by $s, s_D, n$, therefore all the diagrams with the same triplet of powers have the same coefficient $\frac{(-1)^{s_R}}{2^{s_D}}$. This shows that we only need to find one suitable diagram $G$ to guarantee the presence of the respective term $p^q H^n \sigma^{2l}$ in $\mathbb{E}[(\frac{1}{2}D_{2\nu} - R_\nu)^{\star(s+1)}]$.

We construct desired examples by induction on $s_D$. The base of induction is $s_D = 0$. The respective diagrams $G$ have the form $R_\nu^{\star(s+1)}$ and consist of one $p$-node and one $H$-node connected by $\nu + s(\nu - 2)$ edges. Since $\nu$ is even, these diagrams admit an edge-pairing and contribute a term with $(n, q) = (1, 1)$, consistent with Eq. (62).

Now we make an induction step. As the induction hypothesis, suppose that we have a diagram $G$ satisfying our conditions. We will show that merging $G \star D_{2\nu}$ produces another diagram $G'$ satisfying our conditions, with new numbers $n', q'$ of contracted $p$-nodes and $H$-nodes that also satisfy our conditions. We consider two options.

1. **[Retaining $n$]** We construct the new diagram $G'$ such that the new numbers $n', q'$ are related to the numbers $n, q$ for $G$ by

$$n' = n, \quad q' = q + \nu - 1. \tag{64}$$

To this end, we simply take any edge $u_{ki}$ in $G$ and merge $G$ with $D_{2\nu}$ over this edge. This adds a new $H$-node $k'$ and $\nu - 1$ new $p$-nodes $i_1, \ldots, i_{\nu-1}$ to $G$, and replaces the edge $u_{ki}$:

$$u_{ki} \rightsquigarrow u_{k'i} \prod_{m=1}^{\nu-1} u_{ki_m} u_{k'i_m}. \tag{65}$$

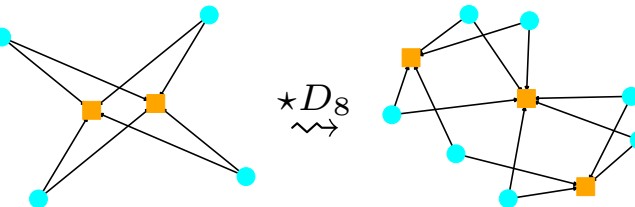

Now, we contract the resulting diagram $G'$ by retaining the contractions in $G$ and additionally contracting the node $k'$ to $k$.

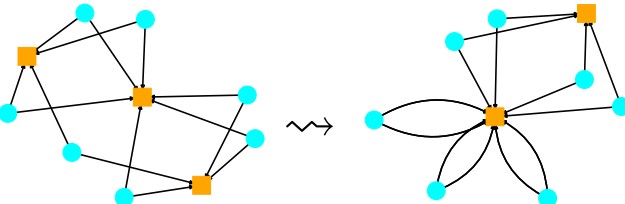

This makes the edges $u_{ki_m}, u_{k'i_m}$ naturally paired, and also allows to identify the new edge $u_{k'i}$ with the old edge $u_{ki}$. We then have an edge pairing in $\widehat{G}'$ obtained by retaining the pairing in $\widehat{G}$ and additionally supplementing it with the pairing of the new edges $u_{ki_m}, u_{k'i_m}$. This creates the desired diagram fulfilling condition (64).

2. **[Increasing $n$]** Alternatively, we can construct a new diagram such that

$$n' = n + 1, \quad q' = q + \frac{\nu}{2} - 1. \tag{66}$$

To this end, we define again $G'$ by merging $G$ with $D_{2\nu}$ over any edge. But now we don't contract the nodes $k$ and $k'$. Instead, we divide the nodes $i_1, \ldots, i_{\nu-1}$ and $i$ into pairs (which is possible since $\nu$ is even) and contract the nodes in each pair. This gives the relation (66).

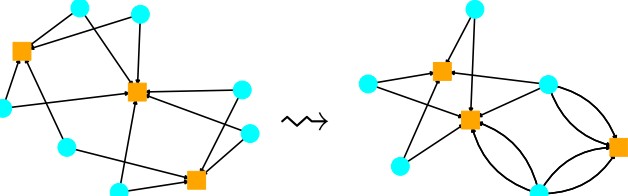

To construct an admissible edge pairing, we form pairs among the edges $(u_{k'i_m})_{m=1,\ldots,\nu-1}$ and $u_{k'i}$ using the introduced contraction. Similarly, we form pairs among the edges $(u_{ki_m})_{m=1,\ldots,\nu-1}$ except for $m_*$ such that $i_{m_*}$ is contracted with $i$ (this $u_{ki_{m_*}}$ is left unpaired because of the missing edge $u_{ki}$ in $G'$). We also retain the edge pairing from $G$, except for the edge $u_{k''u''}$ paired with $u_{ki}$ (again, because $u_{ki}$ is no longer present in $G'$). To complete the pairing in $G'$, we can pair $u_{ki_{m_*}}$ with $u_{k''i''}$, because the node $k''$ is contracted to $k$ while the node $i_{m_*}$ is contracted to $i$ and hence to $i''$.

Clearly, the above two transformations allow to generate all terms (62) (for given $s, s_D, n$ just start from a diagram $R_{2\nu}^{s+1-s_D}$, perform $n$-retaining transformation $s_D + 1 - n$ times and $n$-increasing transformation $n - 1$ times).

### E.1.2 OPTIMALITY

We need to show that each term listed in Eq. (10) is Pareto-optimal with respect to the factor $p^q H^n$, i.e. for given $s, s_D$ there are no diagrams in the binomial expansion $(\frac{1}{2}D_{2\nu} - R_\nu)^{\star(s+1)}$ with exactly $s_D$ factors $D_{2\nu}$ that would contribute a strictly Pareto-dominating factor $p^{q'} H^{n'}$.

Since merging with $D_{2\nu}$ creates one additional $H$-node while merging with $R_\nu$ creates no $H$-nodes, possible values of $n$ are necessarily restricted to $1, \ldots, s_D + 1$. Thus, the optimality condition that we need to prove can be stated as follows: for any contracted diagram with given $s, s_D$ and admitting a pairing, the number of contracted $H$-nodes $n$ and the number of contracted $p$-nodes $q$ satisfy the inequality

$$q \leq 1 + (\nu - 1)s_D - \frac{\nu}{2}(n - 1). \tag{67}$$

We prove this by induction on $s_D$ mimicking the induction used in the proof of realizability. The base of induction is $s_D = 0$, i.e. diagrams $R_\nu^{\star(s+1)}$. Such diagrams consist of one $p$-node and one $H$-node connected by $\nu + s(\nu - 2)$ edges. Since $\nu$ is even, such diagrams admit pairings and thus contrubute a term with $(n, q) = (1, 1)$, satisfying condition (67).

Now we make the induction step. We will use the following general argument, Let $G$ be some diagram obtained by merging $s_D$ diagrams $D_{2\nu}$ and some number of diagrams $R_\nu$, in some order. Suppose that $G$ has a contraction – call it $\widehat{G}$ – that has $n$ $H$-nodes and $q$ $p$-nodes and admits a pairing. We will then construct a diagram $G'$ obtained by merging $s_D - 1$ diagrams $D_{2\nu}$ and some number of diagrams $R_\nu$, in some order, such that $G'$ has some contraction $\widehat{G'}$ admitting a pairing and having $n'$ $H$-nodes and $q'$ $p$-nodes, where

$$q' + \frac{\nu}{2}n' \geq q + \frac{\nu}{2}n - \nu + 1. \tag{68}$$

Condition (67) then follows for $G$ by the induction hypothesis:

$$q \leq q' + \frac{\nu}{2}(n' - n) + \nu - 1 \tag{69}$$

$$\leq 1 + (\nu - 1)(s_D - 1) - \frac{\nu}{2}(n' - 1) + \frac{\nu}{2}(n' - n) + \nu - 1 \tag{70}$$

$$= 1 + (\nu - 1)s_D - \frac{\nu}{2}(n - 1). \tag{71}$$

Now we detail the construction of $G'$. The given diagram $G$ can be viewed as obtained by first merging $s_D - 1$ diagrams $D_{2\nu}$ and some diagrams $R_\nu$ into a diagram $G_1$, then merging $G_1$ with $D_{2\nu}$, and then merging with some number of $R_\nu$'s[1]:

$$G \in (\ldots((G_1 \star D_{2\nu}) \star R_\nu)\ldots) \star R_\nu. \tag{72}$$

Observe that in the present case of symmetric models of even order $\nu$, merging with $R_\nu$ simply adds an even number of edges between some $H$-node and some $p$-node. If any diagram $F$ merged with $R_\nu$ admits a pairing, then $F$ also admits a pairing, with the same contraction (if necessary, we can rewire a pairing in $F \star R_\nu$ so that the edges of $F$ are paired within themselves, and the newly added edges are also paired within themselves).

Thus, without loss of generality we can ignore the trailing mergers with $R_\nu$ in Eq. (72) and simply assume that

$$G \in G_1 \star D_{2\nu}. \tag{73}$$

Merging $G_1$ with $D_{2\nu}$ over an edge $u_{ki}$ in $G_1$ consists in creating $\nu - 1$ new $p$-nodes $i_1, \ldots, i_{\nu-1}$ and one new $H$-node $k'$, and replacing the edge $u_{ki}$ with new edges:

$$u_{ki} \rightsquigarrow u_{k'i} \prod_{m=1}^{\nu-1} u_{ki_m} u_{k'i_m}. \tag{74}$$

Now consider several possibilities regarding the structure of the contracted diagram $\widehat{G}$.

---

[1]Here, the symbol $\in$ means that $G$ is one of the several diagrams produced by merging.

1. **The new $H$-node $k'$ is not contracted to any other node in $\widehat{G}$.** In this case we will show that a desired diagram $G'$ exists with

$$n' = n - 1, \quad q' \geq q - \frac{\nu}{2} + 1, \tag{75}$$

which satisfies condition (68).

Specifically, since the new $H$-node $k'$ is not contracted to any other node, the incident edges $(u_{k'i_m})_{m=1,\dots,\nu-1}$ and $u_{k'i}$ must be paired within each other. This requires at least $\frac{\nu}{2}$ contractions among the $\nu$ $p$-nodes $i_1,\dots,i_{\nu-1}$ and $i$. Consider now the pairings in $\widehat{G}$ of the new edges $(u_{ki_m})_{m=1,\dots,\nu-1}$ involving the old node $k$. Without loss of generality (by rewiring the pairings if necessary), we can assume these pairings to mimic the respective pairings of the edges $(u_{k'i_m})_{m=1,\dots,\nu-1}$ and $u_{k'i}$, since they can be served by the same contractions of the nodes $i_1,\dots,i_{\nu-1}$ and $i$. The only exception involves the edge paired with $u_{k'i}$ – call it $u_{k'j}$ (where $j \in \{1,\dots,i_{\nu-1}\}$). The edge $u_{kj}$ is then paired in $\widehat{G}$ with some other edge $u_{kh}$, where $h$ is some node in $G$ contracted with $i$.

Now define $G'$ simply as $G_1$, and define the contraction $\widehat{G}'$ to be inherited from the contraction $\widehat{G}'$. Since $k'$ was not contracted to any node, we have $n' = n - 1$ for the number of $H$-nodes. Also, as remarked, there were at least $\frac{\nu}{2}$ contractions among the nodes $i_1,\dots,i_{\nu-1}$ and $\nu$, so removing these $\nu$ $p$-nodes means removing at most $\frac{\nu}{2} - 1$ contracted $p$-nodes nodes. This ensures condition (75). The pairing admitted by $\widehat{G}'$ is the pairing inherited from $G$, with the exception that, on the one hand, we now need to pair the edge $u_{ki}$ not present in $G$, and we need to pair the edge $u_{kh}$ that in $\widehat{G}$ was paired with the edge $u_{kj}$ not present in $G'$. However, as pointed out, the nodes $j$ and $h$ were contracted in $G$, so we can just pair the edges $u_{kh}$ and $u_{kj}$.

2. **The new $H$-node $k'$ is contracted to the node $k$.** In this case we will show that a desired diagram $G'$ exists with

$$n' = n, \quad q' \geq q - \nu + 1. \tag{76}$$

Indeed, in this case each edge $u_{ki_m}$ is naturally paired with $u_{k'i_m}$ (for $m = 1,\dots,\nu-1$), and also $u_{k'i}$ can be identified with $u_{ki}$. By rewiring if necessary, we can assume this pairing structure without loss of generality. We can then again choose $G'$ simply as $G_1$ with inherited contraction – this gives above conditions for $n', q'$. The pairing in $G$ is naturally restricted to a pairing in $G'$.

3. **The new $H$-node $k'$ is not contracted to the node $k$, but is contracted to some other nodes in $G$.** In this case we will again show that a desired diagram $G'$ exists with

$$n' = n, \quad q' \geq q - \nu + 1. \tag{77}$$

Let us collectively denote all the nodes contracted to $k$ in $G$ by $K$, and all the nodes contracted to $k'$ by $K'$. Also collectively denote by $I_1,\dots,I_r$ all the contracted groups of $p$-nodes to which the nodes $i_1,\dots,i_{\nu-1}$ and $i$ belong. The number $r$ of such groups is, clearly, not larger than $\nu$.

Since $k'$ is not contracted to $k$, $k$ does not belong to $K'$, and $k'$ to $K$. Accordingly, the new edges $(u_{ki_m})_{m=1,\dots,\nu-1}$ cannot be paired with edges $(u_{k'i_m})_{m=1,\dots,\nu-1}$. If $i_m$ belongs to a contracted group $I_t$, the edge $u_{ki_m}$ is paired with some edge $u_{\widetilde{k}\widetilde{j}}$ with $\widetilde{k} \in K, \widetilde{j} \in I_t$, and the edge $u_{k'i_m}$ is paired with some edge $u_{\widetilde{k}'\widetilde{j}'}$ with $\widetilde{k}' \in K, \widetilde{j}' \in I_t$. The latter also applies to the edge $u$

Now define $G'$ as $G_1$. Define the contraction $\widehat{G}'$ as inherited from $\widehat{G}$, but with the additional contraction of all the groups $I_1,\dots,I_r$ into a single group $I$. Since $k'$ was contracted and $r \leq \nu$, we have conditions (77).

Now we check that there is pairing admitted by this contraction. Consider the set of edges $u_{\widetilde{k}'\widetilde{j}'}$ introduced above. They were paired in $G$ with the edges $u_{k'i_m}$ and $u_{k'i}$ that are no longer present in $G'$. Since $\nu$ is even, the number of such edges $u_{\widetilde{k}'\widetilde{j}'}$ is even. Since we have contracted all the sets $I_1,\dots,I_r$ into $I$, all these edges have the same contracted nodes ($p$-node $I$ and $H$-node $K'$) and can be paired.

A similar argument applies to the edges $u_{\widetilde{k}\widetilde{j}}$, but the number of these edges is odd. Accordingly, we can pair all of them with each other except for one. However, we recall that $G_1$ has the edge $u_{ki}$ not present in $G$. The unpaired edge $u_{\widetilde{k}\widetilde{j}}$ can then be paired with this $u_{ki}$.

All the other edge pairs in $\widehat{G}'$ are inherited from $\widehat{G}$.

This completes the induction step and thus the proof for symmetric models with even $\nu$.

### E.2 ASYMMETRIC MODELS

#### E.2.1 REALIZABILITY

We need to prove that for each $s, n, s_D$ such that

$$0 \leq s_D \leq s+1, \quad 1 \leq n \leq s_D+1, \quad s_R = s+1-s_D \text{ is even and } (s_D, n) \neq (s+1, s+2), \quad (78)$$

there exists a diagram $G$ constructed by merging $s_D$ diagrams $D_{2\nu}$ and $s_R$ diagrams $R_\nu$, and its contraction $\widehat{G}$ admitting a pairing and having $q$ $p$-nodes and $n$ $H$-nodes, where

$$q = 1 + (\nu - 1)(s_D + 1 - n). \quad (79)$$

As in the symmetric, even-$\nu$ case discussed in Section E.1.1, all the diagrams with the same triplet $(q, n, 2l)$ (where $2l$ is the number of edges) contribute a coefficient of the same sign, so we just need to produce one such diagram.

The key difference between the present asymmetric case and the symmetric case discussed in Section E.1.1 is that the edges are now colored, so edge pairing requires not only that the edges have the same node (after contraction), but also the same color.

An important observation is that merging any diagram $G$ with $R_\nu$ changes the parity of the total number of edges of each particular color, while merging with $D_{2\nu}$ preserves this parity. In particular, this explains the constraint in (78) that the number $s_R$ of mergers with $R_\nu$ must be even.

The second special constraint in (78), that $(s_D, n) \neq (s+1, s+2)$, means that diagrams $D_{2\nu}^{\star(s+1)}$, involving only $D_{2\nu}$ but not $R_\nu$, cannot admit an edge pairing without contracting at least two $H$-nodes. This follows by observing that the new $H$-node created by merging any diagram with $D_{2\nu}$ has $\nu$ edges of different colors (one per color), so must be contracted to some other nodes for a valid edge pairing.

The *n-retaining* transformation considered in Section E.1.1 (see Eq. (64)) remains valid in the present colored context since it involves pairing same-color edges of $D_{2\nu}$. In contrast, the *n-increasing* transformation (see Eq. (66)) is no longer valid since it involves pairing edges that are now differently colored.

It is possible to consider a suitable version of the $n$-increasing transformation in the present colored context, but it requires extra assumptions on the structure of the diagram $G$ appearing in the induction step. Therefore, we find it more convenient to directly construct a suitable diagram for given $(s, s_D, n)$.

First, consider the case $s_D = 0$. Note that this case is realized only if $s$ is odd (otherwise $s_R = s+1$ is odd). The relevant diagrams are those appearing in $R_\nu^{\star(s+1)}$. These diagram have one $p$-node and one $H$-node connected by multiple edges. By above remark, since $s + 1$ is even, the parity of the number of edges of each color is even, so there is a valid pairing, as desired.

Note also that, similarly and more generally, given any diagram $G$ admitting an edge pairing, we can always merge it with an even number of diagrams $R_\nu$ without changing the number of nodes and so as to extend the pairing.

Now consider a general case with $s_D \geq 1$. It is useful to construct an auxiliary "ring-like" diagram $D_{2\nu}^{\star s_D}$ as follows. Pick some color, say $m = \nu$. Merge $s_D$ diagrams $D_{2\nu}$, always over an edge of this color. As a result we get a ring-like diagram with $s_D + 1$ $H$-nodes and $\nu + s_D(\nu - 1)$ $p$-nodes:

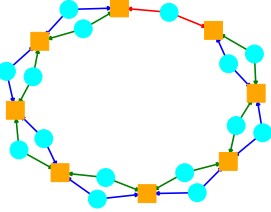

(In this figure $\nu = 3, s_D = 7$.) There are two edges of color $\nu$ (red in above figure) connecting one $p$-node to two $H$-nodes. All the other edges form $s_D$ groups of $2(\nu - 1)$ edges of colors $1, \dots, \nu - 1$ (blue and green in above figure), each connecting $\nu - 1$ $p$-nodes to two $H$-nodes.

Let us now consider separately the cases $n \le s_D$ and $n = s_D + 1$.

1. $n \le s_D$. In this case we can construct a desired diagram with $n$ $H$-nodes and $q = 1 + (\nu - 1)(s_D - 1 - n)$ $p$-nodes by suitably contracting the above ring-like diagram. Specifically, first contract the two $H$-nodes having the $\nu$-colored edge:

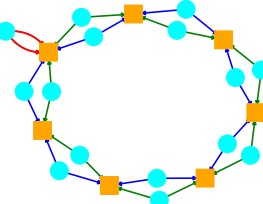

Then additionally contract a contiguous sequence of $s_D - n + 1$ $H$-nodes so as to bring the total number of $H$-nodes to $n$:

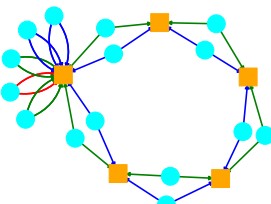

Finally, for each color, contract all the $(\nu - 1)()$ $p$-nodes remaining in the ring and incident to edges of this color:

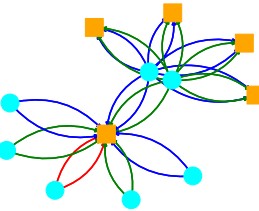

The result is a diagram with exactly $n$ $H$-nodes and $q = 1 + (\nu - 1)(s_D - n + 1)$ $p$-nodes, as desired. Thanks to the implemented contractions of $H$-nodes and $p$-nodes, the diagram admits edge pairing.

The diagram was constructed without merging with diagrams $R_\nu$, but, as remarked, we can merge it with $s_R = s + 1 - s_D$ such diagrams (assuming this number is even) while maintaining the numbers of $H$- and $p$-nodes and extending the edge pairing.

2. $n = s_D + 1$. This case requires all $H$-nodes to stay uncontracted. However, the constructed ring-like diagram $D_{2\nu}^{\star s_D}$ contains two $\nu$-colored edges connected to different $H$-nodes, and so it cannot admit an edge pairing without contracting these nodes. But we can resolve this issue by merging the ring diagram twice with $R_\nu$ over these two special edges:

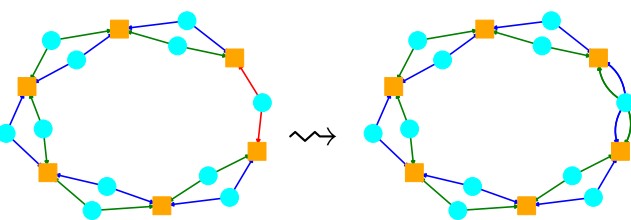

The resulting diagram then has only edges of colors $1, \ldots, \nu - 1$. Edges of each color form a circle going through each $H$-node; all the circles also connect at one $p$-node. We can contract all the $p$-nodes into one node, getting a flower-like diagram with $s_D + 1$ petals, each consisting of $\nu - 1$ pairs of edges of colors $1, \ldots, \nu - 1$:

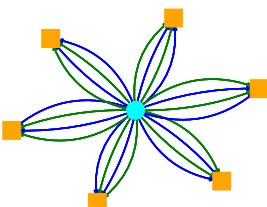

Such a diagram admits an edge pairing and has $(n, q) = (s_D + 1, 1)$, as desired. We can then further merge it with an even number of diagrams $R_\nu$ while keeping these properties.

The described construction requires merging with two diagrams $R_\nu$, so, as expected, it works for all $s$ except $s + 1 = s_D$, when $s_R = 0$ and $n = s + 2$. This is exactly the exceptional, unrealized case.

### E.2.2 OPTIMALITY

We have already argued in Section E.2.1 that the terms with odd $s_D$ and $(s_D, n) = (s+1, s+2)$ are missing from the expansion. It remains to establish a bound on feasibile $q$ analogous to the bound (67) from the symmetric, even $\nu$ scenario:

$$q \leq 1 + (\nu - 1)(s_D + 1 - n). \tag{80}$$

We generally follows the logic of the proof given in Section E.1.2 for that scenario, though the present asymmetric scenario is slightly more complex due to a more subtle influence of diagrams $R_\nu$.

As before, we perform induction on the number $s_D$ of factors $D_{2\nu}$ appearing in a sequence of merging operations (along with some factors $R_\nu$). The base of induction is $s_D = 0$, i.e. diagrams $R_\nu^{s+1}$. They have a vanishing contribution for $s$ even, and a nonzero contribution for $s$ odd with $n = q = 1$, by parity considerations.

For the induction step, suppose again that a diagram $G$ is obtained from a diagram $G_1$ by first merging it with $D_{2\nu}$ and then additionally merging with some number of $R_\nu$'s:

$$G \in (\ldots ((G_1 \star D_{2\nu}) \star R_\nu) \ldots) \star R_\nu. \tag{81}$$

Consider an $m$-colored edge $u_{ki}^{(m)}$ in $G_1$ with a $H$-node $k$ and $p$-node $i$. Without loss of generality, assume $m = \nu$. The merger $G_1 \star D_{2\nu}$ then replaces this edge by

$$u_{ki}^{(\nu)} \rightsquigarrow u_{k'i}^{(\nu)} \prod_{m=1}^{\nu-1} u_{ki_m}^{(m)} u_{k'i_m}^{(m)}. \tag{82}$$

In contrast to the symmetric, even-$\nu$ scenario, we can no longer discard now all the trailing diagrams $R_\nu$, because now they can significantly affect the pairing possibilities.

Observe, however, that merging has *partial commutativity*, in the following sense. Given any three diagrams $G_1, G_2, G_3$, consider the difference between the merging sequences $(G_1 \star G_2) \star G_3$ and $(G_1 \star G_3) \star G_2$. Note that if in $(G_1 \star G_2) \star G_3$ the merger with $G_3$ is performed over an edge in $G_1 \star G_2$ already present in $G_1$, then the same resulting diagram will appear in $(G_1 \star G_3) \star G_2$. In other words, the difference between $(G_1 \star G_2) \star G_3$ and $(G_1 \star G_3) \star G_2$ is exclusively due to the second mergers being over new edges created in the first mergers.

In particular, in sequence (81) we can commute all mergers with $R_\nu$ not performed over the new edges $u_{k'i}^{(\nu)}, (u_{ki_m}^{(m)}, u_{k'i_m}^{(m)})_{m=1}^{\nu-1}$ to be performed before the merger with $D_{2\nu}$. Accordingly, we can assume without loss of generality that the mergers with $R_\nu$ in (81) are performed over these new edges or the edges resulting from subsequent mergers with $R_\nu$.

Moreover, if we merge a diagram $G_1$ with $R_\nu$ over some edge with nodes $k$ and $i$, this simply adds new edges between these two nodes. If we continue the process by merging again with $R_\nu$ over some edges between $k$ and $i$, each merger changes the parity of the number of edges of each color. Accordingly, if the initial diagram admitted an edge pairing, then the diagram resulting after an even number of such mergers will admit an edge pairing. This shows, in particlar, that in (81) we don't need to consider more than one merger with $R_\nu$ over each specific edge among $u_{k'i}^{(\nu)}, (u_{ki_m}^{(m)}, u_{k'i_m}^{(m)})_{m=1}^{\nu-1}$.

We consider now the particular options of transition (analogous to those in Section E.1.2) from $G$ given by (81) to a smaller diagram $G'$ also admitting an edge pairing. To establish desired condition (80), we ensure that at each step we have

$$q' + (\nu - 1)n' \geq q + (\nu - 1)(n - 1). \tag{83}$$

1. **The new $H$-node $k'$ is not contracted to any other node in $\widehat{G}$.** In this case we show that a desired diagram $G'$ exists with

$$n' = n - 1, \quad q' \geq q, \tag{84}$$

   which satisfies condition (83).

   First we claim that, by color parity considerations, existence of an edge pairing in $G$ in this case requires the sequence (81) to include some trailing mergers with $R_\nu$ over the edges $(u_{k'i_m}^{(m)})_{m=1,\dots,\nu-1}$ and $u_{k'i}^{(\nu)}$; moreover all new $p$-nodes $i_1, \dots, i_\nu$ must be contracted together and to $i$.

   Indeed, take some subset $A$ of the nodes $i_1, \dots, i_{\nu-1}, i$ and consider the vector of total color parities of the edges between $k'$ and $A$ (one component of the vector corresponds to one color). Before merging with $R_\nu$, this vector has odd components for nodes in $A$ and even components for nodes in $\{i_1, \dots, i_{\nu-1}, i\} \setminus A$. Merging with $R_\nu$ over any edge between $k'$ and $A$ changes the parity of each of the $\nu$ colors. If $A$ represents a set of contracted nodes, then for an edge pairing we need to make each component in the respective color parity vector even, This is only possible if $A$ is the whole set $\{i_1, \dots, i_{\nu-1}, i\}$ and mergers with $R_\nu$ are performed an odd number of times over some edges between $k'$ and $A$.

   As a result, if we consider the contraction of the diagram $G_1$ inherited from the contraction of the diagram $G$ admitting a pairing, then it has the same number of $p$-nodes and one fewer $H$-node, in agreement with (84). However, we still need to endow $G_1$ with an edge pairing.

   As usual, we form the edge pairing by inheriting it with suitable modifications from the one in $G$. As discussed, the new edges in $G$ connecting $k'$ with $i_1, \dots, i_{\nu-1}, i$ were paired within each other, so we only need to consider pairings involving the edges connecting $k$ with $(i_m)_{m=1,\dots,\nu-1}$ and $i$. In $G$, all these $p$-nodes are contracted and the color parity vector for the edges $u_{ki_1;1}, \dots, u_{ki_{\nu-1}}^{(\nu-1)}$ in $G$ is $(1, \dots, 1, 0)$. Edge pairing is ensured by complementary edges either present in $G_1$ between $p$-node $i$ and some $H$-nodes contracted with $k$, or created by merging with $R_\nu$. The respective complementary parity vector is also $\mathbf{v} = (1, \dots, 1, 0)$; if we exclude merging with $R_\nu$, then it may be $\mathbf{v}' = (0, \dots, 0, 1)$. Now in $G_1$ we have the edge $u_{ki}^{(\nu)}$ instead of the edges $u_{ki_1;1}, \dots, u_{ki_{\nu-1}}^{(\nu-1)}$ in $G$. This edge has exactly the parity vector $\mathbf{v}'$. By merging with $R_\nu$ over this edge, we can also create a set of edges with the parity vector $\mathbf{v}$. Thus, if $G$ admits a pairing, then also either $G_1$ or $G_1 \star G$ admits a pairing.

2. **The new $H$-node $k'$ is contracted to the node $k$.** In this case a desired diagram $G'$ exists with

$$n' = n, \quad q' \geq q - \nu + 1. \tag{85}$$

   Indeed, in this case each edge $u_{ki_m}^{(m)}$ is naturally paired with $u_{k'i_m}^{(m)}$ (for $m = 1, \dots, \nu - 1$), and also $u_{k'i}^{(\nu)}$ can be identified with $u_{ki}^{(\nu)}$. If there were no subsequent mergers with $R_\nu$ over these edges (afer the merger with $D_{2\nu}$), then the desired $G'$ is simply $G_1$ with the pairing inherited from $G$ by removing the pairs $(u_{k'i_m}^{(m)}, u_{ki_m}^{(m)})$.

   Suppose now that there have been mergers with $R_\nu$ over the edges in question, say with $u_{ki_m}^{(m)}$ with some $m \in \{1, \dots, \nu - 1\}$. Such a merger changes the color parity vector of the node $i_m$ from $(0, \dots, 0)$ to $(1, \dots, 1)$. Paring the resulting edges requires the node $i_m$

to be contracted to some other node $i'$ (or several nodes) that provide counterpart edges. These counterpart edges must collectively also have the color parity vector $(1, \ldots, 1)$. If we remove the edges at the node $i_m$, since their color parity vector is $(1, \ldots, 1)$, they can be compensated in the pairs with edges at the node $i'$ by new edges at $i'$ created through merging with $R_\nu$ over one of the existing edges.

This argument shows that in any case we can take $G'$ as $G_1$, possibly merged with some $R_\nu$. The bound (85) holds in all cases (if there are extra mergers with $R_\nu$, the bound can even be strengthened since in these cases some of the vertices $i_m$ must be contracted).

3. **The new $H$-node $k'$ is not contracted to the node $k$, but is contracted to some other nodes in $G$.** In this case, again, a desired diagram $G'$ exists with

$$n' = n, \quad q' \geq q - \nu + 1. \tag{86}$$

The argument is similar to the respective proof for symmetric models in Section E.1.2 and also to arguments above (remove the nodes $i_1, \ldots, i_{\nu-1}, i$ and $k'$ and contract together all the $p$-nodes contracted with some of $i_1, \ldots, i_{\nu-1}, i$; possibly merge $G_1$ with $R_\nu$ to compensate for the removed edges).

This completes the proof of the theorem in the asymmetric scenario.

## F    RECURRENCE RELATIONS AND PARTIAL DIFFERENTIAL EQUATIONS

We clarify the context of Theorem 2 and provide its proof. Recall the statement of Theorem 2:

**Theorem 2.** *Given a formal multivariate power series $f(\mathbf{x}) \sim \sum_{\mathbf{n} \in \mathbb{N}_0^d} C_\mathbf{n} \mathbf{x}^\mathbf{n}$, suppose that its coefficients satisfy the conditions $\sum_{\mathbf{k} \in K} C_{\mathbf{n}+\mathbf{k}} P_\mathbf{k}(\mathbf{n} + \mathbf{k}) = 0, \quad \forall \mathbf{n} \in \mathbb{Z}^d$, where $K$ is a finite subset of $\mathbb{Z}^d$, $P_\mathbf{k}$ are some $d$-variate polynomials, and $C_\mathbf{n} = 0$ if $\mathbf{n} \in \mathbb{Z}^d \setminus \mathbb{N}_0^d$. Then $f$ formally satisfies the differential equation $\sum_{\mathbf{k} \in K} \mathbf{x}^{-\mathbf{k}} P_\mathbf{k}(\mathbf{x}\frac{\partial}{\partial \mathbf{x}}) f = 0$.*

Here, the coefficients $C_\mathbf{n}$ appearing in the definition of the generating function are indexed by $\mathbf{n} \in \mathbb{N}_0$, where $\mathbb{N}_0 = \{0, 1, 2, \ldots\}$. However, padding the coefficients by 0 allows us to formally extend summation to all $\mathbf{n} \in \mathbb{Z}$:

$$f(\mathbf{x}) \sim \sum_{\mathbf{n} \in \mathbb{Z}^d} C_\mathbf{n} \mathbf{x}^\mathbf{n}. \tag{87}$$

It is important for the recurrence to hold for all $\mathbf{n} \in \mathbb{Z}^d$ (otherwise a proper handling of the boundary conditions would be needed).

The differential operator $\sum_{\mathbf{k} \in K} \mathbf{x}^{-\mathbf{k}} P_\mathbf{k}(\mathbf{x}\partial_\mathbf{x})$ appearing in the statement is understood in the sense

$$\sum_{\mathbf{k} \in K} x_1^{-k_1} \cdots x_d^{-k_d} P_\mathbf{k}(x_1 \partial_{x_1}, \ldots, x_d \partial_{x_d}), \tag{88}$$

where $\mathbf{x} = (x_1, \ldots, x_d)$ and $\mathbf{k} = (k_1, \ldots, k_d)$. When applied to a formal power series (87), such a differential operator naturally produces another formal power series with well-defined coefficients. The statement of the theorem is that this series vanishes, i.e. all the coefficients are 0.

*Proof of Theorem 2.* For any polynomial $P$

$$P(\mathbf{x}\tfrac{\partial}{\partial \mathbf{x}}) f \sim \sum_\mathbf{n} C_\mathbf{n} P(\mathbf{n}) \mathbf{x}^\mathbf{n} = \sum_\mathbf{n} C_{\mathbf{n}+\mathbf{k}} P(\mathbf{n} + \mathbf{k}) \mathbf{x}^{\mathbf{n}+\mathbf{k}}. \tag{89}$$

It follows that

$$\sum_{\mathbf{k} \in K} \mathbf{x}^{-\mathbf{k}} P_\mathbf{k}(\mathbf{x}\tfrac{\partial}{\partial \mathbf{x}}) f \sim \sum_{\mathbf{k} \in K} \sum_\mathbf{n} C_{\mathbf{n}+\mathbf{k}} P_\mathbf{k}(\mathbf{n} + \mathbf{k}) \mathbf{x}^\mathbf{n} \sim 0. \tag{90}$$

$\square$

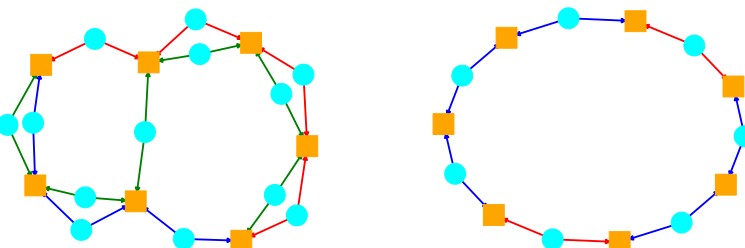

(a) Uncontracted diagrams from $D_6^{\star s}$ (**left**) and $D_4^{\star s}$ (**right**).

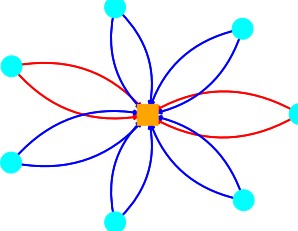

(b) An optimally contracted diagram for the free underparameterized regime: a "flower" with one $H$-node and $\nu + (\nu - 1)s$ "petals". Each petal has two edges of a matching color.

Figure 5: Diagrams from the free evolutions $D_{2\nu}^{\star s}$.

## G   UNDERPARAMETERIZED FREE EVOLUTION

**General diagram expansion.**   Recall that free evolution is a special regime where the target tensor consist of only zeros, or equivalently when the target is just removed from the loss (2). Consequently, the expected loss evolution in our diagram expansion includes only the diagrams $D_{2\nu}$ and no diagrams $R_\nu$:

$$\mathbb{E}[L(t)] \sim \sum_{s=0}^{\infty} \mathbb{E}[(\tfrac{1}{2} D_{2\nu})^{\star(s+1)}] \frac{(-t)^s}{T^s s!} \sim \sum_{s=0}^{\infty} \mathbb{E}[D_{2\nu}^{\star(s+1)}] \frac{(-t)^s}{2^{s+1} T^s s!}. \tag{91}$$

Se Figure 5 (a) for examples of diagrams in $D_{2\nu}^{\star s}$.

The large-$p, H$ behavior of the loss function is determined by Pareto-optimal terms described in Theorem 1 and corresponding to minimal contractions of diagrams in $D_{2\nu}^{\star s}$. The combinatorics of such minimal contractions is complicated for general free regimes. However, it is substantially simplified in extreme cases, in particular in the underparameterized regime ($H \ll p^{\nu-1}$ in the asymmetric or $H \ll p^{\nu/2}$ in the symmetric even-$\nu$ scenario).

In this regime, by Theorem 1, the leading terms in $\mathbb{E}[(\tfrac{1}{2} D_{2\nu})^{\star(s+1)}]$ are given by contracted diagrams that have $n = 1$ $H$-node and $q = \nu + (\nu - 1)s$ $p$-nodes. Such diagrams have the form of "flowers" with $\nu + (\nu - 1)s$ "petals", each consisting of a $p$-node and two edges of matching colors (see Fig. 5 (b)). This contraction has a unique edge pairing.

This picture applies to both symmetric and asymmetric scenario, the difference is only that in the symmetric scenario there is only one color. The arguments in both scenarios will be similar, up to different numerical coefficients due to different color distributions.

**Asymmetric scenario.**   For convenience, define the generating function $f(x)$ by

$$f(x) = \sum_{s=0}^{\infty} C_s x^s, \tag{92}$$

where the coefficient $C_s$ multiplied by $s!$ is the total number of minimally contracted diagrams (flowers) produced in $\mathbb{E}[D_{2\nu}^{\star(s+1)}]$. By Eq. (91), we can relate the loss to $f$ by

$$\mathbb{E}[L(t)] \sim \sum_{s=0}^{\infty} C_s s! p^{\nu+(\nu-1)s} H \sigma^{2(\nu+(\nu-1)s)} \frac{(-t)^s}{2^{s+1}T^s s!} \tag{93}$$

$$= \frac{p^\nu H \sigma^{2\nu}}{2} f(-p^{\nu-1}\sigma^{2(\nu-1)}t/(2T)). \tag{94}$$

Observe that a contracted diagram in $D_{2\nu}^{\star(s+1)}$ is obtained by merging a contracted diagram in $D_{2\nu}^{\star s}$ with $D_{2\nu}$, and there are $2 \cdot 2(1 + (\nu - 1)s)$ possibilities for that, since $D_{2\nu}^{\star s}$ has $2(1 + (\nu - 1)s)$ edges, and there are 2 edges of matching color in $D_{2\nu}$. It follows that

$$sC_s = 4(1 + (\nu - 1)s)C_{s-1}. \tag{95}$$

This condition holds even for $s = 0$, thanks to the vanishing l.h.s. and r.h.s. Using Theorem 2, we then obtain the ODE

$$xf' = 4x(\nu f + x(\nu - 1)f'), \tag{96}$$

or, simplifying,

$$f' = 4(\nu f + x(\nu - 1)f'). \tag{97}$$

Taking into account the initial condition $f(0) = C_0 = 1$, its solution is

$$f(x) = (1 - 4(\nu - 1)x)^{-\frac{\nu}{\nu-1}}. \tag{98}$$

We thus obtain

$$\mathbb{E}[L(t)] \sim \frac{p^\nu H \sigma^{2\nu}}{2}(1 + 2(\nu - 1)p^{\nu-1}\sigma^{2(\nu-1)}t/T)^{-\frac{\nu}{\nu-1}}, \tag{99}$$

as claimed.

**Symmetric even-$\nu$ scenario.** The arguments in this scenario are completely similar, up to replacing the coefficient 4 in Eq. (95) by $4\nu$, since in this case any edge in $D_{2\nu}$ can be used for merging.

This completes the proof of Theorem 3.

## H  UNDERPARAMETERIZED ASYMMETRIC MODEL WITH $\nu = 2$

**The diagram expansion of the asymmetric $\nu = 2$ model.**  Recall our general diagram expansion (9), which in the case $\nu = 2$ reads

$$\mathbb{E}[L(t)] \sim \frac{p}{2} + \sum_{s=0}^{\infty} \mathbb{E}[(\tfrac{1}{2}D_4 - R_2)^{\star(s+1)}]\frac{(-t)^s}{T^s s!}. \tag{100}$$

The diagrams $D_4$ and $R_2$ are show in Fig. 6 (a), (b). The edges in the diagrams can be of two colors, $m = 1, 2$. Merging a given diagram $G$ with diagram $D_4$ over a particular edge $u_{ki}^{(m)}$ in $G$, where $k$ is a $H$-node, $i$ is a $p$-node and $m$ is a color, replaces this edge by a sequence of three edges, thereby adding two new nodes $k', i'$ and two edges of another color:

$$u_{ki}^{(m)} \rightsquigarrow u_{ki'}^{(3-m)} u_{k'i'}^{(3-m)} u_{k'i}^{(m)}. \tag{101}$$

Thus, diagrams obtained by repeated mergers of diagrams $D_4$ (i.e., "free" diagrams) have a circular shape with alternating $H$- and $p$-nodes, with the edges attached to $p$-nodes forming same-color pairs (Fig. 6 (c)).

Merging a given diagram $G$ with diagram $R_2$ over a particular edge in $G$ recolors this edge. As a result, "nonfree" diagrams of $(\tfrac{1}{2}D_4 - R_2)^{\star(s+1)}$ still have a circular shape, but without the color constraint. Considering the binomial expansion of $(\tfrac{1}{2}D_4 - R_2)^{\star(s+1)}$ into variously-ordered sequences of mergers of $s_D$ diagrams $D_4$ and $s_R$ diagrams $R_2$ with $s_D + s_R = s + 1$, the resulting circles have length $2(s_D + 1)$.

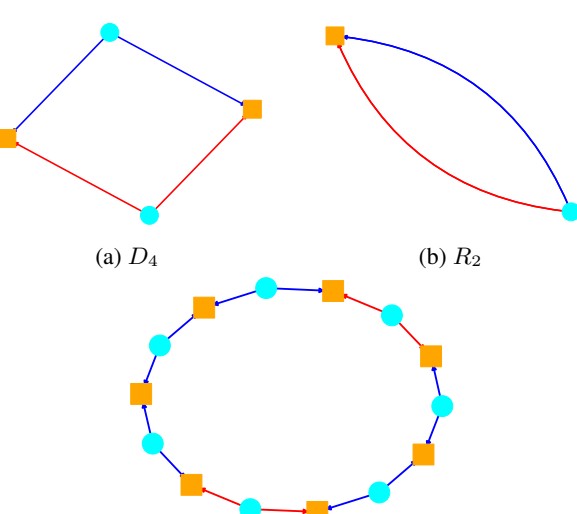

(a) $D_4$                    (b) $R_2$

(c) An uncontracted diagram from the free evolution $D_4^{\star s}$. It has a circular form with alternating $s + 1$ $p$-nodes and $s + 1$ $H$-nodes. Each $p$-node has same-color edges. The provided example is with $s = 6$.

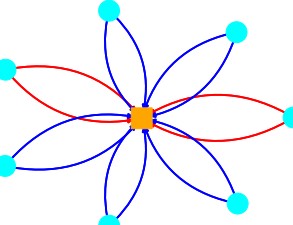

(d) An optimally contracted diagram for the free underparameterized regime: a "flower" with one $H$-node and $s + 1$ "petals".

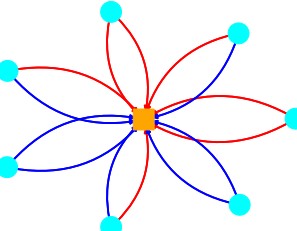

(e) Contracted non-free underparameterized diagrams from $(\frac{1}{2}D_4 - R_2)^{\star s}$. Due to merging with $R_2$, some edges are recolored. When there are petals with differently-colored edges (as in this example), the diagram does not admit an edge pairing without extra contractions.

Figure 6: Diagrams for the asymmetric model of order $\nu = 2$.

**Diagrams in the underparameterized regime.** By Theorem 1 and discussion in Section 3, the underparameterized asymmetric model is represented by Pareto-optimal terms (10) with $n = 1$, i.e., in the case $\nu = 2$,

$$p^{s_D+1}H\sigma^{2(s_D+1)}. \tag{102}$$

These terms result from contractions of diagrams in $(\frac{1}{2}D_4 - R_2)^{\star(s+1)}$ into "flowers" with a single $H$-node (i.e., the $H$-nodes are fully contracted, while the $p$-nodes remain uncontracted), see Fig. 6 (d), (e). The flower has $s_D + 1$ "petals", each consisting of a $p$-node connected to the contracted $H$-node by two edges, possibly of two difefrent colors. If there is at least one petal with differently-colored edges, the diagram does not admit an edge pairing (without extra contractions) and so does not contribute to the leading term.

Thus, we are interested only in those contracted flower diagrams in $(\frac{1}{2}D_4 - R_2)^{\star(s+1)}$ in which petals do not have differently-colored edges. However, the combinatorial coefficients associated with such special diagrams are not simply related to each other. It is convenient to consider diagrams with $q$ petals having differently-colored edges, for general $q$. This more general point of view allows to simplify the connection between combinatorial coefficients, leading to a first-order PDE for the generating function.

**The generalized generating function and its connection to the loss.** Motivated by above considerations, we introduce the generating function

$$f(x, y, z) = \sum_{s_R=0}^{\infty} \sum_{q=0}^{s_R} \sum_{s=\max(0,s_R-1)}^{\infty} C_{s_R,q,s} x^s y^{s_R} z^q. \tag{103}$$

Here, we define $C_{s_R,q,s}$ so that $C_{s_R,q,s}s!$ is the total numerical coefficient corresponding to those terms in the binomial expansion of $(\frac{1}{2}D_4 - R_2)^{\star(s+1)}$ that include $s_R$ factors $R_2$ and result in flower diagrams with exactly $q$ differently-colored petals. The coefficicient $C_{s_R,q,s}$ absorbes the factors $\frac{1}{2}$ and $-1$ appearing in $\frac{1}{2}D_4 - R_2$. Summation in $f$ is restricted to $0 \leq q \leq s_R$ because, clearly, $q$ differently-colored petals require at least $q$ recoloring of edges, i.e. $s_R \geq q$.

In our approximation of keeping only the Pareto-optimal underparameterized terms (102), the loss (100) can be written in terms of the g.f. $f$ by noting that taking only the $q = 0$ terms in $f$ corresponds to setting $z = 0$:

$$\mathbb{E}[L(t)] \sim \frac{p}{2} + \sum_{s=0}^{\infty} \sum_{s_D=0}^{s+1} C_{s+1-s_D,q=0,s} p^{s_D+1} H\sigma^{2(s_D+1)} \frac{(-t)^s}{T^s s!} \tag{104}$$

$$= \frac{p}{2} + \sum_{s=0}^{\infty} \sum_{s_R=0}^{s+1} C_{s_R,q=0,s} p^{s+2-s_R} H\sigma^{2(s+2-s_R)} \frac{(-t)^s}{T^s s!} \tag{105}$$

$$= \frac{p}{2} + p^2 H\sigma^4 f(-p\sigma^2 t/T, 1/(p\sigma^2), 0). \tag{106}$$

**The recurrence.** Diagrams in $(\frac{1}{2}D_4 - R_2)^{\star(s+1)}$ are obtained from diagrams in $(\frac{1}{2}D_4 - R_2)^{\star s}$ by merging with $D_4$ or $R_2$. In the former case, the number $q$ of petals with differently-colored edges does not change, while in the latter case it is either increased by one (if we merge over a same-colored petal) or decreased by one (if we merge over a differently-colored petal). By counting the relevant edges and taking into account the necessary coefficients, we get the recurrence

$$sC_{s_R,q,s} = -2(q+1)C_{s_R-1,q+1,s-1} \tag{107}$$

$$-2(s - s_R - q + 3)C_{s_R-1,q-1,s-1} \tag{108}$$

$$+2(s - s_R + 1)C_{s_R,q,s-1}. \tag{109}$$

Importantly, this recurrence holds for all $s_2, q, s \in \mathbb{Z}$. At the initial value $s = 0$ the reduction to $s - 1$ described above is not applicable, but the recurrence still holds since both the l.h.s. and r.h.s. vanish. The values $C_{s_R,q,s}$ do not vanish only if $0 \leq q \leq s_R \leq s + 1, s \geq 0$, and $q$ and $s_R$ have the same parity.

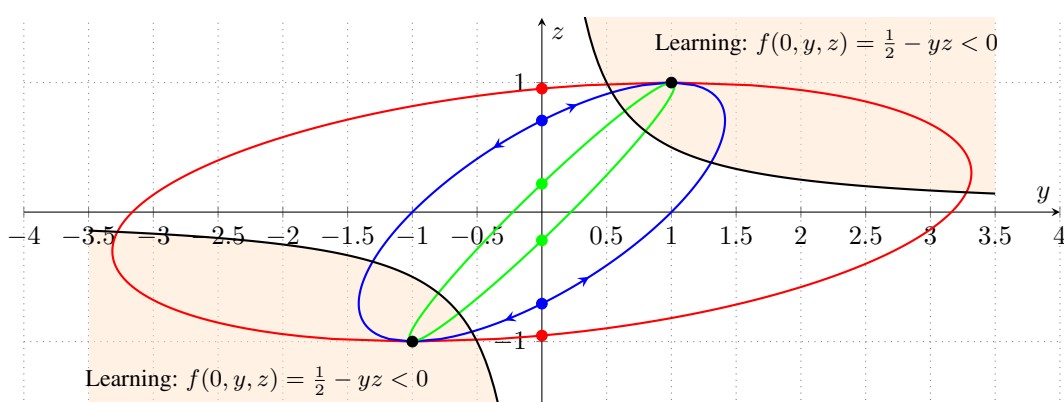

Figure 7: ODE (112) in the $yz$ plane.

**The PDE and reduction to ODE.** By Theorem 2, the recurrence yields the differential equation

$$x\partial_x f = 2x[-yz^{-1}z\partial_z - yz(x\partial_x - y\partial_y - z\partial_z + 2) + (x\partial_x - y\partial_y + 2)]f, \tag{110}$$

or equivalently

$$[(1 + 2xyz - 2x)\partial_x + 2(y - y^2 z)\partial_y + 2(y - yz^2)\partial_z]f = 4(1 - yz)f. \tag{111}$$

This is a 1'st order PDE that can be solved by the method of characteristics. Denote $\mathbf{x} = (x, y, z)$ and write the PDE as

$$\Phi(\mathbf{x})^T \nabla_F(\mathbf{x}) = \phi f(\mathbf{x}) \tag{112}$$

with the vector field

$$\Phi(\mathbf{x}) = \begin{pmatrix} 1 + 2xyz - 2x \\ 2(y - y^2 z) \\ 2(y - yz^2) \end{pmatrix} \tag{113}$$

and function

$$\phi(\mathbf{x}) = 4(1 - yz). \tag{114}$$

Then the solution at a given point $\mathbf{x}_0$ can be found as

$$f(\mathbf{x}_0) = f(\mathbf{x}_1)e^{-\int_{\tau_0}^{\tau_1} \phi(\mathbf{x}(\tau))d\tau}, \tag{115}$$

where $\mathbf{x}(\tau)$ is an integral curve of the field $\Phi$ connecting the point $\mathbf{x}_0$ to the point $\mathbf{x}_1$:

$$\dot{\mathbf{x}} = \Phi(\mathbf{x}), \quad \mathbf{x}(\tau_0) = \mathbf{x}_0, \quad \mathbf{x}(\tau_1) = \mathbf{x}_1. \tag{116}$$

By Eq. (106), finding the loss requires us to find the values of $f$ on the hyperplane $z = 0$. On the other hand, the only terms present in $f$ with $s = 0$ are $C_{s_R=0,q=0,s=0} = \frac{1}{2}$ and $C_{s_R=1,q=1,s=0} = -1$, so $f$ has a simple form on the hyperplane $x = 0$:

$$f(0, y, z) = C_{0,0,0} + C_{1,1,0}yz = \frac{1}{2} - yz. \tag{117}$$

Accordingly, we will obtain the solution $f$ on the hyperlane $z = 0$ by transfering $f$ from the plane $x = 0$ along characteristics.

**Analysis of the ODE.** The ODE (112) has subsystem $\{y, z\}$ independent of $x$. The evolution in this subsystem has the first integral

$$I = \frac{1 - z^2}{(y - z)^2}, \tag{118}$$

showing that the phase curves in the $yz$ plane are arcs of ellipses touching the lines $z = \pm 1$ at the points $(y, z) = \pm(1, 1)$ (see Figure 7).

Another first integral, involving all three variables, is

$$J = 4xy + \ln \frac{1 - z}{1 + z}. \tag{119}$$

We can also note that

$$\frac{d}{d\tau} \ln y = 2(1 - yz), \tag{120}$$

so that

$$\int \phi(\mathbf{x}(\tau)) d\tau = 4 \int (1 - yz) d\tau = 2 \ln y + c. \tag{121}$$

It follows that we can write the multiplier $e^{-\int_{\tau_0}^{\tau_1} \phi(\mathbf{x}(\tau)) d\tau}$ appearing in the solution (115) simply as

$$e^{-\int_{\tau_0}^{\tau_1} \phi(\mathbf{x}(\tau)) d\tau} = \frac{y_0^2}{y_1^2}. \tag{122}$$

**The solution.** By Eqs. (116) and (121), we have

$$f(x, y, 0) = \left(\frac{1}{2} - y_1 z_1\right) \frac{y^2}{y_1^2} \tag{123}$$

along an integral curve connecting the points

$$\mathbf{x}_0 = (x, y, 0), \quad \mathbf{x}_1 = (0, y_1, z_1). \tag{124}$$

It remains to express $y_1, z_1$ in terms of $x, y$. From the first integral $J$ we get

$$z_1 = \frac{e^{-4xy} - 1}{e^{-4xy} + 1}. \tag{125}$$

Then from the first integral $I$ we get

$$y_1 = y\sqrt{1 - z_1^2} + z_1 \tag{126}$$

$$= y\frac{2e^{-2xy}}{1 + e^{-4xy}} - \frac{1 - e^{-4xy}}{1 + e^{-4xy}} \tag{127}$$

$$= \frac{e^{-4xy} + 2ye^{-2xy} - 1}{1 + e^{-4xy}}. \tag{128}$$

It follows that

$$f(x, y, 0) = \left(\frac{1}{2} - y_1 z_1\right) \frac{y^2}{y_1^2(\tau_{x,y})} \tag{129}$$

$$= \frac{y^2[(1 + e^{-4xy})^2 + 2(1 - e^{-4xy})(e^{-4xy} + 2ye^{-2xy} - 1)]}{2(e^{-4xy} + 2ye^{-2xy} - 1)^2} \tag{130}$$

$$= y^2 \frac{1 + 2yr - r^2}{2(y - r)^2}, \quad r = \sinh(2xy). \tag{131}$$

**Interpretation of learning.** The above analysis and Figure 7 provide a simple interpretation of learning in this model. We are interested in the values $f(x, y, 0)$ that correspond to the $y$ axis in Figure 7. Different values of $x$ correspond to different points along the elliptic arc going through the point $(y, 0)$. In the context of loss function (106), the values $y$ are positive, since $y = 1/(p\sigma^2)$ and the sign of $y$ is preserved by the dynamics. On the other hand, the initial $x = -p\sigma^2 t/T < 0$. Taking larger time $t$ corresponds to going along the arc towards the point $(y, z) = (1, 1)$. At sufficiently large $t$ (i.e., large negative $x$) we get into the region $\frac{1}{2} - yz < 0$. Then, by (115) or (123), $f(x, y, 0)$ becomes negative, meaning that the loss becomes less than the loss $\frac{p}{2}$ of the trivial zero model. That makes it natural to identify the region $\frac{1}{2} - yz < 0$ as the *learning region*.

As $t \to +\infty$ (i.e., $x \to -\infty$), the characteristic trajectories converge to $(y, z) = (1, 1)$ and

$$\lim_{x \to -\infty} f(x, y, 0) = -\frac{y^2}{2}. \tag{132}$$

It follows by (106) that

$$\lim_{t \to +\infty} \mathbb{E}[L(t)] \sim \frac{p}{2} - \frac{H}{2}, \tag{133}$$

which is the natural and expected result: using a model of tensor rank $H$ allows us to learn $H$ out of $p$ nonzero components of the target tensor $\delta_{i_1 = \ldots = i_\nu}$.

## I    OVERPARAMETERIZED ASYMMETRIC MODEL WITH $\nu = 2$

In order to compute the loss, we need to count the total number of trees with exactly one p-type vertex that result from contraction of loops in $\left(\frac{1}{2}D_4 - R_2\right)^{\star(s+1)}$ for all $s \geq 0$. These trees are "flowers" with a $p$-vertex at the center and $H$-vertices at the "petals". We call loops that could be contracted to such flowers $p$-contractible. These flowers look like the flowers in Figure 4 for the underparametrized regime, but with exchanged roles of $p$- and $H$-nodes.

We call a vertex adjacent to edges of different colors *trans vertex*. A loops is $p$-contractible iff it has no trans-$H$ vertices.

**Generating function.**    Let us introduce the following generating function:

$$f_{s_R,q}(x) = \sum_{s=\max(0,s_R-1)}^{\infty} C_{s_R,q,s} x^s, \tag{134}$$

where $C_{s_R,q,s}$ is defined as follows. Let

$$\frac{1}{s!}\left(\frac{1}{2}D_4 - R_2\right)^{\star(s+1)} = \sum_{s_R=0}^{s+1} \sum_{G\in\mathcal{G}_{s_R,s}} c_G G, \tag{135}$$

where $\mathcal{G}_{s_R,s}$ is a set of all loops in the above expression resulted from using exactly $s_R$ $R_2$ terms. Then we define

$$C_{s_R,q,s} = \sum_{\substack{G\in\mathcal{G}_{s_R,s}:\\ G \text{ has } q \text{ trans-}H \text{ vertices}}} c_G. \tag{136}$$

When $H\sigma^2 \sim \rho$, while $H/p \to \infty$, we are interested only in $p$-contractible loops. Since a loop is $p$-contractible iff it has no trans-$H$ vertices, the expected loss is given by

$$\mathbb{E}[L(t)] \sim \frac{p}{2} + pH^2\sigma^4 \sum_{s_R=0}^{\infty} \frac{f_{s_R,0}(-H\sigma^2 t/T)}{H^{s_R}\sigma^{2s_R}} \sim \frac{p}{2} + p\rho^2 \sum_{s_R=0}^{\infty} \frac{f_{s_R,0}(-\rho t/T)}{\rho^{s_R}}, \tag{137}$$

where $\rho = H\sigma^2$.

**Recursive relation for C.**    We have the following recurrence:

$$\begin{aligned}
sC_{s_R,q,s} =& -2(q+1)C_{s_R-1,q+1,s-1} - 2(s+2-s_R-(q-1))C_{s_R-1,q-1,s-1}\\
&+ 2qC_{s_R,q,s-1} + 2(s+1-s_R-(q-2))C_{s_R,q-2,s-1}\\
=& -2(q+1)C_{s_R-1,q+1,s-1} - 2(s-s_R-q+3)C_{s_R-1,q-1,s-1}\\
&+ 2qC_{s_R,q,s-1} + 2(s-s_R-q+3)C_{s_R,q-2,s-1}.
\end{aligned} \tag{138}$$

By construction, $C_{s_R,q,s} = 0$ whenever $s < 0$, or $s_R > s+1$, or $s_R < 0$, or $q < 0$.

**Generalized generating function.**    Let us intoduce the following generalized generating function of three variables:

$$f(x,y,z) = \sum_{q=0}^{\infty}\sum_{s_R=0}^{\infty} f_{s_R,q}(x) y^{s_R} z^q = \sum_{q=0}^{\infty}\sum_{s_R=0}^{\infty}\sum_{s=\max(0,s_R-1)}^{\infty} C_{s_R,q,s} x^s y^{s_R} z^q. \tag{139}$$

Given the above, the loss is expressed as follows:

$$\mathbb{E}[L(t)] \sim \frac{p}{2} + p\rho^2 \sum_{s_R=0}^{\infty} \frac{f_{s_R,0}(-\rho t/T)}{\rho^{s_R}} = \frac{p}{2} + p\rho^2 f\left(-\frac{\rho t}{T}, \frac{1}{\rho}, 0\right). \tag{140}$$

The main recurrence Eq. (138) yields the following ODE:

$$\begin{aligned}
&x\partial_x f + 2xy\partial_z f\\
&\quad + 2xyz\left(x\partial_x - y\partial_y - z\partial_z + 2\right)f - 2xz\partial_z f - 2xz^2\left(x\partial_x - y\partial_y - z\partial_z + 2\right)f = 0,
\end{aligned} \tag{141}$$

or, equivalently,

$$\left(1 + 2xyz - 2xz^2\right)\partial_x f + \left(-2y^2 z + 2yz^2\right)\partial_y f$$
$$+ \left(2y - 2yz^2 - 2z + 2z^3\right)\partial_z f = \left(-4yz + 4z^2\right)f. \tag{142}$$

This is a first-order ODE which could be expressed as follows:

$$F(\mathbf{x}) \cdot \nabla f(\mathbf{x}) = \phi(\mathbf{x})f(\mathbf{x}), \tag{143}$$

where $\mathbf{x} = (x, y, z)$, and

$$F(\mathbf{x}) = \begin{pmatrix} 1 + 2xz(y - z) \\ -2yz(y - z) \\ 2(1 - z^2)(y - z) \end{pmatrix}, \qquad \phi(\mathbf{x}) = -4z(y - z). \tag{144}$$

Then the solution at a given point $\mathbf{x}_0$ is given by

$$f(\mathbf{x}_0) = f(\mathbf{x}_1)e^{-\int_{\tau_0}^{\tau_1} \phi(\mathbf{x}(\tau))\,d\tau}, \tag{145}$$

where the integral curve $\mathbf{x}(\tau)$ is defined by

$$\dot{\mathbf{x}}(\tau) = F(\mathbf{x}(\tau)), \qquad \mathbf{x}(\tau_1) = \mathbf{x}_1, \quad \mathbf{x}(\tau_0) = \mathbf{x}_0. \tag{146}$$

We are interested in the solution at $z = 0$. We already know the solution at $x = 0$:

$$f(0, y, z) = C_{0,2,0}z^2 + C_{1,1,0}yz = \frac{z^2}{2} - yz = \frac{z}{2}(z - 2y). \tag{147}$$

**The $\{y, z\}$ subsystem.** The $\{y, z\}$ subsystem is independent on $x$ and allows for the following first integral:

$$I = \frac{y^2}{1 - z^2}. \tag{148}$$

The phase curves are ellipses for $I > 0$, or hyperbolae for $I < 0$, $y^2 + Iz^2 = I$ with common points $(y, z) = (0, \pm 1)$. Since we interested in curves passing $z = 0$, $I$ has to be non-negative. If we take $z_0 = 0$ then $I = y_0^2$.

For the loss to decrease, we need $f(x, y, 0)$ to be negative. For this, we need $f(\mathbf{x}(\tau))$ to stay negative along the integral curve. Suppose $x(\tau) = 0$ for some $\tau^* \in (\tau_0, \tau_1)$. Then $f(0, y(\tau^*), z(\tau^*))$ has to be negative. Then $z(\tau^*) < 2y(\tau^*)$ whenever $z(\tau^*) > 0$, and $z(\tau^*) > 2y(\tau^*)$ otherwise. In particular, $z(\tau^*)$ and $y(\tau^*)$ have to be of the same sign. Since $z = y$ is a stationary point, $z(\tau)$ and $y(\tau)$ keep their (same) signs all along the way. Therefore we could rewrite the above first integral as

$$y = y_0\sqrt{1 - z^2}. \tag{149}$$

Note that

$$\phi(\mathbf{x}) = -4z(y - z) = 2\frac{\dot{y}}{y} = 2\frac{d\ln y}{d\tau}. \tag{150}$$

Therefore

$$e^{-\int_{\tau_0}^{\tau_1} \phi(\mathbf{x}(\tau))\,d\tau} = \frac{y_0^2}{y_1^2} = \frac{1}{1 - z_1^2}. \tag{151}$$

**General $(y, z)$-solution.**

$$\dot{y} = -2yz(y - z), \qquad \dot{z} = 2(1 - z^2)(y - z). \tag{152}$$

Given the above first integral,

$$\dot{z} = 2(1 - z^2)\left(y_0\sqrt{1 - z^2} - z\right). \tag{153}$$

This gives the solution in implicit form:

$$\tau_1 - \tau_0 = \int_{z_0}^{z_1} \frac{dz}{2(1 - z^2)\left(y_0\sqrt{1 - z^2} - z\right)} = \frac{1}{4}\ln\left(\frac{1 - z_1^2}{1 - z_0^2}\right) - \frac{1}{2}\ln\left(\frac{y_0\sqrt{1 - z_1^2} - z_1}{y_0\sqrt{1 - z_0^2} - z_0}\right). \tag{154}$$

We are ultimately interested in the case of $z_0 = 0$:

$$\tau_1 - \tau_0 = \frac{1}{4} \ln\left(1 - z_1^2\right) - \frac{1}{2} \ln\left(\sqrt{1 - z_1^2} - z_1/y_0\right) = \frac{1}{4} \ln\left(\frac{1 - z_1^2}{\left(\sqrt{1 - z_1^2} - z_1/y_0\right)^2}\right). \tag{155}$$

This gives

$$e^{4(\tau_1 - \tau_0)} = \frac{1}{\left(1 - u_1\right)^2}, \qquad u_1 = \frac{z_1}{y_0\sqrt{1 - z_1^2}}. \tag{156}$$

Hence $u_1 = 1 \pm e^{-2(\tau_1 - \tau_0)}$. We know that if $\tau_1 = \tau_0$ then $z_1 = z_0 = 0$, therefore $u_1 = 0$. Then the desired root has a minus sign:

$$u_1 = 1 - e^{-2(\tau_1 - \tau_0)}. \tag{157}$$

This gives a linear equation for $z_1^2$:

$$\frac{z_1^2}{y_0^2(1 - z_1^2)} = \left(1 - e^{-2(\tau_1 - \tau_0)}\right)^2; \tag{158}$$

$$z_1^2 = \frac{a^2}{1 + a^2}, \quad 1 - z_1^2 = \frac{1}{1 + a^2}, \quad a = y_0\left(1 - e^{-2(\tau_1 - \tau_0)}\right). \tag{159}$$

Therefore assuming $\tau_0 = 0$, the general solution for $z(0) = 0$, $y(0) = y_0$ is given by

$$z^2(\tau) = \frac{y_0^2\left(1 - e^{-2\tau}\right)^2}{1 + y_0^2\left(1 - e^{-2\tau}\right)^2}, \qquad y^2(\tau) = \frac{y_0^2}{1 + y_0^2\left(1 - e^{-2\tau}\right)^2}. \tag{160}$$

**The $x$ dynamics.**  Note that

$$\frac{d(xy)}{d\tau} = \dot{x}y + x\dot{y} = y. \tag{161}$$

Therefore

$$x(\tau_1)y(\tau_1) - x_0 y_0 = \int_0^{\tau_1} y(\tau)\, d\tau. \tag{162}$$

**The final solution.**  Since we already know the solution for $x = 0$, we take $x(\tau_1) = 0$. Then the above formula gives a condition for $\tau_1$:

$$-x_0 y_0 = \int_0^{\tau_1} y(\tau)\, d\tau = y_0 \int_0^{\tau_1} \frac{d\tau}{\sqrt{1 + y_0^2\left(1 - e^{-2\tau}\right)^2}}$$

$$= \frac{y_0}{2\sqrt{1 + y_0^2}}\left[\tanh^{-1}\left(\frac{1 + y_0^2\left(1 - e^{-2\tau_1}\right)}{\sqrt{1 + y_0^2}\sqrt{1 + y_0^2\left(1 - e^{-2\tau_1}\right)^2}}\right) - \tanh^{-1}\left(\frac{1}{\sqrt{1 + y_0^2}}\right)\right]; \tag{163}$$

$$\tanh\left(-2x_0\sqrt{1 + y_0^2} + \tanh^{-1}\left(\frac{1}{\sqrt{1 + y_0^2}}\right)\right) = \frac{1 + y_0^2\left(1 - e^{-2\tau_1}\right)}{\sqrt{1 + y_0^2}\sqrt{1 + y_0^2\left(1 - e^{-2\tau_1}\right)^2}}. \tag{164}$$

Introducing $u = y_0\left(1 - e^{-2\tau_1}\right)$, we get a quadratic equation on $u$:

$$a^2 = \frac{(1 + y_0 u)^2}{1 + u^2}, \qquad a = \sqrt{1 + y_0^2}\tanh\left(-2x_0\sqrt{1 + y_0^2} + \tanh^{-1}\left(\frac{1}{\sqrt{1 + y_0^2}}\right)\right); \tag{165}$$

$$(y_0^2 - a^2)u^2 + 2y_0 u + (1 - a^2) = 0; \tag{166}$$

$$u = \frac{-y_0 \pm \sqrt{y_0^2 - (y_0^2 - a^2)(1 - a^2)}}{y_0^2 - a^2} = \frac{-y_0 \pm \sqrt{a^2 - a^4 + y_0^2 a^2}}{y_0^2 - a^2} = \frac{-y_0 \pm a\sqrt{1 - a^2 + y_0^2}}{y_0^2 - a^2}. \tag{167}$$

If we take $x_0 = 0$ then $a = 1$, while $\tau_1$ has to be zero, which means that the expected solution is $u = 0$. Then we have to choose the "plus" sign. Plugging back $a$,

$$
\begin{aligned}
u &= \frac{-y_0 + a\sqrt{1 + y_0^2}\operatorname{sech}(v_0)}{(1 + y_0^2)\operatorname{sech}^2(v_0) - 1} \\
&= \frac{-y_0\cosh^2(v_0) + (1 + y_0^2)\sinh(v_0)}{1 + y_0^2 - \cosh^2(v_0)} = \frac{-y_0\cosh^2(v_0) + (1 + y_0^2)\sinh(v_0)}{y_0^2 - \sinh^2(v_0)},
\end{aligned}
\tag{168}
$$

$$
v_0 = -2x_0\sqrt{1 + y_0^2} + \tanh^{-1}\left(\frac{1}{\sqrt{1 + y_0^2}}\right).
\tag{169}
$$

Then

$$
\begin{aligned}
f(\mathbf{x}(\tau_0)) &= f(\mathbf{x}(\tau_1))e^{-\int_{\tau_0}^{\tau_1}\phi(\mathbf{x}(\tau))\,d\tau} = \frac{y_0^2}{y^2(\tau_1)}\left(\frac{z^2(\tau_1)}{2} - y(\tau_1)z(\tau_1)\right) \\
&= y_0^2\left(\frac{z^2(\tau_1)}{2y^2(\tau_1)} - \frac{z(\tau_1)}{y(\tau_1)}\right) = y_0^2\left(\frac{1}{2}\left(1 - e^{-2\tau_1}\right)^2 - \left(1 - e^{-2\tau_1}\right)\right) = \frac{u^2}{2} - y_0 u.
\end{aligned}
\tag{170}
$$

**Explicit loss expression.**

$$
\mathbb{E}[L(t)] \sim \frac{p}{2} + p\rho^2 f\left(-\frac{\rho t}{T}, \frac{1}{\rho}, 0\right) = \frac{p}{2}\left(1 + \rho^2 u^2 - 2\rho u\right) = \frac{p}{2}(1 - \rho u)^2,
\tag{171}
$$

where

$$
u = \frac{-\rho\cosh^2 v + (1 + \rho^2)\sinh v}{1 - \rho^2\sinh^2 v}, \qquad v = \frac{2t}{T}\sqrt{1 + \rho^2} + \tanh^{-1}\left(\frac{\rho}{\sqrt{1 + \rho^2}}\right).
\tag{172}
$$

This could be further simplified:

$$
\begin{aligned}
1 - \rho u &= \frac{1 - \rho^2\sinh^2 v + \rho^2\cosh^2 v - \rho(1 + \rho^2)\sinh v}{1 - \rho^2\sinh^2 v} \\
&= \frac{(1 + \rho^2) - \rho(1 + \rho^2)\sinh v}{1 - \rho^2\sinh^2 v} = \frac{1 + \rho^2}{1 + \rho\sinh v};
\end{aligned}
\tag{173}
$$

$$
v = \frac{2t}{T}\sqrt{1 + \rho^2} + \sinh^{-1}(\rho);
\tag{174}
$$

$$
\sinh v = \rho\cosh\left(\frac{2t}{T}\sqrt{1 + \rho^2}\right) + \sqrt{1 + \rho^2}\sinh\left(\frac{2t}{T}\sqrt{1 + \rho^2}\right).
\tag{175}
$$

This gives a final expression:

**Theorem 4.** *Consider the case when $H, p, \sigma^{-2} \to \infty$ in such a way that $H/p \to \infty$, while $H\sigma^2 \to \rho > 0$. Then for any $t, T > 0$,*

$$
\mathbb{E}[L(t)] \sim \frac{p}{2}\left(\frac{1 + \rho^2}{1 + \rho^2\cosh\psi + \rho\sqrt{1 + \rho^2}\sinh\psi}\right)^2, \qquad \psi = \frac{2t}{T}\sqrt{1 + \rho^2}.
\tag{176}
$$

For $\rho = 1$ (i.e. in the "conventional" mean-field limit), the above solution simplifies:

$$
\mathbb{E}[L(t)] \sim \frac{2p}{\left(1 + \cosh\left(\frac{\sqrt{8}t}{T}\right) + \sqrt{2}\sinh\left(\frac{\sqrt{8}t}{T}\right)\right)^2}.
\tag{177}
$$

For large $\rho$, we get $\mathbb{E}[L(t)] \sim \frac{p}{2}e^{\frac{-4\rho t}{T}}$, while as $\rho$ vanishes,

$$
\mathbb{E}[L(t)] \sim \frac{p}{2}(1 - \rho\sinh\psi)^2 \sim \frac{p}{2}\left(1 - 2\rho\sinh\left(\frac{2t}{T}\right)\right).
\tag{178}
$$

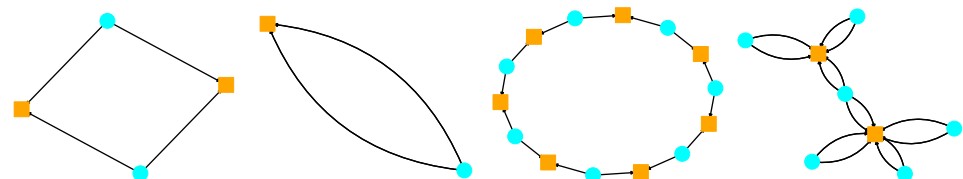

Figure 8: Diagrams in the symmetric $\nu = 2$ scenario. **Left to right:** $D_4$; $R_2$; a circular diagram from $(\frac{1}{2}D_4 - R_2)^{\star(s+1)}$; a minimal contraction of a circular diagram to a tree.

**Large time behavior.** As $t \to \infty$, we arrive at

$$\mathbb{E}[L(t)] \sim 2p \left( \frac{1 + \rho^2}{\rho^2 + \rho\sqrt{1 + \rho^2}} \right)^2 e^{-\frac{4t}{T}\sqrt{1+\rho^2}}. \tag{179}$$

## J    GENERAL SYMMETRIC MODEL WITH $\nu = 2$

**Diagram expansion.** In contrast to the asymmetric scenario, in the symmetric case the edges in the diagrams have only one color. Similarly to the symmetric scenario, all uncontracted diagrams occurring in $(\frac{1}{2}D_4 - R_2)^{\star(s+1)}$ are circular, with alternating $p$- and $H$-nodes and $2(s_D + 1)$ edges (see Fig. 8). The absence of different colors simplifies combinatorics. However, in contrast to the scenarios considered in Sections H and I, we do not impose now the assumption of over- or under-parameterization. As a result, we need to consider now general minimal contractions, not just the flowers as in Sections H and I.

By Theorem 1, a minimal contraction of a circular diagram of length $2(s_D + 1)$ has $n$ $H$-nodes and $s_D + 2 - n$ $p$-nodes, with $n \in \{1, \dots, s_D + 1\}$ (the flower contractions correspond to the extreme cases $n = 1$ and $n = s_D + 1$). It is easy to see inductively that minimal contraction contract the circle to trees as in Figure 8 (right). (Indeed, since the total number of contracted nodes is $s_D + 2$, at least one node of the circle must be uncontracted. Then its neighboring nodes must be contracted to enable edge pairing. By removing this node and its two edges we then reduce the question to a smaller $s_D$.) Alternatively, one can identify a minimal contraction with a non-crossing partition of the set of $H$-nodes (into contracted groups). The number of such contractions is known to be given by *Narayana numbers* $N_{s_D+1,n}$.

**The general generating function.** This discussion shows that within the approximation by minimal diagrams, we can write

$$\mathbb{E}[(\tfrac{1}{2}D_4 - R_2)^{\star(s+1)}] \sim \sum_{s_D=0}^{s+1} \sum_{n=1}^{s_D+1} M_{s,s_D} N_{s_D+1,n} p^{s_D+2-n} H^n \sigma^{2(s_D+1)}, \tag{180}$$

where $M_{s,s_D}$ is the coefficient in the expansion $(\frac{1}{2}D_4 - R_2)^{\star(s+1)} = \sum_{s_D=1}^{s+1} M_{s,s_D} G_{s_D}$ over circular diagrams $G_{s_D}$ of length $2(s_D + 1)$. Given our standard loss expansion (9), we can then write

$$\mathbb{E}[L(t)] \sim \frac{p}{2} + p^2 \sigma^2 \Psi(-t/T, H/p, p\sigma^2) \tag{181}$$

with the generating function

$$\Psi(x, y, z) = \sum_{s_D=0}^{\infty} z^{s_D} \sum_{s=\max(0, s_D-1)}^{\infty} \frac{M_{s,s_D}}{s!} x^s \sum_{n=1}^{s_D+1} N_{s_D+1,n} y^n. \tag{182}$$

**Reduced generating functions.** Note that the coefficient $\frac{M_{s,s_D} N_{s_D+1,n}}{s!}$ appearing in the general generating function is partially factorized, in the sense of being a product of the coefficient $\frac{M_{s,s_D}}{s!}$

depending on $s, s_D$, and the Narayana numbers $N_{s_D+1,n}$ depending on $s_D, n$. It is convenient to introduce two respective two-variable generating functions:

$$f(x, z) \sim \sum_{s=0}^{\infty} \sum_{s_D=0}^{s+1} \frac{M_{s,s_D}}{s!} x^s z^{s_D}, \tag{183}$$

$$h(z, y) \sim \sum_{s_D=0}^{\infty} \sum_{n=1}^{s_D+1} N_{s_D+1,n} z^{s_D} y^n. \tag{184}$$

The g.f. $h$ for the Narayana numbers is known:

$$h(z, y) = \frac{1 - z(y+1) - \sqrt{1 - 2z(y+1) + z^2(y-1)^2}}{2z^2}. \tag{185}$$

We will find $f$ using a reduction to 1st order PDE.

**The generating function $f$.** Reducing the case of $s$ to $s-1$ by writing $(\frac{1}{2}D_4 - R_2)^{\star(s+1)} = (\frac{1}{2}D_4 - R_2)^{\star s} \star (\frac{1}{2}D_4 - R_2)$ and considering different possibilities gives the recurrence

$$M_{s,s_D} = -4(s_D + 1)M_{s-1,s_D} + 4s_D M_{s-1,s_D-1}. \tag{186}$$

Taking into account the additional factor $\frac{1}{s!}$ and using Theorem 2, we find the PDE

$$x\partial_x f = 4x(-(z\partial_z + 1) + z(z\partial_z + 1))f, \tag{187}$$

i.e.

$$(\partial_x + 4z(1-z)\partial_z)f = 4(z-1)f. \tag{188}$$

We write this as

$$\mathbf{\Phi}^T \nabla f = \phi f, \tag{189}$$

where

$$\Phi = \begin{pmatrix} 1 \\ 4z(1-z) \end{pmatrix}, \tag{190}$$

$$\phi = 4(z-1). \tag{191}$$

The solution along an integral curve $\mathbf{x}(\tau)$ is

$$f(\mathbf{x}_0) = f(\mathbf{x}_1)e^{-\int_{\tau_0}^{\tau_1} \phi(\mathbf{x}(\tau))d\tau}. \tag{192}$$

The integral curves are given by

$$\frac{dz}{dx} = 4z(1-z), \tag{193}$$

i.e.

$$4dx = d\ln\frac{z}{1-z}. \tag{194}$$

We get a first integral

$$I = \frac{1-z}{z}e^{4x}. \tag{195}$$

We also note that

$$d\ln z = 4(1-z) = -\phi(z), \tag{196}$$

so that

$$e^{-\int_{\tau_0}^{\tau_1} \phi(\mathbf{x}(\tau))d\tau} = \frac{z_1}{z_0}. \tag{197}$$

Accordingly, along an integral curve

$$f(x_0, z_0) = f(x_1, z_1)\frac{z_1}{z_0}. \tag{198}$$

In particular, we can set $x_1 = 0$, then

$$\frac{1-z_0}{z_0}e^{4x_0} = \frac{1-z_1}{z_1}, \tag{199}$$

implying

$$z_1 = \frac{1}{\frac{1-z_0}{z_0}e^{4x_0} + 1} \tag{200}$$

Then, the general solution can be written as

$$f(x_0, z_0) = f(0, z_1)\frac{z_1}{z_0} = f\Big(0, \frac{1}{\frac{1-z_0}{z_0}e^{4x_0} + 1}\Big)\frac{1}{(1-z_0)e^{4x_0} + z_0}. \tag{201}$$

In our setting

$$f(0, z) = M_{0,0} + M_{0,1}z = \frac{z}{2} - 1. \tag{202}$$

It follows that

$$f(x, z) = f(0, z_1)\frac{z_1}{z_0} \tag{203}$$

$$= \frac{2(z-1)e^{4x} - z}{2(z - (z-1)e^{4x})^2}. \tag{204}$$

**Merging the two g.f.'s.** In general, given two sequences $a_n, b_n$ with g.f.'s $A(z), B(z)$ :

$$A(z) \sim \sum_{n=0}^{\infty} a_n z^n, \tag{205}$$

$$B(z) \sim \sum_{n=0}^{\infty} b_n z^n, \tag{206}$$

the g.f. $C(z)$ for the product sequence $a_n b_n$,

$$C(z) = \sum_{n=0}^{\infty} a_n b_n z^n, \tag{207}$$

can be found by

$$C(z) = \frac{1}{2\pi i} \oint_{\gamma} A(\zeta) B\Big(\frac{z}{\zeta}\Big)\frac{d\zeta}{\zeta} \tag{208}$$

with a suitable contour so that the arguments of $A, B$ lie in the convergence discs.

It follows that we can find the full 3-variate g.f. $\Psi$ by

$$\Psi(x, y, z) = \frac{1}{2\pi i} \oint_{\gamma} h(\zeta, y) f\Big(x, \frac{z}{\zeta}\Big)\frac{d\zeta}{\zeta}. \tag{209}$$

We have

$$f\Big(x, \frac{z}{\zeta}\Big) = \frac{\zeta(z(1 - e^{-4x}/2) - \zeta)e^{-4x}}{(\zeta - z(1 - e^{-4x}))^2}, \tag{210}$$

so

$$\Psi(x, y, z) = \frac{1}{2\pi i} \oint_{\gamma} h(\zeta, y)\frac{(z(1 - e^{-4x}/2) - \zeta)e^{-4x}}{(\zeta - z(1 - e^{-4x}))^2}d\zeta. \tag{211}$$

As the contour $\gamma$, we can take a circle $|\zeta| = \epsilon$ with some small $\epsilon$; then the above integral representation for $\Phi$ is valid for all sufficiently small $z$.

The integral is computed by calculating the residue at the pole $\zeta = z(1 - e^{-4x})$ :

$$\Psi(x, y, z) = \partial_\zeta \Big(h(\zeta, y)(z(1 - e^{-4x}/2) - \zeta)e^{-4x}\Big)\Big|_{\zeta = z(1 - e^{-4x})} \tag{212}$$

$$= \Big(\frac{ze^{-4x}}{2}\partial_z h(z(1 - e^{-4x}), y) - h(z(1 - e^{-4x}), y)\Big)e^{-4x}. \tag{213}$$

## K    EXPERIMENTS

**Finite difference integration of gradient flow.**    In order to compare the theoretical continuous gradient flow equation 1 with numerical experiments, we discretize the dynamics by means of an explicit Euler scheme. Concretely, we fix the maximal integration time $t_{\max}$ and the number of steps $N_{\text{steps}}$, which determines the integration step

$$\tau = \frac{t_{\max}}{N_{\text{steps}}}.$$

The continuous flow

$$\frac{du}{dt} = -\frac{1}{T}\,\partial_u L(\mathbf{u})$$

is then approximated by the finite-difference update rule

$$u^{(k+1)} = u^{(k)} - \frac{\tau}{T}\,\partial_u L(u^{(k)}), \qquad k = 0, 1, \ldots, N_{\text{steps}} - 1.$$

Thus the effective learning rate of the numerical scheme is $\eta = \tau/T$, depending jointly on the physical scale $T$ and the discretization step $\tau$.

We emphasize that the explicit Euler discretization above is precisely the standard gradient descent iteration with step size $\eta = \tau/T$. Classical results in numerical analysis show that, under mild smoothness assumptions, the Euler scheme converges to the solution of the underlying gradient–flow ODE with a global error of order $O(\eta)$. Consequently, for sufficiently small $\eta$, the discrete dynamics remain uniformly close to the continuous gradient flow; conversely, gradient flow provides an accurate infinitesimal description of the behavior of discrete gradient descent.

Studying optimization through its gradient–flow limit is a classical and widely used approach in theoretical machine learning, as the continuous dynamics offer a convenient analytical model while discretization errors can be controlled by standard arguments.

**Memory efficient loss computation.**    The evaluation of the quadratic loss equation 2 requires summation over all $p^\nu$ index tuples $(i_1, \ldots, i_\nu)$, which becomes prohibitive to store in memory when both $p$ and $\nu$ are large. To overcome this difficulty, we employ a batching procedure at the level of loss computation. Specifically, we partition the full index set

$$\{1, \ldots, p\}^\nu = \bigcup_{b=1}^{B} \mathcal{B}_b,$$

where each $\mathcal{B}_b$ is a batch of multi-indices. For a given batch $\mathcal{B}_b$ we compute the partial loss

$$L_{\mathcal{B}_b}(\mathbf{u}) = \frac{1}{2} \sum_{(i_1, \ldots, i_\nu) \in \mathcal{B}_b} \left( f_{i_1, \ldots, i_\nu} - F_{i_1, \ldots, i_\nu} \right)^2, \tag{214}$$

and accumulate its gradient contribution. Iterating over all batches and summing their contributions yields exactly the full loss equation 2 and its gradient:

$$L(\mathbf{u}) = \sum_{b=1}^{B} L_{\mathcal{B}_b}(\mathbf{u}).$$

In this way the computation can be performed in a memory-efficient manner, since only one batch is loaded and processed at a time. Importantly, unlike stochastic gradient descent, this batching procedure does not approximate the loss but reproduces it exactly after all batches have been processed, so the optimization dynamics still correspond to the full gradient descent with loss equation 2.

### K.1    UNDERPARAMETRIZED FREE EVOLUTIONS

**Asymmetric models.**    In asymmetric scenario we performed our experiments with $\nu = 3$. For this $\nu$ the requirement for underparametrized regime is $H \ll p^2$, so we scaled $H \sim p$, and performed three experiments with $p = \{32, 64, 128\}$. We also keep the parameter $c = \frac{4p^2\sigma^4}{T} = 0.2$ (constant), so that the loss behavior (after proper normalization) is independent of $p, H, \sigma$ (see Eq. (17)).

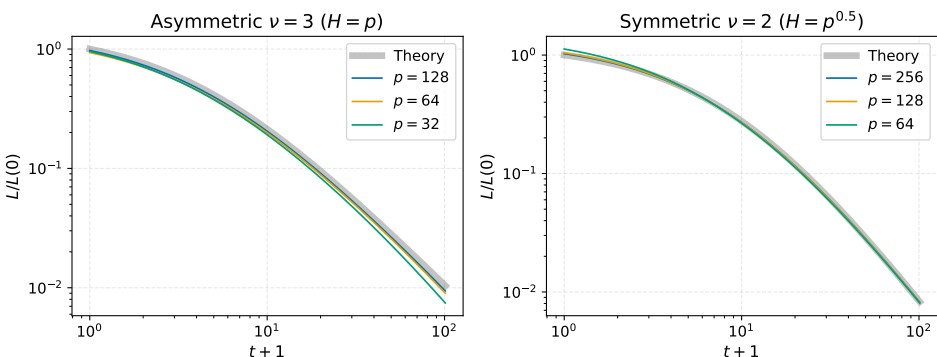

Figure 9: Experimental confirmation of our theory for underparametrized free evolution (Appendix G). Theoretical curves were plot based on Eq. (17). See Appendix K.1 for experimental details.

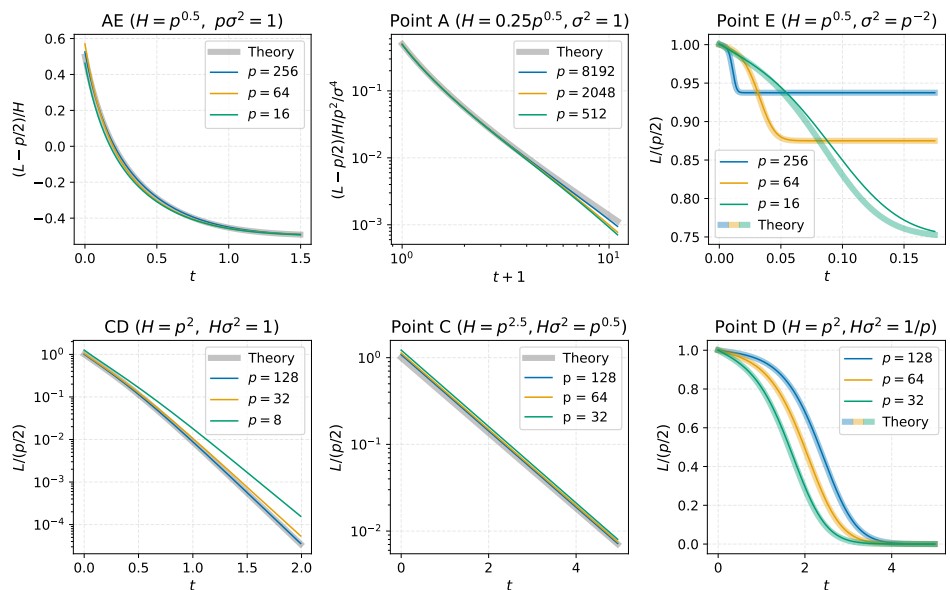

Figure 10: Experimental confirmation of our theory for asymmetric $\nu = 2$ problems. The first panel shows the underparameterized case (Appendix H), where theoretical lines represent Eqs. (20) and (21). The second panel shows the overparameterized case (Appendix I) and theoretical lines represent Eq. (22). See Appendix K.2 for experimental details.

**Symmetric models.** In symmetric case we worked with matrices ($\nu = 2$). We experimented with $p = \{64, 128, 256\}$, while scaling $H \sim p^{0.5}$, so that $H \ll p$. We keep parameter $c = \frac{4p\sigma^2}{T} = 0.1$.

**Results.** The results suggest a well correspondence between theory (Eq. (17)) and experiment, which becomes better with growing values of $p$ and $H$. See Fig. 9.

### K.2 ASYMMETRIC $\nu = 2$ PROBLEM

Our theory for asymmetric $\nu = 2$ case covers under- ($H \ll p$) and overparameterized ($H \gg p$) limits, which are described by the segments E-A and C-D respectively on the pareto-front diagram (see Fig. 2).

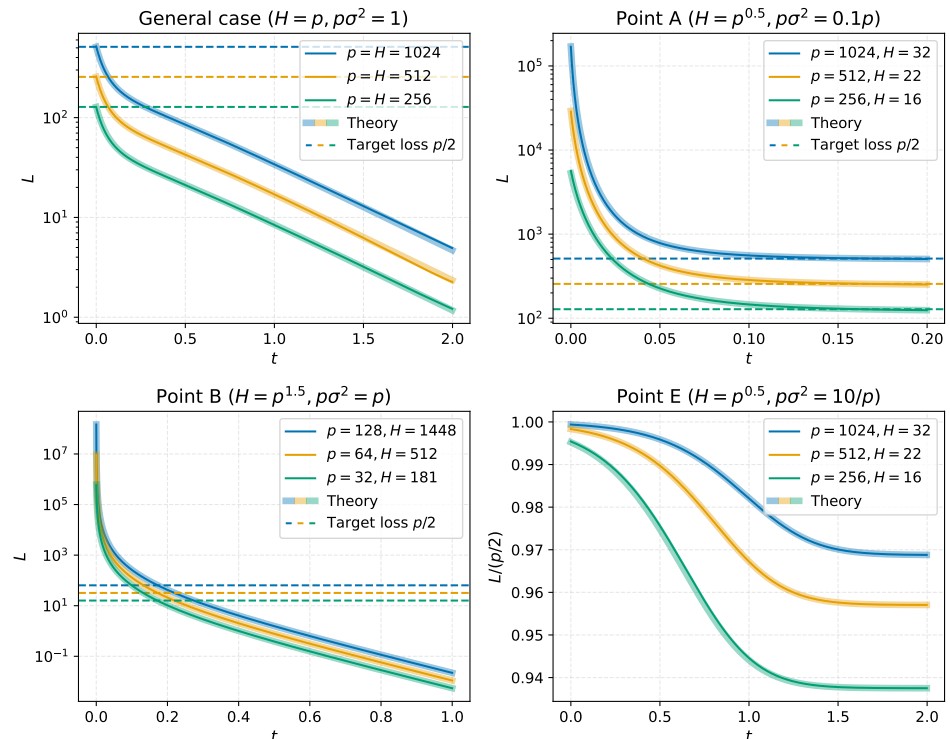

Figure 11: Experimental verification of our theory for symmetric $\nu = 2$ problem (Appendix J). Here theoretical curves represent Eqs. (23) and (28). See Appendix K.3 for experimental details.

**Segment E-A.** Parameter $p\sigma^2$ describes the exact location of a point at the segment E-A on the diagram. In our experiments we covered all three limits of segment E-A:

1. General case with finite $p\sigma^2 = 1$.

2. Point A limit, when $p\sigma^2 = p \to \infty$ as $p \to \infty$.

3. Point E limit, when $p\sigma^2 = 1/p \to 0$ as $p \to \infty$.

In all experiments we set time scale as $T = p\sigma^2$, which is a natural scaling of parameter $T$ in this scenario.

**Segment C-D.** In this overparameterized scenario, parameter $H\sigma^2$ describes the point at the segment C-D. Similarly, we performed three sets of experiments, all with timescale $T = H\sigma^2$ except for point D, where we set $T = 1$.

1. General case with finite $H\sigma^2 = 1$.

2. Point C limit, when $H\sigma^2 = p^{0.5} \to \infty$ as $p \to \infty$.

3. Point D limit, when $H\sigma^2 = 1/p \to 0$ as $p \to \infty$.

**Results.** In certain cases it is possible to renormalize loss in a way that it exhibits a limiting shape independent from $p, H, \sigma$. We tried to always perform such renormalization, when it is possible, in order to see how experimental curves converge to theoretical predictions (Eqs. (20) to (22)) as we grow $p, H$. In short, experiments confirmed our theory. See Fig. 10 for results and specific values of parameters that we used in all experiments.

### K.3 SYMMETRIC $\nu = 2$ PROBLEM

In this part we describe our experiments in case of symmetric model with $\nu = 2$. For this problem we derived the exact formula for the expected loss evolution (see Appendix J), and our goal is to verify these theoretical predictions. The main result for expected loss evolution is valid for any scenario, the only requirement is $p, H \to \infty$, so we performed four different experiment sets under different conditions for parameters scaling and all with timescale $T = 1$ in order to validate our theory.

**General case.** In this scenario, we performed experiments with three different values of $p = \{256, 512, 1024\}$. We set the width $H = p$ and scale parameter variance so that $p\sigma^2 = 1$.

**Point A limit.** This case is a limit of underparameterized free evolution, so we tried $p = \{256, 512, 1024\}$, while we scaled the width $H \sim p^{1/2}$ and keep $\sigma^2 = 0.1$, so that $p\sigma^2 = 0.1p \to \infty$, as $p \to \infty$.

**Point B limit.** Point B is an overparametrized free evolution. We expremented with $p = \{32, 64, 128\}$ and scaled $H \sim p^{3/2}$. Parameter variance was set to $\sigma^2 = 1$, so $p\sigma^2 = p \to \infty$ as $p \to \infty$.

**Point E limit.** Point E is the case of underparametrized theory with the strongest interaction with target. We cosidered $p = \{256, 512, 1024\}$ and scaled $H = p^{1/2}$. We also scaled parameter variance so that $p\sigma^2 = 10/p \to 0$ as $p \to \infty$.

**Results.** We see a great correspondence of experimental lines with theory (Eqs. (23) and (28)) in all cases for all values of $p, H$ that apparantly were chosen sufficiently large (see Fig. 11).

## L   LIMITATIONS

**The identity target.** In this paper, we have focused on the learning target having a particular form of the identity tensor. There were three main reasons for that:

1. Even with this seemingly simple target, the explicit solutions we obtain for the loss dynamics $\mathbb{E}[L(t)]$ are fairly complicated. It seems natural to first study the more tractable scenarios before passing to more complex ones.

2. The identity target is transformed in a natural way with growing $p$. General targets require special (less transparent or justified) assumptions controlling their scaling with $p$.

3. As discusseed in Section B, the identity tensor target is effectively present in some standard learning tasks such as modular addition.

**The model.** We have only considered one particular type of the model (CP-type tensor approximation). We expect that our approach can be extended to other models, such as deep linear networks. Also, we expect it to be possible to extend our approach to neural networks with nonlinear activation functions (say, polynomial).

**Omission of the symmetric scenario with odd $\nu$.** The classification of our Theorem 1 does not cover odd $\nu$ for the symmetric scenario. Our preliminary results suggest that this scenario is more complex and requires some essential additional considerations not relevant for the covered scenarios of asymmetric and symmetric even-$\nu$ models.

