# OpenReview forum: "Solvable Gradient Flow Through Diagram Expansions"
_ICLR.cc/2026/Conference — Submitted to ICLR 2026_

### Official Review · Reviewer_Cfis · 2025-10-25

**Soundness:** 3
**Presentation:** 3
**Contribution:** 2
**Rating:** 4
**Confidence:** 4

**Summary:**

The paper introduces a diagrammatic expansion for analyzing gradient flow learning of order ν identity tensor, from a sum of rank one factors with Gaussian initialization, that yields a formal time power series for the expected loss, whose coefficients are computed by diagram pairings and contractions. This allows to classify scaling regimes, such as NTK, mean field, and no learning, by a hyperparameter polygon, that organizes how the target size p, the model size H, and the initial \sigma, scale. The series are turned into generating functions, that satisfy first order PDEs, which are solved by characteristics, producing closed form learning curves, such as an explicit loss decay in free evolution and solvable expressions in under and over parameterized cases.  These predictions are compared with certain numerical gradient descent experiments and match closely. Summary: the overall contributions are:  (i) a diagrammatic calculus for gradient flow dynamics with explicit coefficient counting, (ii) the hyperparameter polygon taxonomy linking canonical limits, such as NTK and mean field, and (iii) closed form solutions for loss dynamics that align with experiments.

**Strengths:**

(i) Analytic framework that combines a formal loss time series with a diagrammatic calculus to get an explicit dynamics of the expected loss for the identity tensor factorization task. In some regimes, the power series can be extended to a multivariate generating function, that obeys a first order PDE, which can be solved in certain cases. (ii) A hyperparameter polygon, that allows for a classification of large scale limits that recover known regimes, such as the NTK and mean field. (iii) Numerical experiments align well with the theory across regimes.

**Weaknesses:**

(i) A narrow task: the theory is demonstrated on learning the identity tensor, (ii) The analysis requires large p,H, while finite width corrections are not characterized,(iii) The theory is for gradient flow, and the numerics use Euler discretization to mimic it, which differs from practical SGD, (iv) The method is not rigorous, the PDEs for the generating functions are derived from formal series, while convergence and error bounds are not proven.

**Questions:**

Suggestions for improvement of the paper: (i) Make the formal method more rigorous by addining the remainder, when going from the formal series to PDE in one regime, which would elevate the method from formal to controlled, (ii) Quantify the effect of dropping non Pareto optimal terms, by providing bounds, or empirical checks, for the size of discarded terms vs. kept terms across polygon regions, (iii) Add an estimate of the leading finite size corrections to the large p and H formulas, e.g. via next to leading order contractions, and compare to numerics. This will narrow the gap between asymptotics and practice.  (iv) Bridge gradient flow and the discrete training by a discretization error analysis.
(v)  Add, using theory or numerics, an example of a simple non identity structured target.

---

> ### Author Response · Authors · 2025-11-21
>
> Thank you for the very interesting list of suggestions. Please see our general response that clarifies the mathematical aspects and lists many planned extensions of our method.
>
> 1. We certainly agree that making the method controlled is a very desirable objective. We also find your specific suggestions to be on the whole quite reasonable and interesting.
>
> 2. At the same time, we are not convinced of a sufficient outcome/effort ratio for the program that you propose, especially in the context of the present submission. Our paper already contains multiple results, is 53 pages long, and we can barely fit our main findings into the main text. At the same time, your program is no small endeavor; it will likely not be feasible in its entirety, and will probably not change our specific results on the phase structure and analytic solutions.
>
> 3. Our loss expansion is a well-defined asymptotic series, with well-defined and tractable coefficients. Our main contribution is to show that we can extract information about the phase structure and even the large-time loss behavior from this series. These are new and non-obvious results. Our main Theorem 1 on the structure of leading terms is completely rigorous within our asymptotic expansion framework, and we do not think that it is easy.
>
> 4. However, this limiting expansion, while analyzable term-wise, is not even generally classically summable (we do not discuss this point in the paper, but we know this from our further studies); the respective power series has zero convergence radius and requires Borel or other special summation. Theoretical bounds on the remainders seem far from easy in our general $\\nu$ setup. Perturbative diagram-based analysis is a reasonable direction, but also far from easy: even the classification of leading terms in Theorem 1 requires substantial combinatorial effort (and is even trickier in the case of symmetric models with odd $\\nu$, which we do not include in this paper).
>
> 5. The theoretical remainder bounds may be easier to establish in scenarios and regimes with better analytical properties and the availability of additional analytical tools (NTK, free evolutions, $\\nu = 2$). However, this would likely require a case-by-case analysis and would only partially address our phase diagrams. Empirical bounds for the remainders are reasonable, but this does not seem to go far beyond the direct comparison of analytical solutions with numerics, which we already perform.
>
> 6. In summation, we think that the steps you propose are reasonable, but rather in the context of long-term research. As in physics the analysis of Feynman diagram expansions is generally separate from rigorous field-theoretic constructions, we think that your program would be an important but largely parallel line of research.

---

### Official Review · Reviewer_GNcE · 2025-10-25

**Soundness:** 2
**Presentation:** 1
**Contribution:** 3
**Rating:** 4
**Confidence:** 2

**Summary:**

The authors consider a tensorial generalization of the matrix factorization problem in linear deep learning.
In this case, they are able to compute the expected value of the loss during the learning evolution as modeled by gradient flow using Feynman diagram expansion techniques.
Using this expansion, they classify the learning regime of the dynamics (lazy, rich, etc.) depending on the variance used to initialize the model parameters as well as the values of other parameters playing the role of model complexity and target size.

**Strengths:**

* The author are able to provide an explicit formula for the expected value expansion for the loss under gradient flow
* The classification of previously identified learning regimes (lazy, rich, etc.) depending on the model hyper-parameters is very satisfying
* The use of Feynman expansions in this way, and the general approach is novel to deep learning and mathematically interesting

**Weaknesses:**

* The highly mathematical technicality of the paper made it hard for me to assess its correctness. (Part of me thinks this paper may be more appropriate for an applied mathematical journal submission.) The exposition is in general could be improved by focusing in the main paper on a simpler case (for instance matrix factorization), while deferring the more general case of tensor factorization to the appendix.

* The possibility of the Feynman expansion technique seems to rely heavily on the non-linearity of the network. It would help if there were some comments on how to extend the method beyond this linear toy model. Is this approach a dead-end beyond linear networks, or can it be applied to more useful settings? Is it possible to comment on this briefly?

* The learning-regime classification depends on the assumption of gradient flow. However, in practice, optimization is done in discrete steps whose size influences the learning regime (higher learning rates have been show to bias the model trajectories to flatter, more generalizable solutions for example). Can the author comment on the possibility of extending their method to practical discrete optimization?

**Questions:**

See weakness section.

---

> ### Author Response · Authors · 2025-11-21
>
> Thank you for the careful reading of our paper and useful feedback. Please take a look at our general response to reviewers that clarifies some mathematical aspects and lists planned extensions of our results.
>
> > The exposition is in general could be improved by focusing in the main paper on a simpler case (for instance matrix factorization)
>
> Thanks for this suggestion. We have considered this possibility, but we think that restricting to matrices would weaken the paper. It was important for us to show that our asymptotic expansion can be used to classify large-scale learning regimes and explicitly compute the loss of the free evolution even for large tensor orders $\\nu$, where the available analytic methods are scarce.
>
> > Is this approach a dead-end beyond linear networks?
>
> Not at all; it extends naturally at least to networks with polynomial activations. Please see our general response.
>
> > Can the author comment on the possibility of extending their method to practical discrete optimization?
>
> While our framework can in principle cover discrete gradient descent steps, such a formulation is less natural as it does not involve generating functions. However, we emphasize that all expectations that occur when tracing gradient descent steps could be well expressed and computed in terms of diagrams. We also expect our method to be applicable to the noisy (diffusive) gradient flow, akin to SGD, with some analytic noise model.

---

> ### Comment · Reviewer_GNcE · 2025-11-26
>
> I thank the authors for their answers to my questions, and for the additional explanations to all reviewers.
>
> I currently maintain my score, as I believe the current paper would need a significant revision to be widely useful to the ML community.

---

### Official Review · Reviewer_aX8t · 2025-10-28

**Soundness:** 1
**Presentation:** 2
**Contribution:** 3
**Rating:** 4
**Confidence:** 2

**Summary:**

The authors analyze from an asymptotic perspective the gradient flow training dynamics of a tensor approximation problem. Specifically, they consider the problem of approximating in the least squares sense a diagonal identity tensor of order \nu and dimension p by a tensor of rank at most $H$, and focus on the gradient flow dynamics with i.i.d. random Gaussian initialization of variance $\sigma$.
By exploiting the polynomial nature the loss, they express its time derivatives of any order as polynomials and use so-called "diagrams" (that take the form of certain multi-graphs) to provide "formal expressions" of the involved polynomials using a so-called "diagram merging" operations. They then invoke Wick's theorem and the diagram viewpoint to decompose the expectation of these polynomials (with respect to the Gaussian initialization) into terms expressed as $p^q H^n \sigma^{2\ell}$. Invoking diagram arguments that I was not able to follow, they conduct a Pareto-optimal analysis to identify which exponents $q,n,\ell$ can dominate the other, depending on the relative scaling of $p$, $H$ and $\sigma$. This Pareto-optimal viewpoint is then exploited to identify scaling regimes, which in turn allow to express an asymptotic partial differential equation leading to explicit "asymptotic solutions" in certain regimes.

**Strengths:**

It is very welcome to analyze the gradient flow dynamics of large-scale learning problems, and the idea that this can be achieved with certain polynomial loss functions is very attractive. The exploitation of the fact that the time derivatives are also polynomials and that their dominant terms can be identified is also very appealing and interesting.
If the results are correct, they do provide some interesting insight on the behavior expected in different scaling regimes (relating the problem dimension $p$, its rank $H$, the initialization scale $\sigma$, and the learning rate $1/T$).

**Weaknesses:**

The paper heavily relies on concepts, terminology and viewpoints that seems to require a (statistical) physics background, making it very hard to access for a general reader with a standard machine learning or mathematics background.
The use of so-called "diagrams", which are certainly not well-known to the average ICLR audience and are very tersely described, makes it particularly difficult to follow the reasoning. I was not able to check whether the results are correct, due to the heavy use of such informal arguments, combined with "formal expansions" (where actual convergence analysis of series would also seem necessary) and  "non-rigorous" aspects of the proofs in particular when deriving asymptotic PDEs.  These are the main reasons for my grade on "soundness" and "presentation".

While I reckon that the general approach (cf my summary above) *seems* sensible (hence my grade 3 on "contribution"),  it would need to be much more clearly and convincingly conveyed by postponing as much as possible the "diagram" viewpoint and making it much more explicit via polynomials and lemmas where appropriate.

While the title and beginning of the abstract suggest general results on gradient flows for large learning problems, the setting consider in the paper is very specific, focused on very particular tensor optimisation problems with identity target tensor. Despite claims below (3), this seems very far from being representative of general learning problems.

The paper lacks references to the vast literature on the considered tensor optimization problem, which is known under various names such as Canonical Polyadic Decomposition or PARAFAC decomposition, and is known to lead in general to a number of topological difficulties (for example, a minimizer of the loss does not necessarily exist - see e.g. De Silva & Lim "Tensor Rank And The Ill-Posedness Of The Best Low-Rank Approximation Problem"). There is a significant gap between the specific model considered here (in particular, using the identity tensor as a target) and the evoked richness of the "class of problems"  below (3).

The learning rate $1/T$ appearing the gradient flow equation (1) seems superfluous at first sight, since the solution to (1) with $T=1$ yields the solution for general T via a simple time rescaling. It would be helpful to mention this fact while explaining that  $T$ will nevertheless be useful to highlight how certain "time scales" appear in the asymptotic analysis.

**Questions:**

Can the approach be expressed without reliance on "diagrams" but simply on polynomials, using a notion of "polynomial merging" in (7) ?

---

> ### Author Response · Authors · 2025-11-21
>
> Thank you for the careful reading of our paper and useful feedback. Please see our detailed general response to the reviewers that covers many of your concerns. In brief:
>
> 1. We admit that the paper can indeed be improved by using more standard mathematical language and concepts such as polynomials. The mathematical status of different constructions and results can be explained more clearly. We are revising the paper accordingly.
> 2. Our approach is mathematically rigorous and/or consistent within the framework of asymptotic expansions. However, we argue that establishing classical convergence of these expansions is a very challenging problem in general. We do not think that resolving these challenges is a necessary prerequisite for using or analyzing such expansions.
> 3. We list many possible directions along which our study can be extended.
>
> > The paper lacks references to the vast literature on the considered tensor optimization problem
>
> Thanks for pointing this out; we will add these references. We are aware of the ill-posedness of the CP decomposition, but did not discuss it because of the lack of space and our focus on gradient descent for a particular target for which such topological difficulties do not seem to be present.
>
> > It would be helpful to mention this fact while explaining that $T$ will nevertheless be useful to highlight how certain time scales appear in the asymptotic analysis.
>
> Thanks, indeed, the parameter $T$ is important for rescaling and identifying the relevant time scales. We will point this out in the updated paper.
>
> > Can the approach be expressed without reliance on diagrams but simply on polynomials, using a notion of polynomial merging in (7)?
>
> Yes, certainly, but we believe that conceptually the diagrams are easier to perceive than polynomials. For example, the 3rd diagram in the first row of figure 1 can be written as the polynomial
>
> $$
> \\sum_{k_1,k_2,k_3=1}^H \\sum_{i_1,i_2,i_3,i_4,i_5=1}^p
> u_{k_1 i_1}^{(1)} u_{k_2 i_1}^{(1)}
> u_{k_1 i_2}^{(2)} u_{k_2 i_2}^{(2)}
> u_{k_1 i_3}^{(1)} u_{k_3 i_3}^{(1)}
> u_{k_1 i_4}^{(2)} u_{k_3 i_4}^{(2)}
> u_{k_2 i_5}^{(3)} u_{k_3 i_5}^{(3)},
> $$
>
> which looks certainly harder to comprehend. Many typical structures of the polynomial occurring in the study have a suggestive visual diagrammatic interpretation (“flowers”, “trees”, “circles”, etc.; see figure 1).

---

### Official Review · Reviewer_gZLv · 2025-10-30

**Soundness:** 3
**Presentation:** 2
**Contribution:** 2
**Rating:** 4
**Confidence:** 4

**Summary:**

In this work, the authors use novel analytical techniques to study the dynamics of a gradient flow algorithm on a specific learning task, which can be understood as learning the identity tensor using a sum of rank one components. They show that the loss for this
task, when the tensor is order 3, is similar to the population loss obtained for a modular arithmetic task in the Fourier basis, thus arguing that it captures real learning phenomena.
Using a diagrammatic approach that mixes tensor networks and Feynman diagrams, they are able to derive closed form expressions for the time evolution of the loss function averaged over the initialisation in some scaling regimes. Furthermore, they systematically
catalog different learning phases as a function of the scalings between the number of model parameters, the dimensionality of the target to be learned and the variance of the initialisation of the weights.

**Strengths:**

The analytical approach used in this paper is, as far as I can tell, very original and unique and it makes the model analytically solvable in some cases.
The expansion in time, and the subsequent mix of Feynman diagrams and tensor networks is a hybrid approach which combines techniques that come from different fields of computer science and physics. I particularly like that with this single approach, various scaling regimes for the hyper parameters of the model emerge naturally by looking at the dominating diagrams, establishing connections to well known topics such as NTK, Lazy Regime and Feature Learning, and derive natural scalings for the initialisation variance and learning rate.
This is a nice alternative to Dynamical Field Theory for the calculation of the dynamics of gradient descent, and it would be interesting to see if it can be applied for other learning tasks.

**Weaknesses:**

The main weakness is the relevance of the model studied.
The connection to modular arithmetic for $\nu=3$ is interesting, however the link seems vague and very specific. Furthermore, the main interest of such modular arithmetic tasks is
typically the phenomenon of Grokking, which is not studied in the paper.
It would have been nice to see the time evolution of the learning phases for $\nu=3$, and perhaps hope to observe the Grokking phenomenon analytically.
The time evolution is only obtained for the $\nu=2$ case, which I can only connect to two layer linear networks, but with the unusual property that no dataset is present in the learning. Even disregarding this, no particular insights on the dynamics are derived. Ultimately it seems that the main feat of this paper is to derive the various scaling regimes, NTK, Mean Field etc..., with a single approach. I
wonder however if this technique can be used to study other learning models, and perhaps obtain more intuition in the dynamics of learning.
However, the relevance of the studied model seems limited, and no new interesting phenomena is explained using the analytical
solution.
The importance of the paper seems to be more technical than anything else.

**Questions:**

For the presentation:
It would be nice if the notation were more explicitly stated, and quantities listed as vectors, matrices tensors, etc...
For example in the first page we read ${\bf u}=\\{u\\}$, what does this mean?
It seems like a pretty ambiguous notation.
Also on page 2, it is stated that the case $\nu=2$ is equivalent to learning the identity matrix with matrices $UU^T$. What are the dimensions of the matrix $U$? Obviously this can all be derived by the reader, but if you explicitly give the dimensions of the matrix and state how this is connected to a deep linear network it makes it easier to read.
The literature review in the appendix is very long, and touches many topics which are only mildly related to the paper. I would suggest shortening it, and maybe inserting some of the most important references in the main text.
For the relevance for the paper:
I wonder if the case $\nu=2$, for which most of the results are derived, can also be connected to other known learning tasks, as in the case $\nu=3$. This would give more relevance to the results of this paper. The symmetric case for example seems similar to an a linear autoencoder, can the authors comment on this?
Can you find some interesting phenomena in your solved trajectories which can be maybe observed in more realistic learning tasks?
A numerics section would give the paper more concreteness, and show that this diagrammatic method is really explaining some phenomena.

---

> ### Author Response · Authors · 2025-11-22
>
> Thank you for the careful reading and comprehensive feedback on our paper. Please see our general response to the reviewers, where we discuss mathematical aspects of our work and list various planned extensions.
>
> > For the presentation: It would be nice if the notation were more explicitly stated...
>
> Thanks for these suggestions on improving the text; we agree and are currently polishing the paper.
>
> > The literature review in the appendix is very long, and touches many topics which are only mildly related to the paper. I would suggest shortening it..
>
> Again, we agree and are currently cutting the least relevant pieces.
>
> >I wonder if the case $\\nu = 2$, for which most of the results are derived, can also be connected to other known learning tasks, as in the case $\\nu = 3$. This would give more relevance to the results of this paper. The symmetric case for example seems similar to an a linear autoencoder, can the authors comment on this?"
>
> The matrix case (not necessarily symmetric) indeed describes a linear autoencoder as soon as we introduce a Gaussian dataset. If we, in contrast, introduce a dataset of one-hot vectors, as we describe in our general comment, and a non-trivial target, we get a well-known problem of *matrix completion*. We mention also that a more applied problem of *image inpainting* is a special case of matrix completion.
>
> > However, the relevance of the studied model seems limited, and no new interesting phenomena is explained using the analytical solution. The importance of the paper seems to be more technical than anything else. "
>
> We don't think that this description of our work is very fair. Our focus is not on explaining a specific practical phenomenon - we would need to write a different paper for that. Our contribution is a new analytic framework. It includes new conceptual and mathematical tools allowing to address and systematically and specifically answer questions on the phase structure and loss evolution in large learning tasks. In the particular example of "identity learning" analyzed in this paper we reveal a rich phase structure (even two different structures, depending on the symmetric/asymmetric scenario) and detailed descriptions of loss trajectories in some regimes. None of these results is obvious. The possibility to accurately recover the $t\to\infty$ dependence of $L(t)$ from its Taylor expansion at $t=0$ is not obvious. We don't see why our contribution is technical and lacking interest.
>
> While the present paper is restricted to the identity target as the simplest example, in our general response to reviewers we list many possible extensions.
>
> > The time evolution is only obtained for the $\\nu = 2$ case, which I can only connect to two layer linear networks, but with the unusual property that no dataset is present in the learning. Even disregarding this, no particular insights on the dynamics are derived."
>
> Our method allows for introducing a Gaussian dataset, and study both the train and the generalization losses, as we explain in our general comment. We do not agree that no insights on the dynamics are derived. The solution for the symmetric matrix case is highly non-trivial and by no means described by exponential or any other simple functional dependencies. Same goes for explicit solutions for the mean-field and underparameterized regimes in the asymmetric case. Also, we are not aware of any studies of free evolutions; as our explicit solution shows, the loss decays as a power-law, which is far from obvious.
>
> > I wonder however if this technique can be used to study other learning models, and perhaps obtain more intuition in the dynamics of learning.
>
> Yes, please see our general comment, section "Extensions of our theory". In particular, our method is compatible with deep fully-connected networks with polynomial activation functions trained on Gaussian data.
>
> > Can you find some interesting phenomena in your solved trajectories which can be maybe observed in more realistic learning tasks?
>
> 1. Our asymmetric model has an NTK learning regime, while the symmetric, even-$\nu$ model does not.
>
> 2. The "free evolution" regime is relevant for any target whenever the randomly initialized weights are large so that the magnitude of the initial tensor implemented by the model is much larger than the magnitude of the target. In this case, in the initial stage of learning the target does not significantly affect the learning process, and the model approximately "self-deflates" to the zero tensor.
>
>     In Theorem 3 we show that the underparameterized free evolution has the same power-law "deflation rate" in both symmetric and asymmetric scenarios. However, for the overparameterized free evolution the rates are drastically different. The overparameterized symmetric model still has the same power law rate, while the asymmetric model has a fast, exponential rate. (We didn't put the overparameterized result in the paper because of the lack of space.)

---

### Official Review · Reviewer_No34 · 2025-11-02

**Soundness:** 3
**Presentation:** 3
**Contribution:** 2
**Rating:** 4
**Confidence:** 4

**Summary:**

This paper studies the gradient flow dynamics of learning a generic tensor under squared loss, aiming to provide insights on the gradient descent dynamics in large learning problems. To be specific, the authors consider a standard gradient flow with learning rate $1/T$: $$\frac{du}{dt} = - \frac{1}{T} \partial_u L(u),$$ where the weights of the model $u$ is a tensor. The target is assumed to be the identity tensor, and the loss $L$ is the standard quandratic loss. The main contributions of this paper are as follows: (i) The authors use a technique, called "diagram calculus", to computed the expected loss function at time $t$, in order to analyze the gradient flow dynamics. The exact formula derived turns out to be connected to the Feynman diagram; (ii) The contribution of each diagrammatic term in the expansion scales with the problem's parameters. By identifying the corresponding hyperparameter polygon, the paper identifies several scalings of the learning regime, e.g., NTK, mean-field and free evolutions; (iii) The paper also explores several special cases, where the gradient flow has a formal solution gives by power series, or an explicit solution given by the method of characteristics.

**Strengths:**

The use of "diagram calculus" to study gradient flow dynamics is interesting and novel, especially it is able to provide explicit, analytical expressions for $\mathbb{E} [L(t)]$ the expected loss of gradient flow at time $t$. It is also interesting to see that different learning regimes correspond to different "hyperparameter polygons" in the diagram.

**Weaknesses:**

I have a few concerns regarding the rigor and generality about the main results of this paper: (i) Lack of rigor. The main theorem (Theorem 1) is a mathematically rigourous derivation of power series expansion of $\mathbb{E} [L(t)]$. However, the proof of Theorem 2 is only a formal power series derivation rather than a rigorous proof. (ii) Limited generality. The entire theory, although elegent, is only developed for learning a specific tensor model with quadratic loss. It would be better to discuss more examples from deep learning for which the abstract theory from this paper can be applied to investigate the learning behavior. Otherwise, it might be clear why it is interesting to develop this "diagram calculus"-based method for studying gradient flow dynamics.

**Questions:**

Below are some specific comments and questions for the authors:
(i) On the bottom of page 6, partial differential equation is abbreviated as PDF. This also occurs in several other places in the paper, please correct.
(ii) Some appendix numbers are not properly referenced, e.g., in the statement of Theorem 1.
(iii) The main results are stated for expected loss, namely $\mathbb{E} [L(t)]$. Since the randomness here comes from the initialization, which is Gaussian, I was wondering whether there is a high-probability statement for $L(t)$? For example, some Gaussian concentration inequalities might help.

---

> ### Author Response · Authors · 2025-11-21
>
> Thank you for the careful reading of our paper and for your feedback. Thanks also for pointing out the typos and missing references; we will certainly fix them.
>
> Regarding your comments on (i) lack of rigor and (ii) limited generality, please see our common response to reviewers:
>
> 1. We admit that the exposition needs some polishing and clarifications; the revision is under way.
> 2. We clarify the status of our mathematical results. In particular, we argue that our framework of large-model asymptotic expansions is consistent and constructive; the results obtained within this framework are rigorous and non-obvious; however, a rigorous justification of this framework itself seems to be a major challenge going beyond this paper.
> 3. There are in fact many natural directions in which our results can be extended and generalized.
>
> >I was wondering whether there is a high-probability statement for $L(t)$?
>
> This is a very good question. Our framework allows us to analyze the relative magnitude of the squared expected loss $(\mathbb{E}[L(t)])^2$ and the variance $\\mathbb{E}[L^2(t)] - (\\mathbb{E}[L(t)])^2$ on the level of asymptotic diagram expansions, similarly to how we analyze just the expected loss $\\mathbb{E}[L(t)]$. The expressions $(\\mathbb{E}[L(t)])^2$ and $\\mathbb{E}[L^2(t)] - (\\mathbb{E}[L(t)])^2$ admit similar asymptotic expansions in $t$, and concentration can be studied by comparing the respective leading terms in the expansion coefficients.
>
> The diagram expansion for $\\mathbb{E}[L^2(t)]$ is represented by suitably contracted pairs of diagrams of $L(t)$. Then, the difference $ \\mathbb{E}[L^2(t)] - (\\mathbb{E}[L(t)])^2$ corresponds to those contractions of these pairs that connect one diagram to another. The variance being small compared to $(\\mathbb{E}[L(t)])^2$ means that these connecting contractions are dominated by non-connecting (factorizable) ones as $H,p \\to \\infty$.
>
> We have preliminary results suggesting that, indeed, the variance $\\mathbb{E}[L^2(t)] - (\\mathbb{E}[L(t)])^2$ is dominated by $(\\mathbb{E}[L(t)])^2$ in this sense, but this effect is tricky and may not hold in all scenarios that we consider; this requires more careful study. Nevertheless, we do not foresee any conceptual difficulties in this line of work.

---

### Author Response · Authors · 2025-11-21
**General Response to Reviewers (Part IV)**

### Generalization

Our method naturally applies to optimization problems on **finite datasets** and allows one to track both the empirical (**train**) and the distribution (**test**) losses. We list two classes of such problems below.

1. **Modular arithmetic.** For such problems, we could think of one-hot vectors as model inputs, so the training dataset $S$ is nothing but a subset of entries of the target tensor. The $\\nu$ corresponding $p$-vertices in the diagrams associated with the training loss are tied together with this training subset; now they sum to $|S|$ together instead of $p^\\nu$ when there were no ties. One merges diagrams as before, but keeps the ties induced by the training dataset. Note that the diagrams associated with the test loss do not impose any constraints.

2. **Supervised learning with Gaussian data.**
   The simplest example is linear regression with

   $$L_{\\text{train}}(u) = \\frac{1}{2N} \\left\\| X^\\top u - X^\\top w \\right\\|_2^2,$$

   and

   $$
   L_{\\text{test}}(u)
   = \\frac{1}{2} \\mathbb{E}_{x \\sim \\mathcal{N}(0,I_p)}\\left[\\left( x^\\top u - x^\\top w \\right)^2\\right]
   = \frac{1}{2} \\left\\| u - w \\right\\|_2^2,
   $$
   where $X \\in \\mathbb{R}^{p \times N}$ is the training dataset of size $N$ and $u,w \\in \\mathbb{R}^p$ are model and teacher parameter vectors, respectively. We do not train $X$ and $w$, while sampling their entries independently from $\\mathcal{N}(0,1)$. The above losses result in diagrams with three types of nodes, $p$-, $N$-, and output, and three types of edges, $X$-, $u$-, and $w$-. Since we train only the vector $u$, we are allowed to merge diagrams only by $u$-edges, while we pair all same-type edges as before. The same approach generalizes naturally to **linear nets of any depth**.

### Nonlinear models

Our approach is **not restricted to linear models** either: it naturally covers supervised learning problems with models with **polynomial activation functions**. One of the simplest examples of such problems would be learning random Gaussian labels on a random Gaussian dataset with a two-layer network with quadratic activations:
$$
L(U,v)
= \\frac{1}{2N} \\left\\| v^\\top (U X)^{\\odot 2} - y^\\top \\right\\|_2^2,
$$
where $X \\in \\mathbb{R}^{p \\times N}$ is a Gaussian dataset of size $N$, $U \\in \\mathbb{R}^{H \\times p}$ and $v \\in \\mathbb{R}^H$ are model parameters to learn, and $y \\in \\mathbb{R}^N$ is a vector of random Gaussian targets. We could as well introduce a teacher instead of random targets.

---

### Author Response · Authors · 2025-11-21
**General Response to Reviewers (Part III)**

## Extensions of our theory

While in the present manuscript we study only the problem of identity tensor decomposition, the method of diagram expansions we use could be well applied to a much wider class of problems. Below we list several examples of such problems that might be interesting to the Machine Learning community.

### Alternative targets

1. **Our method is by no means restricted to the identity target.** We could as well consider any target that is a Kronecker delta of a system of linear conditions on tensor indices modulo $p$. For example, the problem of summing $\\nu - 1$ integers modulo $p$ gives an order-$\\nu$ target tensor with indices $i_1,\ldots,i_\nu$, with ones at positions satisfying $\\sum_{k=1}^{\\nu-1} i_k = i_\\nu \\mod p$ and zeros elsewhere.

   The corresponding loss function still induces two diagrams: one, denoted by $D_{2\\nu}$, is “free” — it encodes interactions of the model with itself but not with the target. The other one, denoted by $R_\\nu$, encodes interactions of the model with the target. The free diagram stays the same as for the identity target, while the interacting one changes: now it consists of $\\nu$ distinct $p$-vertices tied with a condition.

   We proceed as before: we merge subsequent diagrams by an edge so that endpoints match. We then pair the remaining edges. The conditions remain and define how the remaining $p$-nodes should be summed: they sum into their number minus the rank of the system of conditions.

2. Apart from zero-one target tensors, our method complies with target tensors taken from a sequence of deterministic **trace-class operators indexed with $p$**: $\\{F_p\\}_{p=1}^\\infty$. An example of such sequence is a sequence of diagonal operators with **eigenvalues following a power-law**: such power-laws are ubiquitous in real-life data, giving a common assumption for target spectrum. We have to supply each $p$-node with a power $q$ of the target tensor associated with this node. Though all $p$-nodes have $q = 1$ before contractions, when two nodes contract, their powers add. Each contracted node induces $\\mathrm{tr}[F_p^q]$ instead of $p$ in the sum.

3. Our method is also compatible with **random** targets that can be expressed as **products of Gaussian matrices**.
   As an example, consider **Wishart ensemble decomposition**:

   $$L(U) = \\frac{1}{2} \\left\\| U^\\top U - W^\\top W \\right\\| \\to \\min_U,$$

   where we sample the entries of $W$ independently from a Gaussian and do not optimize them. The above problem statement generalizes naturally to higher-order symmetric and asymmetric tensors. When both dimensions of $W$ grow proportionally, the limit spectrum of $W^\\top W$ does not concentrate at any finite set of points.

4. The three classes of target tensors mentioned above do not cover the whole class of targets compatible with our method. We expect any **linear combination** of compatible target tensors to be also compatible. The resulting “mixed” target tensors might induce rich spectral features, combining continuous parts with multiple discrete eigenvalues. As observed for linear networks initialized close to the origin, the target principal components are learned in a strictly sequential manner starting from the strongest. We therefore expect the loss evolution, which our method explicitly gives, to decompose into distinct phases according to the current eigenvalue to learn.

---

### Author Response · Authors · 2025-11-21
**General Response to Reviewers (Part II)**

**Proofs of convergence.** Several reviewers indicate that it is desirable to provide rigorous proofs of convergence of our asymptotic power series and/or estimate the associated remainders.

- We fully agree that this would be very desirable, but we believe that this objective is quite difficult and most likely not even generally feasible. The limiting asymptotic expansion that we propose is generally well-defined term-wise, but need not be convergent in the usual sense. In particular, we have preliminary results for some models and regimes where the limiting asymptotic power series has a zero convergence radius; accordingly, the series does not converge in the usual sense (but can be summed for $t$ in a suitable domain $D \\subset \\mathbb{C}$ by, for example, our summation method).
- We can reproduce some of our PDE-based explicit formulas for $\\mathbb{E}[L(t)]$ using random matrix theory. This can be seen as a validation of our approach. However, we can do this only on a case-by-case basis, not by some general theory. Because of this and a lack of space, we have not included these derivations in the paper.
- These difficulties closely parallel the well-known difficulties associated with Feynman diagram expansions in quantum field theory and statistical physics. While present in most textbooks, these expansions are notoriously hard to rigorously justify. Usually these expansions are only considered term-wise, as in our case. Rigorous constructions of interacting theories are normally performed by other methods such as cluster expansions; only after the theory is rigorously constructed it can sometimes be connected to a Feynman diagram expansion [1]. Some theories have never been rigorously constructed: for example, this question for the Yang–Mills theory is one of the six unsolved Millennium problems [2]. Feynman power series typically have zero convergence radius and require Borel or other special summation methods. Nevertheless, the nonrigorous Feynman diagram expansions are much more common than rigorous cluster expansions in physics textbooks, since they are conceptually and computationally most natural and simple, and streamline the analysis of various phenomena (in particular scaling-related).
- Similarly, in our ML setting we do not think that the difficulties of classical summation should prevent us from analyzing the limiting asymptotic expansion. We demonstrate that this expansion is a natural, well-defined and useful constructive object. Whether or not it converges in the classical sense, it allows us to systematically derive non-obvious results on the phase structure and in some cases even on the large-$t$ behavior of the model.

[1] V. Rivasseau. Constructive field theory in zero dimension. *Advances in Mathematical Physics*, 2009(1), 2009.

[2] <https://www.claymath.org/millennium/yang-mills-the-maths-gap/>

---

### Author Response · Authors · 2025-11-21
**General Response to Reviewers (Part I)**

We thank the reviewers for a careful reading of our paper. We are impressed by the high quality of all the reviews and sincerely appreciate the feedback. We address the common concerns of the reviewers below.

## Mathematical clarity and rigor

Several reviewers express concerns regarding mathematical aspects of the exposition (clarity and rigor). We accept some parts of this criticism (particularly with regard to clarity).

**General exposition.** Several reviewers point out that the exposition is too terse and relies on informal arguments in the diagram language foreign to ML audience; the focus can be shifted towards standard mathematical language. We tend to agree with this point and are now revising the paper to make it more easily readable.

**Mathematical status of the results.** We also admit that the mathematical status of specific results presented in the paper can be described more clearly. We are revising the paper to this end. Below we emphasize some key points.

1. The expansion of the expected loss $\\mathbb{E} [L(t)]$ studied in the paper is mathematically well-defined for finite models as an *asymptotic series* at $t \\to 0$.
2. We then study the large-model/target limit of this asymptotic series; it is mathematically well-defined *term-wise* (under appropriate time rescaling).
3. We show that this term-wise limit of the loss expansion *contains important information* about the evolution of large-size models. In particular, we show that there is a *mathematically consistent and systematic way* to identify *different large-scale learning regimes* by associating them with different subsets of leading monomials in the loss expansion coefficients.
4. Accordingly, we give a *complete and rigorous classification* of these regimes for two scenarios (asymmetric and symmetric with even $\\nu$). The resulting sets of regimes are *different* (no NTK regime in the symmetric, even $\\nu$ scenario). We expect that the set of regimes can be even richer in more complex models.
5. We also show that, at least in certain scenarios and regimes, the limiting loss expansion admits a *formal summation*. We propose a general summation method based on connecting the coefficients by recurrences and solving a first-order PDE. This method is *not universally applicable*, but we show that it is applicable in several scenarios and produces *nontrivial quantitative predictions* (analytic solutions) that *agree very well* with the experiment.

In summation, we propose a *new and mathematically consistent methodology* that produces *nontrivial new results*. Within this methodology — i.e., accepting the limiting power series expansion and our PDE-based summation method — we provide *rigorous proofs* of the large-scale phase structure for several specific models and *analytic derivations* of explicit solutions for several learning regimes.

---

### Author Response · Authors · 2025-12-03
**Changes in the latest revision**

Following the reviewers' suggestions, we
1. Removed the least relevant parts of the literature overview.
2. Included a section on canonical polyadic tensor decomposition there.
3. Clarified a relation of our gradient flow problem statement to gradient descent.
4. Introduced several minor corrections.

---

### Meta-Review · Area_Chair_fobU · 2025-12-17

**Summary:**

The paper proposes a novel analytic framework using diagrammatic expansions (Feynman diagrams) to study the gradient flow dynamics of tensor decomposition problems. Specifically, it focuses on learning an identity tensor using a sum of rank-one tensors with Gaussian initialisation. The authors derive formal power series for the expected loss, identify various learning regimes (ie NTK, mean-field, free evolution) based on hyperparameter scaling, and provide explicit analytical solutions for loss evolution in certain regimes.

Reviewers broadly appreciated the originality of the framework and the classification of learning regimes. The main concerns, however, relate to the limited rigour and generality of the results, as well as issues of clarity and presentation. While the authors have made a substantial effort to address these points in the revision, some of the underlying concerns remain. As a result, even accounting for potential score updates, the paper still lies near the threshold between acceptance and rejection.

In light of the critiques which will persist even after the authors' response, I recommend rejection.

**Reviewer Concerns:**

### Strengths

1. **Novelty of the approach.** The use of *diagram calculus* was praised as original and mathematically interesting by Reviewers No34, gZLv, and GNCE.
2. **Categorization of learning phases** (NTK, mean-field, lazy vs. rich) using "hyperparameter polygons" was praised by Reviewers No34 , gZLv , GNCE , and Cfis.

### Weaknesses

1. **Lack of rigour.** Reviewer No34, aX8t, and Cfis flagged the reliance on "formal" expansions without rigorous convergence proofs. *The authors argued that proving convergence is "most likely not even generally feasible" and that formal series are standard in physics.*
2. **Limited generality and relevance.** Reviewer gZLv, aX8t, GNCE, and No34 felt the model (learning identity tensor) had limited relevance and the connection to broader tasks, furthermore it is unclear how the result extend beyond linear networks. *The authors provided explanations of how the method could be extended to other problems (ie matrix completion, polynomial activations) in their rebuttal comments.*
3. **Clarity.** Reviewer aX8t, gZLv and GNCE complained that the heavy reliance on physics jargon and ambiguous notation. *The authors agreed to polish notation, fix typos, and shorten the appendix. They also acknowledged the need to add missing references.*

**Reviewer Scores:**

All reviewers gave 4 (Marginally below acceptance threshold).
Only reviewer GNcE confirmed the original score.

### Speculations

- **Reviewer No34 and aX8t**: Their main concerns were lack of rigor and limited generality. Since, the authors admitted rigour is not feasible and defended their generality without changing the core manuscript structure. It is unlikely this will lead a score increase.
- **Reviewer Cfis**: Suggested to improve the paper by adding error bounds/remainders to make it controlled rather than just formal. The authors explicitly rejected this request, stating the "outcome/effort ratio" was insufficient and it was "not even generally feasible." Therefore, their score will likely remain a 4.
- Finally **Reviewer gZLv**: Since their concerns on clarity and generality have been addressed (at least partially), the score could increase.

---

### Decision · Program_Chairs · 2026-01-26

Reject